# Microphysical Processes of Super Typhoon Lekima (2019) and Their Impacts on Polarimetric Radar Remote Sensing of Precipitation

Yabin Gou[1, 2, 3, 4], Haonan Chen[5], Hong Zhu[1], and Lulin Xue[6]

[1]Hangzhou Meteorological Bureau, Hangzhou 310051, China
[2]Zhejiang Institute of Meteorological Sciences, Hangzhou 321000, China
[3]Plateau Atmosphere and Environment Key Laboratory of Sichuan Province, School of Atmospheric Sciences, Chengdu University of Information Technology, Chengdu 610225, China
[4]Department of Geoscience and Remote Sensing, Delft University of Technology, Stevinweg 1, 2628 CN Delft, The Netherlands
[5]Colorado State University, Fort Collins, CO 80523, USA
[6]National Center for Atmospheric Research, Boulder, CO 80307, USA

*Corresponding author address*: Haonan Chen (haonan.chen@colostate.edu)

**Abstract.** The complex precipitation microphysics associated with super typhoon Lekima (2019) and its potential impacts on the consistency of multi-source datasets and radar quantitative precipitation estimation were disentangled using a suite of *in situ* and remote sensing observations around the waterlogged area in the groove windward slope (GWS) of Yan Dang Mountain and Kuo Cang Mountain, China. The main findings include the following: (i) The quality control processing for radar and disdrometers effectively enhanced the self-consistency between radar measurements, such as radar reflectivity ($Z_H$), differential reflectivity ($Z_{DR}$), and the specific differential phase ($K_{DP}$), and the consistency between radar, disdrometers, and gauges. (ii) The microphysical processes, in which breakup overwhelms coalescence in the coalescence-breakup balance of precipitation particles, noticeably make radar measurements prone to be breakup-dominated in radar volume gates, which accounts for the phenomenon where high number concentration rather than large size of drops contributes more to a given attenuation-corrected $Z_H$ ($Z_H^C$) and the significant deviation of attenuation-corrected $Z_{DR}$ ($Z_{DR}^C$) from its expected values ($\hat{Z}_{DR}$) estimated by DSD-simulated $Z_{DR}$-$Z_H$ relationships. (iii) The twin-parameter radar rainfall estimates based on measured $Z_H$ ($Z_H^M$) and $Z_{DR}$ ($Z_{DR}^M$), and their corrected counterparts $Z_H^C$ and $Z_{DR}^C$, i.e., $R(Z_H^M, Z_{DR}^M)$ and $R(Z_H^C, Z_{DR}^C)$, both tend to overestimate rainfall around the GWS of YDM, mainly ascribed to the unique microphysical process in which the breakup-dominated small-sized drops above transition to the coalescence-dominated large-sized drops falling near the surface. (iv) The improved performance of $R(Z_H^C, \hat{Z}_{DR})$ is attributed to the utilization of $\hat{Z}_{DR}$, which equals physically converting breakup-dominated measurements in radar volume gates to their coalescence-dominated counterparts, and this also benefits from the better self-consistency between $Z_H^C$, $\hat{Z}_{DR}$ and $K_{DP}$, and their consistency with the surface counterparts.

**Keywords:** polarimetric radar; remote sensing of precipitation; quality-control; typhoon microphysics; extreme weather.

# 1. Introduction

Weather radars form the cornerstone of national weather warnings and forecast infrastructure in many countries. Doppler radar networks play an indispensable role in modern meteorological and hydrological applications, such as quantitative precipitation estimation (QPE) in support of the application of some hydrological models for water resource management, especially during high-impact weather events in urban environments (Chandrasekar et al.,2018, Cifelli et al., 2018, Chen et al., 2018). Although technological advances such as dual-polarization have tremendously improved weather radar applications in hydrometeorology remote sensing, it is still a challenge to incorporate complex precipitation dynamics and microphysics in an adaptive manner to optimize the quantitative applications of polarimetric radar measurements, including horizontal reflectivity $Z_H$, differential reflectivity $Z_{DR}$, copolar correlation coefficient $\rho_{HV}$, differential propagation phase $\Phi_{DP}$ and its range derivative $K_{DP}$ (specific differential phase). Traditional utilization of these measurements has only been able to extract some information on complex spatiotemporal precipitation variability.

In general, three main factors contribute to radar QPE uncertainties: radar measurement error, parameterization error of various radar-rain rate ($R$) relationships, and random error. In practical applications, it is crucial to consider these three factors as a whole to ensure radar rainfall estimates approximate the surface rainfall truth as much as possible. Among conventional radar QPE algorithms, those developed based on $Z_H$ measurements are typical and are still in use today. For instance, an earlier version of the radar QPE algorithm in the National Oceanic and Atmospheric Administration (NOAA) Multi-radar Multi-Sensor System (MRMS) and its refined version both utilize multi-radar hybrid $Z_H$ to derive the radar-based rainfall field (Zhang et al., 2011, 2016). The recent update of MRMS further incorporated specific attenuation ($A_H$) and $K_{DP}$ to enhance the $Z_H$-based algorithm (Wang et al., 2019; Ryzhkov et al., 2022), and such an update can benefit from (i) the insensitivity of $A_H$ to raindrop size distribution (DSD) variability (Ryzhkov et al., 2014); (ii) $K_{DP}$ is a better indicator of rain rate and liquid water content (LWC, g·m$^{-3}$) than $Z_H$ since $K_{DP}$ connects more tightly to the precipitation particle size distribution; (iii) $R(K_{DP})$ and $R(A_H)$ inherit the immunity of $\Phi_{DP}$ to miscalibration, attenuation, partial beam blockage, and wet radome effects (Park et al., 2005; Ryzhkov et al. 2014, 2022), which are hard to address when using $Z_H$ for radar QPE, especially at higher frequencies such as C- and X-bands (Park et al., 2005; Matrosov.2010; Frasier et al., 2013). However, since $A_H$ and α are simultaneously derived, $R(A_H)$ partly inherits the sensitivity of α to temperature(Ryzhkov et al., 2014), which occurs with the ascending altitude of the propagation of one radar beam. Multi-parameter radar QPE algorithms further integrate $Z_{DR}$ with $Z_H$, $K_{DP}$, or $A_H$ to infer more information about raindrop shape, such as the double-measurement algorithm $R(Z_H, Z_{DR})$, $R(K_{DP}, Z_{DR})$, $R(A_H, Z_{DR})$ and the triple-measurement radar QPE algorithm as $R(Z_H, Z_{DR}, K_{DP})$ (Matrosov,2010; Gosset et al., 2010; Schneebeli and Berne, 2012, Keenan et al., 2001; Chen et al., 2017; Gou et al.,2019), but these algorithms all assume that $Z_{DR}$ measurements are well calibrated and attenuation-corrected (Ryzhkov et al., 2005; Bringi et al., 2010).

In addition to radar measurements, disdrometer and rain gauge data are often used to determine the optimal parameters of radar-based QPE algorithms (Lee and Zawadzki, 2005; Tokay et al.,2005). For example, the MRMS system utilizes long-term $Z_H$ and gauge rainfall measurements to obtain climatological *Z-R* relationships for each precipitation type (Zhang et al., 2011,2016). In Gou et al. (2018, 2020), rain gauge measurements are used to dynamically adjust *Z-R* relationships to reflect the microphysical evolutions of precipitation systems. Nevertheless, the accuracy of meteorological gauge rainfall recordings is usually configured as 0.1 mm, and rain gauges may record less rainfall than reality due to debris blockage (tree leaves, insects, etc.) and the quick spinning of tipping buckets in a heavy shower situation. In addition, the surface wind may hinder some raindrops from falling into the tipping bucket, and the mechanical failures of the tipping bucket will record abnormally high or low rainfall, which introduces significant errors to the gauge network. Similarly, disdrometer measurements can be affected by strong winds and mixed-phase hydrometeors falling through the laser sampling area of the disdrometer, resulting in degraded quality of the DSD recordings (Tokay and Bashor, 2010). Since the DSD data collected by disdrometers are indispensable and sometimes are the only resources that can be used for precipitation microphysical analysis and the establishment of polarimetric radar rainfall relationships, meticulous quality control (QC) must be conducted on the disdrometer measurements (Friedrich et al., 2013).

Another issue that is important but rarely considered in radar QPE is the changing microphysics that occurs during the falling processes of precipitation particles between radar volume gates and surface ground, which is often indicated by inconsistent radar observations with their surface counterparts. The $Z_H$ measurements in the melting layer (ML) of a stratiform rain system, which features falling melting snowflakes or ice crystals, usually need to be corrected for the bright band for subsequent rainfall retrievals, especially when little rain is reported on the ground (Chen et al., 2020). A severe updraft may introduce a large $Z_H$ and $Z_{DR}$ column (Snyder et al., 2015; Carlin et al., 2017), while the surface rain gauge may record little or time-lagged rainfall, which is frequently perceived in the front of a squall line system or wind gust system. In addition, the contamination of mixed-phase hydrometeor particles on $K_{DP}$ and $A_H$ may lead to $R(K_{DP})$ and $R(A_H)$ being overestimated (Gou et al., 2019b), and the falling wet hailstones may also contaminate radar-measured $Z_H$, $K_{DP,}$ and $A_H$ (Donavon and Jungbluth, 2007; Ryzhkov et al., 2014), leading to an overestimated hotspot on the derived rainfall field if such contaminations are not well addressed.

The complex microphysical variations mentioned above may coexist in a large-scale precipitation system such as a typhoon. Before the polarimetric update, the impacts of the coexisting precipitation types on the radar QPE can be exploited through the vertical profile of reflectivity (VPR, Xu et al., 2008; Zhang et al.,2011, 2016). During the landfall of typhoon Hakui (2012), the VPR characteristics of coexisting tropical, convective and stratiform rain account for the spatial precipitation variability (Gou et al., 2014). Super Typhoon Lekima (2019) was the first super typhoon that landed on the eastern coast of Zhejiang after the polarimetric radar update, which provided an opportunity to exploit more microphysical signatures of the typhoon. Lekima landed on the coastal area of Chengnan town in WL city at 1745 UTC, 9 August 2019, and the maximum wind near its center

was about 52 m·s$^{-1}$, which made it the strongest typhoon landing on the mainland of China in 2019. According to the statistics of the Chinese Meteorological Administration, Lekima was detained on land for 44 hours; the affected area with rainfall measurements over 100 mm was about 361000 square kilometers during this period, and 19 national meteorological stations broke their historical daily rainfall recordings. During landfall, high waves were stirred up along the coastline, as depicted in Fig. 1a, and the landslide in Fig. 1b blocked the river and temporarily formed a dyke with a sudden rise of the water level of the river before the collapse of the landslide dyke, resulting in 22 casualties around this area. Waterlogging submerged the road network and many buildings in the urban area of WL, LH, YH, and XJ in TZ city (see Figs. 1c-1f). Millions of people evacuated from TZ city or were trapped in the disaster area. A total of 57 casualties were reported due to the landslides, floods, and waterlogging during the landfall of Lekima.

This paper investigates the microphysical characteristics of the typhoon-induced storm after its landfall, using observations from an S-band polarimetric radar deployed at Wenzhou (hereafter referred to as WZ-SPOL), six Thies disdrometers, and a local rain gauge network around the disaster area. So far, the reason for the significant convective asymmetries in the concentric eyewalls before its landfall has been ascribed to the phase locking of vortex Rossby waves (VRW), and the cloud and precipitation microphysics caused by this phase-locking VRW-triggered asymmetric convection have been respectively revealed (Dai et al. 2021; Huang et al. 2022), mainly based on the WZ-SPOL radar and another Doppler weather radar in Taizhou (TZ). The DSD differences in the eyewall and spiral rainbands based on surface disdrometer measurements have also been demonstrated (Bao et al., 2020). However, the microphysical processes inherent in Lekima after its landfall have not been thoroughly investigated.

The novel contributions of this paper are summarized as follows: 1) An enhanced QC procedure for disdrometer measurement is developed and analyzed through cross-comparison with rain gauge and WZ-SPOL radar measurements. 2) The microphysical process with overwhelming breakup over coalescence during the landfall of Lekima is revealed based on radar and surface disdrometers. 3) The impacts of dominant breakup/coalescence on radar QPE are investigated through an $R(Z_H, Z_{DR})$ estimator, and this algorithm integrates the expected $Z_{DR}$ (i.e., $\hat{Z}_{DR}$) estimated from attenuation-corrected $Z_H$ (i.e., $Z_H^C$) to mitigate the negative effects of the unique microphysical process, in which dominant breakup in the air transitioned to dominant coalescence near the surface around the GWS of YDM.

The remainder of this article is organized as follows: Section 2 introduces the study domain, hardware configuration, and data processing methodologies. Section 3 details the precipitation microphysics associated with Lekima (2019). The impacts of dominant collision-breakup or collision-coalescence on radar QPE performance are also quantified in Section 3. Section 4 summarizes the main findings of this study and suggests future directions in implementing this work in an operational environment.

## 2. Study Domain and Data Processing

### 2.1. Study Domain

As shown in Fig. 2a, this paper focused on the north side of WZ city and the south side of TZ city. These two cities are both regional central cities of Eastern China: WZ is an important trade city with more than 9 million residents and an urban popularity density of 2,900 $km^{-2}$. TZ is an important seaport city in southeastern China with 6 million residents and an urban popularity density of 688 $km^{-2}$. Historical typhoons have landed on the coastlines of these two cities, indicating the necessity and importance of monitoring typhoons coming into this area. With this aim, the S-band weather radar in WZ was upgraded to a polarimetric radar system in 2019 to enhance its precipitation-monitoring capability. WZ-SPOL radar is deployed on a hill (735 m) near the coastline, as depicted in Fig. 2a. It sufficiently covers the flood and waterlogging disaster area caused by the landfall of Lekima. Two mountains lay between WZ and TZ, Kuo Cang Mountain (KCM) and Yang Dang Mountain (YDM). Although the mountainous terrain causes no serious beam blockage issues, the vertical gap between the radar beam center and the surface enlarges with ascending volume gates, as depicted in Fig. 2c. In addition, KCM and YDM both feature a typical groove topography, as indicated by the dashed lines in Fig. 2a, which benefits the assembling and uplifting of water vapor on the lower atmospheric layers.

Six Thies laser-optical disdrometers have been deployed at the national meteorological stations around the target area since 2017 (see Figs. 2a and 1b). These include Xian Ju (XJ), Lin Hai (LH), Win Ling (WL), Hong Jia (HJ), Yu Huan (YH) and Dong Tou (DT), and they provide particle size and terminal velocity (size–velocity) pairs with a one-minute time resolution. These size–velocity pairs are utilized to calculate rainfall intensity and to simulate dual-polarization radar measurements near the surface.

In addition, 356 tipping-bucket rain gauge stations (see Fig. 2b) are uniformly deployed around ten towns that have suffered from landslide and waterlogging disasters within the coverage of a radius of 135 km from the WZ-SPOL radar. The time resolution of the gauge measurements is also configured as one minute; if hourly gauge measurements are interrupted temporarily due to network issues, such as transmission congestion, these interrupted recordings will not be utilized. If we suppose that a gauge rainfall recording exceeds 1 mm, but the ratio between hourly gauge rainfall and any hourly radar estimates exceeds 10 (or less than 0.1 for the intercomparison), then this gauge measurement is suspected to be falsely reported and will not be used. This ratio (10, suggested in Marzen., 2004) is large enough to eliminate significant outliers but keep most other valuable gauge rainfall recordings.

## 2.2. Radar Configuration and Data Processing

The WZ-SPOL radar adopts the simultaneous horizontal and vertical polarization mode. For the routine operations, the standard volume coverage pattern (VCP21) is configured, which has elevation angles including $0.5°$, $1.5°$, $2.4°$, $3.3°$, $4.3°$, $6.0°$, $9.9°$, $14.5°$ and $19.5°$. The azimuthal radial resolution is set as $0.95°$ and the range gate resolution is configured as 250m for all elevation angles. Radar-measured $Z_H$, $Z_{DR}$, $\rho_{HV}$, radial velocity ($V_R$), and $\Psi_{DP}$ are archived in the radar data acquisition (RDA) system and then transferred to the radar products generation system to produce some predefined standard radar products. QC

processing for WZ-SPOL radar data is performed using the following steps:

(i) Ground clutter (GC) identification and mitigation.

Two parts are included in this step. The clutter mitigation decision (CMD) algorithm (Hubbert et al. 2009) has been integrated into the RDA software to filter the ground clutters in real-time, but some residual static ground clutters (RSGC) still exist in

the WZ-SPOL radar measurements at $0.5°$ elevation angle. To further eliminate the RSGC, WZ-SPOL radar $Z_H$ measurements on $0.5°$ elevation angle from August 2019 are utilized. The max number ($N_{max}$) of the pixel with $Z_H$>0dBZ within 55 km from the WZ-SPOL radar is 6981, and the observation number ($N_{obs}$) of each pixel within this range is normalized by dividing $N_{max}$. Then, an RSGC statistical map is derived, as shown in Fig. 3a, representing the relative frequency (Freq. % of $Z_H$>0dBZ) within the coverage of the WZ-SPOL radar. In this map, the pixels with Freq>50% are deemed to be contaminated by the

RSGC, and they form an RSGC mask in Fig. 3b to eliminate RSGC-contaminated $Z_H$ and $Z_{DR}$ at the $0.5°$ elevation angle of the WZ-SPOL radar.

(ii) $\Psi_{DP}$ processing

A nine-gate smoothing is first carried out to suppress the noise signals along the $\Psi_{DP}$ range profile. Then, a procedure is

180 executed to correct the aliased $\Psi_{DP}$ based on the standard deviation of $\Psi_{DP}$ in nine consecutive range gates (Wang et al., 2009), and $360°$ is added to the aliased $\Psi_{DP}$ to guarantee a monotonically increasing $\Psi_{DP}$ range profile. In addition, the iterative filtering method in Hubbert and Bringi (1995) is used to filter the backscatter differential phase, and a zero-started filtered $\Phi_{DP}$ ($\Phi_{DP}^{filter}$) range profile is obtained by removing the initial phase of $\Phi_{DP}$. The $\Phi_{DP}^{filter}$ range profile is utilized to estimate $K_{DP}$ through a linear fitting approach (Wang and Chandrasekar, 2009) with an additional non-negative constraint on $K_{DP}$.

(iii) Attenuation correction for $Z_H$

The ZPHI approach proposed by Bringi et al. (2001) is extended for correcting S-band $Z_H$ measurements.

$$A_H(r) = \frac{[Z_H^M]^b[10^{0.1ba\Delta\Phi(r_0,r_m)}-1]}{I(r_0,r_m)+[10^{0.1ba\Delta\Phi(r_0,r_m)}-1]I(r,r_m)} \tag{1a}$$

$$\Delta\Phi_{DP}(r_0,r_m) = \Phi_{DP}(r_m) - \Phi_{DP}(r_0) \tag{1b}$$

$$I(r_0,r_m) = 0.46b \int_{r_0}^{r_m}[Z_H^M(r)]^b ds \tag{1c}$$

$$I(r, r_m) = 0.46b \int_r^{r_m} [Z_H^M(r)]^b ds \qquad (1d)$$

$$\Phi_{DP}^{rec}(r_0, r_m) = \int_{r_0}^{r_m} \frac{A_H(s,\alpha)}{\alpha} ds \qquad (1e)$$

$$C(r_0, r_m) = \int_{r_0}^{r_m} |\Phi_{DP}^{rec}(s, \alpha) - \Phi_{DP}| ds \qquad (1f)$$

$$Z_H^C(r) = Z_H^M(r) + 2 \int_0^r A_H(s, \alpha) ds \qquad (1g)$$

where $Z_H^M$ and $Z_H^C$ denote the measured and attenuation-corrected reflectivity, respectively; $\Phi_{DP}$ refers to the filtered differential phase; $\Phi_{DP}^{rec}$ is a reconstructed differential phase through the ZPHI processing chain with an optimal coefficient $\alpha$ iteratively searched in the range [0.01, 0.12] by step 0.01 until the cost function $C(r_0, r_m)$ of the difference between $\Phi_{DP}$ and $\Phi_{DP}^{rec}$ in Eq. (1f) is minimized. The coefficient $b$ is assumed to be 0.62 for the S-band (Ryzhkov et al.,2014).

The ZPHI approach utilizes $Z_H^M$ and $\Delta\Phi_{DP}$ in Eq. 1b to calculate attenuation $A_H$. Here, it should be noted that three constraints are imposed on the ZPHI processing chain to ensure its practical performance, including a non-negative constraint on $A_H$, $\rho_{HV}$ constraint on the range gates partitioning, and convergence constraint to avoid incorrect calculation termination (Gou et al., 2019a). Finally, $Z_H^M$ is corrected to $Z_H^C$ according to Eq. 1g.

(iv) $Z_{DR}$ processing

The $Z_{DR}$ offset is usually routinely obtained in zenith mode, with which near-zero $Z_{DR}$ is anticipated in light rain scenarios, and then this offset is fed back to the RDA system to ensure slight $Z_{DR}$ bias. Bringi et al. 2001 showed that all $Z_{DR}$ values at the far side of one radial profile are expected to approximate 0 dB if the "intrinsic" $Z_H$ is small enough (i.e., $Z_H^C<20$dBZ) and attenuation-corrected $Z_{DR}$ ($Z_{DR}^C$) should approximate to their $\hat{Z}_{DR}$ along the whole radial profile; thus, appropriate $Z_{DR}$ bias adjustment may effectively help in such a $Z_{DR}$ approximation. In this process, near-zero $\hat{Z}_{DR}$ is also anticipated for far-side volume gates containing ice crystals with $Z_H^C<20$dBZ. Here, the exponential $Z_{DR}$-$Z_H$ relationship is established as Eq. 2a based on the quality-controlled DSD datasets from all national meteorological stations (denoted as $S_0$) detailed in Section 2.3 and the analysis in Section 3.1. Therein, $Z_H$, $Z_{DR}$, and $K_{DP}$ are simulated using the T-matrix method, assuming the raindrop aspect ratio in Brandes et al. (2002) at a temperature of 20°C. Then, the differential attenuation factor ($A_{DP}$) in Eq. 2b is calculated by adjusting $\beta$ to obtain $Z_{DR}^C$ according to Eq. 2c. The optimal $\beta$ can be iteratively determined for $A_{DP}$ by minimizing the differences between $Z_{DR}^C$ and $\hat{Z}_{DR}$ in Eq. 2d, along the whole radial range profile. Additional $\Delta Z_{DR}$ is also iteratively imposed on the whole range profile with a step of 0.1 dB to mitigate the residual $Z_{DR}$ bias caused by miscalibration or wet radome effects. Then, $Z_{DR}^M$ is corrected by Eq. 2c to $Z_{DR}^C$ through $A_{DP}$ calculated by the optimal $\beta$. $Z_{DR}^C$ is utilized for the subsequent analysis and radar rainfall estimation.

$$\hat{Z}_{DR}(r) = 1.3038 \times 10^{-4} Z_H(r)^{2.4508} \qquad (2a)$$

$$A_{DP}(r; \beta) = \frac{\beta}{\alpha_{opt}} A_H(r; \alpha_{opt}) \qquad (2b)$$

$$Z_{DR}^C(r; \beta) = \Delta Z_{DR} + Z_{DR}^M(r) + 2 \int_0^r A_{DP}(s, \beta) ds \quad (2c)$$

$$C_{DR} = \int_0^r |Z_{DR}^C(r; \beta) - \hat{Z}_{DR}(r)| \, dr \quad (2d)$$

### 2.3. DSD Data Processing

The Thies disdrometer measurements configured with 1 min sampling intervals collected between 0000 UTC, 09 August 2019, and 0000 UTC, 11 August 2019, are utilized. These measurements were variously affected by the strong winds, with the hourly maximum wind speed exceeding 20 m·s$^{-1}$, as depicted in Fig. 4. Particularly, YH, WL, and DT suffered more seriously (>40 m·s$^{-1}$) after 1600 UTC, 09 August 2019. Theoretically, the size–velocity measurements of raindrops, which are recorded by disdrometers in pairs, should be uniformly distributed as in the drop velocity model in Beard (1977), which can be represented

as

$$V_B(D_i) = 9.65 - 10.3 \times e^{-0.6D_i} \quad (3)$$

where $D_i$ is the diameter of the $i_{th}$ size class (diameter interval) and $V_B$ is estimated by $D_i$. However, real velocity measurement ($V_M$) of disdrometers may deviate seriously from $V_B$ due to the strong wind effects. For instance, many size–velocity pairs at all six stations are biased with $V_M < 0.5V_B$ and distributed in all predefined size classes; more deviated size–velocity pairs of

WL, YH, and DT are featured with $V_M < 0.5V_B$ in Figs. 5d-5f than in XJ, LH, and HJ in Figs. 5a-5c, which can also be ascribed to high wind speeds. Consequently, these size–velocity pairs need to be preprocessed, and the QC procedure utilized hereafter includes the following three steps:

(i) For wind-contaminated size–velocity pairs, if the $V_M$ of the $i_{th}$ size class is located inside [$0.5V_B$, $1.5V_B$] (enclosed by the blue lines in Fig. 5), the size–velocity pairs are deemed to agree well with Eq. 3 and will be kept; the other outliers will

be eliminated.

(ii) For the potential hail ($D_i > 5$mm) and graupel ($D_i$ in [2mm, 5mm]), two size–velocity relationships listed in Friedrich et al. (2013) as

$$V_H(D_i) = 10.58 \times (0.1D_i)^{0.267} \quad (4a)$$

$$V_G(D_i) = 1.37 \times (0.1D_i)^{0.66} \quad (4b)$$

are selected to estimate the velocity of potential hail ($V_H$) and graupel ($V_G$) corresponding to $D_i$. The size–velocity pairs that fulfilled $|V_B - V_M| < |V_H - V_M|$ or $|V_B - V_M| < |V_G - V_M|$ will be kept because they are more prone to raindrops; otherwise, these measurements are eliminated from the original dataset depicted in Fig. 5.

(iii) The residual contaminations, which the abovementioned processing cannot directly eliminate due to their similar size–velocity characteristics to raindrops, need another analysis based on DSD-derived median volume diameter ($D_0$) and $Z_{DR}$.

Larger $Z_{DR}$ values are anticipated for melting solid particles than raindrops with similar diameters. The final QC processing result of the DSD dataset is presented in Section 3.1.

## 3. Analysis and Results

### 3.1 The Consistency between Multi-source data

#### 3.1.1 The Surface Consistency between Disdrometer and Rain Gauge

A DSD dataset is critical for establishing relationships between polarimetric radar variables for radar QPE algorithms. Disdrometers and rain gauges are usually deployed at the same meteorological site; although they sample the precipitation differently, their rainfall measurements in the same area should agree with each other. However, DSD-derived rainfall at six stations, directly calculated from the size–velocity pairs in Fig. 5 without any QC processing (denoted as $R_M$), all presented unrealistically large values: maximum $R_M$ at LH, XJ, and HJ exceeded 200 mm, that at DT exceeded 400 mm, and that at WL

and YH unbelievably exceeded $3 \times 10^3$ mm and $10^4$ mm during typhoon Lekima. For convenient comparison of disdrometers with gauge rainfall series, $R_M$ is rewritten as

$$R_{TM} = \begin{cases} R_T + (R_M - R_T)/C_T & R_M > R_T \\ R_M & R_M \leq R_T \end{cases} \quad (5)$$

where $R_{TM}$ stands for the transformed rainfall value and $R_T$ stands for a rainfall threshold that is set a little larger than the maximum hourly gauge rainfall. $C_T$ is also manually set for each station, and $C_T$ partly indicates that $R_M$ is at least $C_T$ times

higher than gauge rainfall. The $R_M$ part exceeding $R_T$ can shrink into a limited range interval, and $R_T$ and $C_T$ serve for comparing $R_M$ and DSD-derived rainfall after QC processing (denoted as $R_{QC}$) in the same figure, as depicted in Fig. 6. Accordingly, Fig. $C_T$ of YH and WL in Fig. 6 is huge (800 and 500, $\geq 20$ at the other stations). Meanwhile, DSD-derived maximum $Z_H$, $Z_{DR}$, $K_{DP}$, and $R$, respectively, exceeded 85 dBZ, 5.5 dB, 1500 deg·km$^{-1}$, and 15000 mm/h (see Figs. 7a-7c), and they are also abnormally larger than the final QC-processed counterparts (rectangles in Figs. 7a-7c). If these unrealistic DSD-derived radar

variables were directly utilized to establish the parameters of any radar QPE algorithm, an unrealistically overestimated radar rainfall field would be obtained. Afterward, the QC procedure in Section 2.3 is first imposed on the size–velocity pairs, and its performance and effectiveness are investigated through comparison with gauge rainfall recordings.

According to a visual comparison in Fig. 6, the severe overestimation of $R_M$ at all six stations is reduced after processing wind

effects, and a better approximation is noticeable at XJ, HJ, LH, and DT in Figs. 6a-6c and 6e, where the extra hail and graupel processing hardly change the residual differences. In contrast, the $R_{QC}$ time series at WL agrees well with its gauge rainfall recordings after the hail processing but is underestimated after extra graupel processing (see Fig. 6d). This implies that WL suffers some solid particle contaminations. Still, these solid particles may melt and have similar size–velocity characteristics to raindrops, and their removal is responsible for the final underestimation of $R_{QC}$ at WL after QC processing. YH also suffered

solid particle contaminations. During its peak rainfall recording period between 1600 UTC and 2200 UTC, 09 August 2019, $R_{QC}$ in Fig. 6e changes relatively less after the hail processing and still deviates largely from gauge rainfall recordings; conversely, $R_{QC}$ better approximates gauge rainfall after the graupel processing. This indicates that the terminal velocity of

these filtered particles is more prone to graupel (not deduced by size). Section 3.2.1 further verifies the falling solid particles.

These residual solid particles could result in a false relationship between $D_0$ and $Z_{DR}$. As shown in Fig. 8a, the fitted curve uniformly passed through the scattergram, representing an excellent fitting relationship between $D_0$ and $Z_{DR}$. However, as mentioned above, these DSD-derived $D_0$ and $Z_{DR}$ still suffer some solid particle contaminations after processing the wind effects. Even after hail and graupel processing, the scattergram in Fig. 8b still presents a significant overfitted relationship between $D_0$ and $Z_{DR}$. The scatters with $Z_{DR}>2.5$dB are related to melting solid particles with $D_0$ ranging from 1.5 mm to 4 mm,

and some have raindrop-like sizes (<2mm). Finally, DSD-derived radar variables constrained by $Z_{DR}<2.5$dB are assumed to be contributed by pure raindrops, and they are utilized to fit the $D_0$-$Z_{DR}$, LWC-$K_{DP}$, and $K_{DR}$-$Z_H$ relationships in Figs. 8c-Fig. 8e and the $Z_{DR}$-$Z_H$ relationship in Eq. 2a (see Fig. 8f).

$$D_0 = 0.2987 \times Z_{DR}{}^3 - 1.3229 \times Z_{DR}{}^2 + 2.1931 \times Z_{DR} + 0.3543 \qquad (6)$$

$$LWC = 2.0949 \times K_{DP}^{0.6889} \qquad (7)$$

$$K_{DP} = 1.5473 \times 10^{-13} \times Z_H^{8.8365} \qquad (8)$$

Combining these relationships and another relationship between the normalized concentration of raindrops ($N_w$), LWC, and the mean volume diameter of the drop size distribution ($D_m$) as:

$$N_w = \frac{4^4}{\pi \rho_w} \frac{LWC}{D_m^4} \qquad (9)$$

$$D_m = \frac{4+\mu}{3.67+\mu} D_0 \qquad (10)$$

where $\rho_w$ is the water density (1 g·cm$^{-3}$). Based on Eqs (6)-(10), high-resolution DSD parameter fields can be derived from WZ-SPOL radar measurements.

### 3.1.2 The Self-consistency between Radar Measurements

The self-consistency can demonstrate the credibility of polarimetric radar measurements through scattergrams (Fig. 9). The scattergrams in Figs. 9b and 9d are obtained from all $Z_H^C$, $Z_{DR}^C$ and $K_{DP}$ measurements described in Fig. 11. The ZHPI approach

(Bringi et al.2001) with more constraints described in Gou et al. (2019a) effectively mitigates the attenuation effects on $Z_H$ and $Z_{DR}$ of the WZ-SPOL radar. The spatial fields of $Z_H^M$ and $Z_{DR}^M$ are not presented (they will not be used for the subsequent analysis), but it is noticeable that radar-measured $Z_H^M$, $Z_{DR}^M$ and $K_{DP}$ are not self-consistent before attenuation correction processing: it is obvious for $Z_H^M>40$dBZ and $K_{DP}>1$deg·km$^{-1}$ that $K_{DP}$-$Z_H^M$ scatters deviates positively from the theoretical $K_{DP}$-$Z_H$ curve (Eq. 8 as depicted in Fig. 8e), indicating that larger reflectivity values are anticipated for these $K_{DP}$ measurements. In

addition, an overall deviation of $Z_{DR}^M$-$Z_H^M$ distribution in Fig. 9c from the theoretical $Z_{DR}$-$Z_H$ curve (the black curve stands for Eq. 2a as depicted in Fig. 8f) addresses a nonnegligible negative $Z_{DR}$ bias before the differential attenuation correction. In contrast, the scattergram core areas in Figs. 9b and 9d (defined as $\log_{10}(N)>1.6$) exhibit more compact distribution along

theoretical $K_{DP}$-$Z_H$ and $Z_{DR}$-$Z_H$ curves, demonstrating the effectiveness of the attenuation correction to enhance the self-consistency between $Z_H^C$, $Z_{DR}^C$ and $K_{DP}$.


Radar measurements are feedback from drops in the air, but disdrometers collect DSD near the surface. In this sense, the comparison above also means that radar measurements tend to be more consistent with their surface counterparts after the correction. However, this does not mean that they completely agree; conversely, $Z_{DR}^C$ still deviates largely from $\hat{Z}_{DR}$ when reflectivity exceeds 35 dBZ in Fig. 9d. In addition, the time series in Fig. 10 shows that extremely large DSD-derived $Z_H$, $Z_{DR}$,

and $K_{DP}$ in Fig. 7 (time series not presented) have diminished, and they begin to approximate their radar-measured counterparts. The hail/graupel processing effectively improves the consistency between the gauge and disdrometer, as mentioned above; furthermore, DSD-derived $Z_H$ and $K_{DP}$ also simultaneously tend to better approximate radar-measured $Z_H^C$ and $K_{DP}$. Meanwhile, the residual differences between radar and DSD are still prominent in terms of $Z_{DR}$, and larger DSD-derived $Z_{DR}$ than radar-measured $Z_{DR}^C$ occurs at WL and YH, indicating that larger-sized drops are collected by WL and YH than radar volume gates

above.

Considering that Eq. 2a is fitted based on $S_0$ and $R_{QC}$ at WL agrees better with gauge rainfall if no graupel processing, $S_0$ can be refined: $S_I$ excludes large-sized drops by removing WL; $S_{II}$ further excludes large-sized drops from WL and YH; $S_{III}$ re-includes more large-sized drops by adding the size–velocity pairs removed by graupel processing at WL. In this way, three

new $Z_{DR}$-$Z_H$ relationships are re-established as

$$\hat{Z}_{DR}(r) = 3.477 \times 10^{-4} Z_H(r)^{2.161} \qquad \text{DSD} \in S_I \qquad (11a)$$

$$\hat{Z}_{DR}(r) = 5.033 \times 10^{-4} Z_H(r)^{2.0383} \qquad \text{DSD} \in S_{II} \qquad (11b)$$

$$\hat{Z}_{DR}(r) = 1.0652 \times 10^{-4} Z_H(r)^{2.508} \qquad \text{DSD} \in S_{III} \qquad (11c)$$

The further removal of the DT dataset from $S_{II}$ will change the $Z_{DR}$-$Z_H$ relationship in Eq.11b very little (data not presented).

Although there is an uncertainty that large-sized drops may source either from melting solid particles or the collision-coalescence, more large-sized drops in $S_0$ and $S_{III}$ make Eqs. 2a and 11c (higher $\hat{Z}_{DR}$ estimates) prone to the outcome of the dominant collision-coalescence process; conversely, more small-sized drops in $S_{II}$ and $S_{III}$ make Eqs. 11a and 11b prone to dominant collision-breakup. Resultantly, Eqs. 11a and 11b exhibit smaller $Z_{DR}$ than that of Eqs. 2a and Eq. 11c for a given $Z_H^C$, which agrees well with the simulation result in Kumjian and Prat. 2014. In Fig. 9d, radar-measured $Z_{DR}^C$ tends to be more

consistent with Eqs. 11a and 11b for a given $Z_H^C$ than Eqs. 2a and 11c in the scattergram core area, and this $Z_{DR}^C$-$Z_H^C$ scattergram reflects that the governing collision-breakup processes in radar volume gates restrain the drop size increase due to the coalescence-breakup balance, which means $Z_{DR}^C$ does not grow similarly to coalescence-dominated volume gates.

## 3.2 Microphysics of the Landfall of Lekima (2019)

When super typhoon Lekima landed on the eastern coast of China, several beneficial conditions for its evolution were
perceived: (i) the severe interaction between the mountain and the typhoon caused terrain-enhanced precipitation; (ii) the wind
speed shear (the bold black curves in Figs. 11a-11d) with noticeable $V_R$ differences benefited the strengthening development
of convective storms; (iii) the typhoon carried abundant warm moisture which can condensate if confronted with cold air. The
characteristics of Lekima can be described based on $Z_H^C$: the outer and inner eyewalls were both featured with $Z_H^C$>55dBZ
before landfall in Fig. 11e, indicating the enhanced convective development of the concentric eyewalls before its landfall;
afterward, the inner eyewall was destroyed and merged with the outer eyewall into a convective storm with an enlarged area,
with $Z_H^C$>55dBZ dwelling around the GWS of YDM (in Fig. 11f), and then the storm area with $Z_H^C$>55dBZ transitioned to the
north GWS of YDM (in Fig. 11g) but strongly weakened when it passed over the mountain ridge between YDM and KCM (as
depicted in Fig. 11h). More complex microphysical processes than these described also occurred during the landfall of Lekima.

### 3.2.1 Polarimetric Signatures of Solid Particles

The time series of vertical polarimetric radar measurement (Figs. 12-17), which is constructed with an altitude resolution of
100 m based on the technique in Zhang et al., 2006, is chosen to describe the microphysical evolutions upon each station;
DSD-derived radar measurements in Section 3.1 assist in interpreting what occurred near the surface. The combination of radar
and DSD can effectively explain the potential microphysical processes in the vertical gap between the air and the surface.

The freezing level (FL) is significant in the vertical measurements (see Figs. 12, 14, and 17) and its altitude is about 7 km: the
layers with near-zero $Z_{DR}^C$ measurements dominate above the FL, indicating the dominant dry snow aggregates ($Z_H^C$<30dBZ);
$\rho_{HV}$ is relatively weaker (<0.98) below the FL, indicating the dominant mix-phase particles in the ML (near 6km). In addition,
the sustaining large $K_{DP}$ (>1deg·km$^{-1}$) upon WL and HJ (Figs. 12 and 14) after 1800 UTC, 09 August 2019 (after landfall)
indicates the high concentration of solid particles above the FL. In addition, the significant upward extension of $Z_H^C$ (>40dBZ)
and $Z_{DR}^C$(>1dB) columns marked with black rectangles indicate the developing convective storms; the black ellipses indicate
potential updrafts coupled with the storm; the blue ellipses indicate subsiding signatures of falling solid particles deducing
from gradual decreasing heights of $\rho_{HV}$<0.98 over time. The convective storms are accompanied by abundant water content,
as indicated by significant $K_{DP}$ (>0.5deg·km$^{-1}$) columns extending upwards, which benefited the size increases of the falling
solid particles. The microphysical processes of the solid particles differed at each station.

WZ-SPOL radar initially measured similar $Z_{DR}$ but larger $Z_H$ and $K_{DP}$ compared with DSD at the WL station (rectangle 1 in
Fig. 10) before the landfall of Lekima, and more concentrated hydrometeors aloft accompanying the updrafts compared to the
surface in this process account for this phenomenon, which is verified in the first rectangle of Figs. 12a and 12c. Furthermore,

two consecutive severe updrafts passed over WL, one from the outer eyewall and the other from the inner eyewall, causing the significant upward extension of $Z_H^C$, $Z_{DR}^C$ and $K_{DP}$ columns below the FL, as depicted in two black rectangles in Fig. 12. As illustrated in two ellipses, some ice particles might ascend with the first updraft, then fall and melt in the time gap between two updrafts, with the signature of $\rho_{HV}$ <0.98 reaching the lowest layer of 1.8 km; they instantly suffered from another size increase process confronting the second updraft (in the second ellipse) and then fell with the subsiding signals of $\rho_{HV}$ and $Z_{DR}^C$ (in the blue ellipse): $Z_{DR}^C$<0.5dB was sustained when $\rho_{HV}$ gradually transitioned from $\rho_{HV}$<0.84 around the FL to $\rho_{HV}$<0.98 on the lowest layer, indicating the existence of some near-spherical but mixed-phase particles during this falling process. These solid particles partly account for the larger DSD-derived $Z_{DR}$ near the surface than WZ-SPOL radar (rectangle 2 in Fig. 10), but the coalescence of raindrops might also partly account for this DSD-derived larger $Z_{DR}$.

Similar updrafts occurred upon the YH station (rectangle 3 in Fig. 10), and the WZ-SPOL radar measured similar $K_{DP}$ but weaker $Z_H$ and $Z_{DR}$ compared with DSD before the hail/graupel processing. Featuring with similar $Z_H^C$ extending upwards upon the YH station, large $Z_{DR}^C$(>1.2dB) and weak $K_{DP}$(<0.2 deg·km$^{-1}$) accompanied the updrafts with $\rho_{HV}$<0.98 in the black ellipse of Fig. 13, indicating that dominant large-sized mixed-phase particles were developing around the ML. Then, in the blue ellipse, the subsiding signals of $\rho_{HV}$<0.84 formed in Fig. 13d after 1630 UTC and tended to decrease their heights over time; finally, they transitioned to $\rho_{HV}$>0.98 on the top of the $Z_{DR}^C$(>2dB) columns, attributing to the transformation of melting solid particles into big raindrops. Compared with surface DSD, the decrease in radar-measured $Z_H^C$ and $K_{DP}$ reflects the reduction of LWC in the vertical gap; this LWC reduction did not contribute to the size increase of drops because radar and DSD presented similar $Z_{DR}$. Another possible explanation is that some LWC is absorbed by the falling solid particles, contributing to the filtered $Z_{DR}$ part in the hail/graupel processing.

Another solid particle falling occurred upon HJ, which is to the north of the landfall positions of Lekima. Even with the unnoticeable upward $Z_H^C$ enhancement between 1700 UTC and 1800 UTC, 09 August 2019, as depicted in Fig. 14a, the large $Z_{DR}^C$ signals of the ML in Fig. 14b diminished due to the updraft; $Z_{DR}^C$ and $K_{DP}$ both increased, and $\rho_{HV}$ reduced steadily after 1800 UTC above the FL in the black ellipses of Figs. 13b-13d. Subsiding signals of $\rho_{HV}$<0.84 also emerged after 1800 UTC, resulting in the $\rho_{HV}$ reduction from 0.98 to 0.96 on the lowest layer. Conditioning $V_M$ by [0.5$V_B$, 1.5$V_B$] eliminated some size–velocity pairs of the solid particles at HJ because the lower the density of particles, the slower the terminal velocity of these particles. Conversely, the rising overestimation of $R_{QC}$ by reconditioning $V_M$ by [0.4$V_B$, 1.5$V_B$] in Fig. 6b (the dotted green line) further verified this possibility.

These common characteristics feature in HJ, DT, LH, and XJ in Figs. 14-17: $K_{DP}$ (<0.5 deg·km$^{-1}$) above the FL indicated a lower concentration of ice particles upon DT, LH, and XJ than upon the other three sites, which refrains the size increase of falling solid particles through the aggregation process; the insignificant $Z_H^C$ (<45dBZ) and $K_{DP}$ (<0.2 deg·km$^{-1}$) extending

upwards reflect the relatively low concentration of hydrometeors below the ML upon HJ, LH, and XJ, which refrains the further size increases of melting ice particles in the warm rain environment. The exceptions in $Z_{DR}^C$ columns upon DT between 1800UTC and 1900 UTC in Fig. 15b were attributed to the falling melting ice particles upon an updraft with high LWC ($K_{DP}$>1deg in Fig. 15c); those in LH between 1800UTC and 1900 UTC were attributed to the sustaining weak updrafts ($Z_H^C$ <45dBZ) but more concentrated ice particles above the FL. The deep ML ($\rho_{HV}$<0.98) also features these stations, and this signature even extends down to the lowest layer of LH and HJ with $Z_{DR}^C$>2dB dwelling below the FL. In addition, most ice particles upon these four stations might have melted in the air before being collected by disdrometers near the surface, which effectively accounts for the small rainfall differences between disdrometers and rain gauges. However, the residual differences between radar and DSD are mainly related to the warm process of raindrops below the ML.

### 3.2.2 Polarimetric Signatures of Raindrops

The deviation of $Z_{DR}^C$ from $\hat{Z}_{DR}$ is a nonnegligible phenomenon during landfall of Lekima: underestimated $Z_{DR}^C$ in Figs. 11i-11l compared with $\hat{Z}_{DR}$ in Figs. 11m-11p emerged in areas with significant $K_{DP}$ in Figs. 11q-11t, which simultaneously emerged around the GWS of YDM. Apparently, $Z_{DR}^C$ cannot completely approximate $\hat{Z}_{DR}$ after correction; intrinsically, the microphysical composition issue, either dominant large-sized or small-sized raindrops filling in radar volume gates, resultantly determines final $Z_{DR}^C$-$Z_H^C$ distribution. One typical radial range profile in Fig. 18 is detailed to clarify this phenomenon. The ellipse-surrounded storm area contributes the most attenuation and differential attenuation with maximum $\Delta Z_H$=7.9dBZ and $\Delta Z_{DR}$=0.645dB, respectively, in Figs. 18a and 18b. Although the correction can result in enhanced consistency between $Z_H$ and $K_{DP}$ (see Section 3.1.2) and some $Z_{DR}^C$ have indeed partly approximated well to $\hat{Z}_{DR}$ (outside the ellipse in Fig. 18b), the other $Z_{DR}^C$ within the storm (in the ellipse) still have a residual $Z_{DR}$ bias of about -1dB. Additionally, $\rho_{HV}$ ranging from 0.99 to 1 (in Fig. 15c) further indicates the dominance of pure liquid precipitation; high LWC and $N_w$ can be deduced from Eqs. 8 and 9 ($K_{DP}$≈3.5deg·km$^{-1}$; $\Delta\Phi_{DP}$≈68.5 deg in Fig. 12c). $Z_H$ is a composite integral of hydrometeors with different sizes and number concentrations, and $Z_{DR}$ is sensitive to hydrometeor size; therefore, high concentrations of small-sized drops rather than large-sized drops contribute more to radar-measured $Z_H^C$ in radar volume gates. This unique composition resultantly causes an overestimated $\hat{Z}_{DR}$ estimated by $Z_H^C$, or conversely, underestimated $Z_{DR}^C$ compared with $\hat{Z}_{DR}$.

The hydrometeor size sorting (HSS) partly accounts for the position inconsistency between $Z_H^C$ and $Z_{DR}^C$, and it is significant in the rectangle-surrounded area of the inner eyewall, characterized by a maximum of $Z_{DR}^C$ in Fig. 12i on the significant upwind gradients of $K_{DP}$ in Fig. 12q (Homeyer et al., 2021; Hu et al., 2020). Since $Z_H^C$ in Fig. 12e and $K_{DP}$ in Fig. 12q are consistent with each other, the large $\hat{Z}_{DR}$ estimated by $Z_H^C$ also horizontally deviates from the area with large $Z_{DR}^C$. Differential sedimentation of hydrometeors of various sizes is the intrinsic reason for HSS (Feng and Bell., 2019), which is significant in the outer eyewall. The higher LWC (>3 g·m$^{-3}$) features the outer eyewall as depicted in Fig. 16e; the area with large $Z_{DR}^C$ (>2dB)

consists of dominant larger-sized drops with $D_0 > 1.8$ mm in Fig. 16a, but relatively lower concentration with $\log_{10}(N_w) < 4.4$ m$^{-3}$·mm$^{-1}$ in Fig. 16a than in its downwind area. Meanwhile, lower LWC ($<2$ g·m$^{-3}$) features a cyclonical downwind area, but this

area consists of dominant higher concentrated ($\log_{10}(N_w) > 4.4$ m$^{-3}$·mm$^{-1}$) small-sized drops ($D_0 < 1.625$ mm). However, HSS cannot account for the overall underestimation of $Z_{DR}^C$ when pixels with large $Z_H^C$, $Z_{DR}^C$ and $K_{DP}$ coincide.

The collision process in warm rain has three probable colliding outcomes: bounce, coalescence, and breakup. In one volume gate, bounce cannot change raindrop size and concentration; coalescence boosts the size increase, but breakup increases the

concentration. The existence of large raindrops with $D_0 > 1.8$ mm around the GWS of YDM (in Figs. 19b and 19c) indeed back the occurrences of collision-coalescence processes, which corresponds to $Z_{DR}^C$ ($>1.8$ dB in Figs. 11j and 11k). However, if the size increases contribute enough in one volume gate, $Z_{DR}^C$ might have well approximated $\hat{Z}_{DR}$ in the storm area and agreed better with $Z_H^C$. In addition, raindrops cannot continue increasing their size; spontaneous breakup (Srivastava.,1971) or collision-breakup due to vertical wind shear (i.e., Deng et al., 2019) co-occurs during the falling process of drops:

(i) The first evidence comes from radar-measured $Z_{DR}^C$-$Z_H^C$ scattergram in Fig. 11d, and it tends to be more consistent with $Z_{DR}$-$Z_H$ relationships dominated by small-sized drops related to the breakup, not large-sized drops related to the coalescence. This also agrees with the simulation results in Kumjian and Prat.2014.

        (ii) The second natural phenomenon is the decreasing $Z_{DR}$ downward in the lower atmospheric layers. Although some $Z_{DR}^C$ columns were indeed enhancing downward in Figs. 12-17, particularly in the time frames with significant updrafts with

$Z_H^C$ extending upwards upon WL and YH, more time frames presented a dominant decreasing $Z_{DR}^C$ toward the ground, such as at DT, HJ, LH, and XJ.

        (iii) The residual differences between radar and DSD are evident for the possible process in the vertical gap between radar volume gates and the surface. If dominant collision-coalescence occurred, DSD-derived $Z_{DR}$ should be more significant than their radar counterparts in the air. However, the opposite is true at XJ, HJ, and LH, as depicted in Fig. 10. Meanwhile,

DT exhibits similar $Z_{DR}$, $Z_{H,}$ and $K_{DP}$ to its radar counterparts after the landfall of Lekima, which is also evidence against the contribution from coalescence.

The collision-coalescence indeed occurs, but the breakup balances the size increase. This is evident in the evolutions of $D_0$ and $N_w$ constrained by a given LWC, which is typical around the GWS of YDM. In Figs. 19c, 19g, and 19k, the identical LWC fill

in the rectangle-surrounded and ellipse-surrounded regions; the latter exhibits larger $D_0$ ($>1.75$ mm) but lower $N_w$ with $\log_{10}(N_w) < 4.4$ m$^{-3}$·mm$^{-1}$; conversely, the former shows smaller $D_0$ ($<1.75$ mm) but higher $N_w$ with $\log_{10}(N_w) > 4.4$ m$^{-3}$·mm$^{-1}$. Similar situations occurred in the two left columns in Fig. 19, and sparse large-sized $D_0$ is only prominent in a small area (in ellipse and rectangle); high $N_w$ but small $D_0$ are features of the other parts of the typhoon. The LWC in one range gate will contribute not only to the size increase but also to the concentration, attributing to the balance between coalescence and breakup.

Combining the abovementioned observations, the overwhelming breakup of large-sized drops over coalescence firmly restrains the magnitudes of radar-measured $Z_{DR}^C$ for a given $Z_H^C$, accounting for the noticeable deviation of $Z_{DR}^C$ from $\hat{Z}_{DR}$ (in Fig. 11). Despite all this, collision-coalescence accompanied by the terrain-enhanced precipitation occurred when Lekima took high LWC ($>$2g·m$^{-3}$) and passed over YDM, as depicted in Figs. 19f and 19g, resulting in an overall LWC reduction around the GWS of KCM (i.e., Fig. 19g to Fig. 19.h). During this period, $D_0$ and $N_w$ simultaneously increased: $D_0$ increased by about 0.5 mm from Fig. 19a to $D_0>$1.5 mm in Fig. 19c; $\log_{10}(N_w)$ increased about 0.4~0.8 m$^{-3}$·mm$^{-1}$ from Fig. 19i to 19k, and these enhancements coincided well with the GWS of YDM. The gradual but insignificant enhancement persisted around the GWS of KCM, including an LWC increase by about 1g·m$^{-3}$ (i.e., Figs. 19e-19h), a diameter transition from $D_0<$1.25 mm to $D_0>$1.5 mm (i.e., Figs. 19a-19d), and growth of $\log_{10}(N_w)$ about 0.4 m$^{-3}$·mm$^{-1}$ in sparse pixels (i.e., Figs. 19i-19l), but this enhancement was relatively weaker than that around the GWS of YDM. This comparison indicates that extensive large-sized drops had formed and fallen around the GWS of YDM before Lekima moved to the north, which effectively accounts for the flood disasters. However, the utilization of radar-measured $Z_{DR}^C$ may not derive accurate radar rainfall fields.

## 3.3 Radar QPE Analysis

### 3.3.1 The Performances of Radar QPE

Utilizing the DSD dataset from $S_0$, three primary radar rainfall rate relationships for $R(Z_H)$, $R(K_{DP})$, and $R(Z_H, Z_{DR})$ are respectively established as

$$R(Z_H) = 0.0544 \times Z_H^{0.608} \tag{12a}$$

$$R(K_{DP}) = 45.0484 \times K_{DP}^{0.7679} \tag{12b}$$

$$R(Z_H, Z_{DR}) = 0.0086 \times Z_H^{0.9153} Z_{DR}^{-3.8606} \tag{12c}$$

based on the standard weighted least squares nonlinear fitting method and DSD-derived radar variables (depicted in Fig. 20). In addition, $Z_{DR}^M$, $Z_{DR}^C$ and $\hat{Z}_{DR}$ are respectively integrated with $Z_H$ to exploit the impacts of the abovementioned microphysical process on radar QPE algorithms. The pixel-to-pixel linear average accumulation scheme is utilized to retrieve radar six-hour rainfall fields for these radar QPE estimators and then is evaluated independently by comparing gauge six-hour rainfall measurements through the absolute normalized mean error ($E_{NMA}$), root-mean-square error ($E_{RMS}$), and correlation coefficient ($E_{CC}$) as

$$E_{NMA} = \frac{\sum_{i=1}^n |r_i - g_i|}{\sum_{i=1}^n g_i} \times 100\% \tag{13a}$$

$$E_{RMS} = \sqrt{\frac{1}{n} \sum_{i=1}^n (r_i - g_i)^2} \tag{13b}$$

$$E_{CC} = \frac{\sum_{i=1}^{n}(r_i - \bar{r})(g_i - \bar{g})}{\sqrt{\sum_{i=1}^{n}(r_i - \bar{r})^2}\sqrt{\sum_{i=1}^{n}(g_i - \bar{g})^2}} \qquad (13c)$$

where $r_i$ and $g_i$ refer to radar rainfall estimates and gauge rainfall. Six-hour radar rainfall fields retrieved by $R(Z_H^M)$, $R(Z_H^C)$, $R(K_{DP})$, $R(Z_H^M, Z_{DR}^M)$, $R(Z_H^C, Z_{DR}^C)$, and $R(Z_H^C, \hat{Z}_{DR})$ are derived in Fig. 21, as well as the scattergram between radar rainfall estimates and gauge rainfall measurements depicted in Fig. 22, to reveal their practical performances around the disaster area.

500

$R(Z_H^M)$ presents lower rainfall estimates in Fig. 21a than the other radar rainfall estimators in Figs. 21b-21f, although they have similar rainfall center shapes. In terms of statistical scores in Table 1, $R(Z_H^M)$ performs not worst among all radar rainfall estimators. Its $E_{RMS}$, $E_{NMA}$, and $E_{CC}$ even outperform $R(Z_H^M, Z_{DR}^M)$ by 57%, 31.6% and 7.9%, and outperform $R(Z_H^C, Z_{DR}^C)$ by 63.8%, 34.9% and 6%, respectively. However, its underestimation can be easily perceived from the scatters in Fig. 22a when rainfall recordings exceed 100 mm in the center rainfall area. This phenomenon can be ascribed to the attenuation on $Z_H^M$ caused by the highly concentrated hydrometeors in the storm during the landfall of Lekima.

In contrast, $R(Z_H^C)$ in Fig. 21b presents higher rainfall estimates and $R(Z_H^C)$ mainly overestimates since more scatters are distributed above the diagonal line ($y = x$) as depicted in Fig. 22b, and its $E_{CC}$ outperforms that of $R(Z_H^M)$ by 4.2%, even with larger $E_{RMS}$ and $E_{NMA}$ scores. The overestimation of $R(Z_H^C)$ in the rainfall center area conversely demonstrates the effectiveness of the attenuation correction based on the ZPHI approach because the same $R(Z_H)$ relationship is utilized for the rainfall retrieval; the only difference is the replacement of $Z_H^M$ with $Z_H^C$.

$R(K_{DP})$ in Fig. 21c presents a similar rainfall field structure to $R(Z_H^C)$. The scores of $R(K_{DP})$ are just a little superior to that of $R(Z_H^C)$ in Table 1, with its $E_{RMS}$, $E_{NMA}$, and $E_{CC}$ outperforming that of $R(Z_H^C)$ by 3.1%, 3.2% and 0.5%, respectively. The scattergrams in Figs. 22b and 22c are also similar to each other, indicating that $R(K_{DP})$ and $R(Z_H^C)$ both overestimate, although $R(K_{DP})$ is less overestimated when rainfall recordings are less than 100mm. Their similar performances can be attributed to the consistency between radar-measured $K_{DP}$ and $Z_H^C$ measurements as described in Section 3.1.2.

$R(Z_H^M, Z_{DR}^M)$ and $R(Z_H^C, Z_{DR}^C)$ in Figs. 21d and 21e both present significantly higher estimates in the rainfall center area than the others, which results in severe overestimation according to the scattergrams in Figs. 22d and 22e. Furthermore, $R(Z_H^C, Z_{DR}^C)$ obtains the worst $E_{RMS}$ and $E_{NMA}$ scores of all radar rainfall estimators, and this can be explained based on the $Z_H$-related and $Z_{DR}$-related calculation items as demonstrated in Fig. 23: $Z_H^C$ obtains much higher rainfall estimates through $Z_H$-related items than $Z_H^M$. However, the calculation needs to be further adjusted through the $Z_{DR}$-related item: the larger $Z_{DR}$ measurements correspond to fewer final rainfall estimates. A -0.5dB $Z_{DR}$ bias could result in relatively less rainfall adjustment, according to Fig. 23. The attenuation effects on $Z_{DR}^M$ make the corresponding rainfall calculation less adjusted, which can effectively account

for the overestimation of $R(Z_{\mathrm{H}}^{\mathrm{M}}, Z_{\mathrm{DR}}^{\mathrm{M}})$. However, the correction cannot make $Z_{\mathrm{DR}}^{\mathrm{C}}$ completely consistent with $Z_{\mathrm{H}}^{\mathrm{C}}$, but it is underestimated, as demonstrated in Section 3.1.2, which is related to the dynamic microphysical process described in Section 3.2.


The spatial texture of $R(Z_{\mathrm{H}}^{\mathrm{C}}, \hat{Z}_{\mathrm{DR}})$ in Fig. 21f presents slightly fewer rainfall estimates than $R(Z_{\mathrm{H}}^{\mathrm{C}})$ and $R(K_{\mathrm{DP}})$ in Figs. 21b and 21c, and the scattergram in Fig. 22f shows that $R(Z_{\mathrm{H}}^{\mathrm{C}}, \hat{Z}_{\mathrm{DR}})$ agrees better with the gauge rainfall than in Figs. 22b and 22c. $R(Z_{\mathrm{H}}^{\mathrm{C}}, \hat{Z}_{\mathrm{DR}})$ effectively reduces the overestimates and is obviously superior to $R(Z_{\mathrm{H}}^{\mathrm{M}}, Z_{\mathrm{DR}}^{\mathrm{M}})$ and $R(Z_{\mathrm{H}}^{\mathrm{C}}, Z_{\mathrm{DR}}^{\mathrm{C}})$. The $E_{\mathrm{RMS}}/E_{\mathrm{NMA}}$ score of $R(Z_{\mathrm{H}}^{\mathrm{C}}, \hat{Z}_{\mathrm{DR}})$ performs even better than $R(Z_{\mathrm{H}}^{\mathrm{C}})$ and $R(K_{\mathrm{DP}})$, respectively, by 8.6/5% and 5.7/1.8%, although its $E_{\mathrm{CC}}$ score

is slightly worse by 0.2% and 0.3%. The superiority of $R(Z_{\mathrm{H}}^{\mathrm{C}}, \hat{Z}_{\mathrm{DR}})$ can also be apparently attributed to the incorporation of $\hat{Z}_{\mathrm{DR}}$. $\hat{Z}_{\mathrm{DR}}$ is not a real radar measurement; it is directly estimated from $Z_{\mathrm{H}}^{\mathrm{C}}$ from the theoretical DSD-derived $Z_{\mathrm{DR}}$-$Z_{\mathrm{H}}$ relationship in Eq. 2a. $\hat{Z}_{\mathrm{DR}}$ is naturally self-consistent with $Z_{\mathrm{H}}^{\mathrm{C}}$ and $K_{\mathrm{DP}}$, since $Z_{\mathrm{H}}^{\mathrm{C}}$ and $K_{\mathrm{DP}}$ have agreed well with their DSD-derived counterparts regarding the $K_{\mathrm{DP}}$-$Z_{\mathrm{H}}$ distributions and pixel-to-pixel comparisons in Section 3.1. The utilization of the DSD-derived $Z_{\mathrm{DR}}$-$Z_{\mathrm{H}}$ relationship intrinsically assumes that composition in radar volume gates has a similar size and

concentration to its surface counterparts; therefore, $\hat{Z}_{\mathrm{DR}}$ can be seen as an equivalent radar variable. The replacement of $Z_{\mathrm{DR}}^{\mathrm{C}}$ with $\hat{Z}_{\mathrm{DR}}$ is also equivalent to imposing the surface raindrop size and concentration on radar measurements. The relatively larger $\hat{Z}_{\mathrm{DR}}$ than $Z_{\mathrm{DR}}^{\mathrm{C}}$ means a more significant adjustment can be performed for rainfall estimation using $R(Z_{\mathrm{H}}^{\mathrm{C}}, \hat{Z}_{\mathrm{DR}})$, according to Fig. 23, and this also indicates that the anticipated giant raindrops had fallen around the GWS of YDM. Except for the simultaneous $D_0$ and $N_{\mathrm{w}}$ increases, the following alternative $\hat{Z}_{\mathrm{DR}}$ indirectly verifies the dominant collision-coalescence around

this area.

**3.3.2 The Impacts of Microphysical Processes on Radar QPE**

The consistency between radar and surface measurements is critical for radar QPE algorithms, but the microphysical process in the vertical gap between air and surface may worsen the practical performances of radar QPE. This is the case around the

GWS of YDM: the primary outcome of the collision process transitions from a dominant breakup in the air to dominant coalescence near the surface due to the topographical enhancement. Using radar measurements on the lowest elevation angle to retrieve radar QPE implicitly assumes that they are representative of surface precipitation, but they are not in this situation; only $R(Z_{\mathrm{H}})$, $R(K_{\mathrm{DP}})$, and $R(Z_{\mathrm{H}}, Z_{\mathrm{DR}})$ relationships established based on the DSD dataset represent the feedback near the surface. Although $R(Z_{\mathrm{H}}^{\mathrm{C}}, \hat{Z}_{\mathrm{DR}})$ performs best, $\hat{Z}_{\mathrm{DR}}$ can also be estimated by Eqs.11a-11c. However, $Z_{\mathrm{DR}}^{\mathrm{C}}$ changes little if a smaller/larger

$\hat{Z}_{\mathrm{DR}}$ estimated by Eqs.11a-11c is imposed in the correction procedure, as $Z_{\mathrm{DR}}^{\mathrm{C}}$-$Z_{\mathrm{H}}^{\mathrm{C}}$ scattergrams shown in Figs. 24a-24c. Furthermore, the corresponding three $R(Z_{\mathrm{H}}, Z_{\mathrm{DR}})$ relationships based on $S_{\mathrm{I}}$-$S_{\mathrm{III}}$ can be established as

$$R(Z_{\mathrm{H}}, Z_{\mathrm{DR}}) = 0.0088 \times Z_{\mathrm{H}}^{0.917} Z_{\mathrm{DR}}^{-3.9203} \qquad \mathrm{DSD} \in S_{\mathrm{I}} \qquad (14a)$$

$$R(Z_\mathrm{H}, Z_\mathrm{DR}) = 0.0085 \times Z_\mathrm{H}^{0.9222} Z_\mathrm{DR}^{-4.0371} \qquad \mathrm{DSD} \in S_\mathrm{II} \qquad\qquad (14\mathrm{b})$$

$$R(Z_\mathrm{H}, Z_\mathrm{DR}) = 0.0078 \times Z_\mathrm{H}^{0.9342} Z_\mathrm{DR}^{-4.2321} \qquad \mathrm{DSD} \in S_\mathrm{III} \qquad\qquad (14\mathrm{c})$$

The alternative utilization of $S_\mathrm{I}$-$S_\mathrm{III}$ slightly changes the parameters of $R(Z_\mathrm{H}, Z_\mathrm{DR})$ with minor rainfall rate differences estimated by Eqs. 12c, 14a-14c. However, the impacts on $\hat{Z}_\mathrm{DR}$ are nonnegligible, particularly for a given $Z_\mathrm{H}^\mathrm{C}$ exceeding 35 dBZ, and smaller $\hat{Z}_\mathrm{DR}$ means weaker adjustment for the $Z_\mathrm{HL}$-related item, as depicted in Fig. 23.

As in the analysis in Section 3.1.2, radar-measured $Z_\mathrm{DR}^\mathrm{C}$-$Z_\mathrm{H}^\mathrm{C}$ in volume gates tend to be more consistent with $S_\mathrm{I}$ and $S_\mathrm{II}$ because
breakup overwhelms in the coalescence-breakup balance, so if breakup still dominates when these drops further fall on the ground, $R(Z_\mathrm{H}^\mathrm{C}, \hat{Z}_\mathrm{DR})$ estimated by Eqs. 14a and 14b should perform better than that estimated by Eq. 12c. However,` in reality, their spatial fields in Fig. 25a and 25b and scattergrams in Figs. 26a and 26b conversely present a similar overestimation as $R(Z_\mathrm{H}^\mathrm{M}, Z_\mathrm{DR}^\mathrm{M})$ and $R(Z_\mathrm{H}^\mathrm{C}, Z_\mathrm{DR}^\mathrm{C})$, which contradicts the anticipated results. Such a contradiction means $\hat{Z}_\mathrm{DR}$ estimated by Eqs. 11a and 11b is not representative enough for surface precipitation. In contrast, $R(Z_\mathrm{H}^\mathrm{C}, \hat{Z}_\mathrm{DR})$ in Fig. 25c shows even lower rainfall
estimates than that in Fig. 21f (obtained through Eq. 12c based on $S_0$), which can also be seen by comparing the scattergrams in Fig. 26c and Fig. 22f. In addition, when large-sized drops are gradually excluded from the DSD dataset for $\hat{Z}_\mathrm{DR}$ and $R(Z_\mathrm{H}^\mathrm{C}, \hat{Z}_\mathrm{DR})$, $E_\mathrm{CC}$ changes little in Table 2, whereas $E_\mathrm{RMS}$ and $E_\mathrm{NMA}$ both exhibit a monotonic increasing tendency, implying the nonnegligible contribution of large-sized drops around the GWS of YDM.

The dominant breakup in the air but dominant coalescence around the GWS of YDM can be ascribed to the overshooting of radar beams and the topographical enhancement. In this sense, the utilization of $\hat{Z}_\mathrm{DR}$ instead of $Z_\mathrm{DR}^\mathrm{C}$ equals a physical conversion of breakup-dominated outcome in one volume gate for a given $Z_\mathrm{H}^\mathrm{C}$ into their coalescence-dominated counterparts in an average sense. In this conversion process, consistency between radar-measured $Z_\mathrm{H}^\mathrm{C}$ and $K_\mathrm{DP}$ in the air and the surface counterparts (DSD-derived $K_\mathrm{DP}$ and $Z_\mathrm{H}$) has been achieved, as indicated in Section 3.1.2, demonstrating the mass conservation
characteristics of falling drops. Therefore, radar-measured $Z_\mathrm{H}^\mathrm{C}$ and $K_\mathrm{DP}$ around the GWS of YDM may change insignificantly, which makes it conducive for $R(Z_\mathrm{H}^\mathrm{C}, \hat{Z}_\mathrm{DR})$ to obtain a better radar rainfall field.

### 3.4. Discussion

The microphysical processes during the landfall of typhoon Lekima have been revealed based on the analysis of consistency between measurements from radar, disdrometers, and rain gauge networks. The cause of the flood disaster around the GWS
of YDM and its impacts on the practical performance of radar QPE algorithms have been investigated. Several critical issues should be considered for radar quantitative applications in future:

   (i) High-quality DSD datasets could lay a solid foundation for microphysical analysis and polarimetric radar applications, but selecting representative datasets for different microphysical processes is critical to determine parameters for

quantitative applications, such as the construction of relationships between $Z_H$, $Z_{DR}$, $K_{DP}$, and $R$. The size–velocity QC procedure could be deeply refined for radar QPE in cold seasons. So far, one-dimensional disdrometers (OTT or Thies) are the main facilities to collect DSD measurements in the national meteorological stations over China. However, both GWS areas in this article have no DSD measurements for directly revealing and validating the critical precipitation process in the typhoon center area. Furthermore, there are more similar GWS areas in south China, thus deploying some two-dimensional disdrometers in these vital target locations could be beneficial for future research.

(ii) The polarimetric radar measurements are indispensable for microphysical analysis and quantitative applications. In particular, $Z_{DR}$ provides critical signatures for analyzing the collision process in this super typhoon event. Currely, more X-band polarimetric radar systems have been planned and/or deployed to fill the gap of operational S-band radar network. Although $Z_{DR}$ of S-, C-, and X-band radar is sensitive to drop size in different degrees, $Z_{DR}$ biases in X-band radar measurements can be more serious in a super typhoon case due to radome attenuation. The correction methods of $Z_H$ and $Z_{DR}$ in this article could potentially be further refined for X-band applications.

(iii) The spatial variability of precipitation could be far more complex, and it is oversimplified to assert that convective or stratiform rainfall always exhibits breakup or coalescence (Kumjian and Prat. 2014). It is noticed that the practical performances of $R(Z_H^C, \hat{Z}_{DR})$ relies on determining optimal $\hat{Z}_{DR}$ based on the representative DSD dataset of the microphysical process, which is the main limitation of $R(Z_H^C, \hat{Z}_{DR})$. $R(K_{DP})$ or $R(A_H)$ are insensitive to such uncertainty, and they can outperform $R(Z_H^C, \hat{Z}_{DR})$ if they are further optimized. In addition, a single relationship between $R$ and radar measurements might not be applicable to all range gates within the radar coverage; for example, $R(Z_H)$, $R(K_{DP})$, and $R(Z_H, Z_{DR})$ relationships listed in Table 3 are different; therefore, the residual differences between radar estimates and gauge measurements are still significant for $R(Z_H^C)$, $R(K_{DP})$, and $R(Z_H^C, \hat{Z}_{DR})$. Merging radar with gauge measurements may partly reduce such differences if surface gauge rainfall bias caused by strong wind can be mitigated effectively.

(iv) The vertical gap between radar measurements and surface hinders deriving more optimal relationships and the complete vertical view of the microphysical processes, which is critical in precipitation events such as this super typhoon case. Sophisticated correction models are necessary to mitigate uncertainty cased by the vertical gap, such as the classical models for vertical extrapolation if only radar measurements on higher altitudes are available, either caused by complete beam blockage of mountainous terrain or the high altitudes of radar sites. Efficient implementation of the correction models requires prior knowledge of vertical microphysical precipitation variations. Still, the precipitation process should be determined to effectively match the model with radar measurements . In this typhoon case, the microphysical process is much more complicated, but if the coalescence-breakup balance of the collision process can be measured quantitatively and incorporated into radar QPE algothms in the future, a more reasonable model can be established to enhance radar QPE performance.

## 4. Summary

This paper utilized a range of data, including observations from WZ-SPOL radar, disdrometers, and gauge rainfall measurements, to analyze the microphysical processes during the landfall of Lekima (2019). The investigation focused on demonstrating the impacts of precipitation microphysics on the consistency of multi-source measurements and radar QPE performance. The main findings are summarized as follows:

(i) Measurements from radar, disdrometers, and rain gauges are more consistent after the QC processing, including attenuation correction of radar observations, and wind and hail/graupel processing of size–velocity measurements from disdrometers.

(ii) The breakup overwhelms coalescence as the primary outcome of the collision process of raindrops, noticeably making radar-measured $Z_{DR}^C$-$Z_H^C$ be breakup-dominated, which accounts for that high drop concentration rather than large drop size contributes more to a given $Z_H^C$ and the residual deviation of $Z_{DR}^C$ from $\hat{Z}_{DR}$.

(iii) $R(Z_H^C)$ performs comparably well with $R(K_{DP})$ owing to attenuation correction, but $R(Z_H^C, Z_{DR}^C)$ performs worse with serious overestimation. This is related to the unique microphysical process around the GWS of YDM, in which the breakup-dominated small-sized drops in radar sampling volumes were located above the surface but coalescence-dominated large-sized drops were near the surface.

(iv) $R(Z_H^C, \hat{Z}_{DR})$ outperforms $R(Z_H^C)$ and $R(K_{DP})$ in terms of the $E_{RMS}$ and $E_{NMA}$ scores, and the utilization of $\hat{Z}_{DR}$ instead of $Z_{DR}^C$ is close to physically converting breakup-dominated measurements in radar range gates to coalescence-dominated counterparts, which boosts better self-consistency between $Z_H^C$, $\hat{Z}_{DR}$ and $K_{DP}$, and their consistency with the surface counterparts derived from disdrometer measurements.

The complex precipitation microphysics may have other unknown impacts on the self-consistency of radar measurements and the consistency between multi-source datasets, which is still a challenge for future research. An in-depth understanding of such microphysical processes is critical for improving radar quantitative remote sensing of precipitation. Deployment of cost-effective zenith radar (X- or Ka-band) networks may be an effective complement of operational weather radar networks. Collaborative observaitons of various remote sensing facilities such as these can not only help to resolve more microphysical processes in the vertical gaps currently missed by scanning radars, but also support the development of more reasonable models to mitigate the resulting application uncertainty, especially in complex terrain regions.

**Author contributions**

Yabin Gou carried out the data collection and detailed analysis. He was also part of the polarimetric radar data processing and product generation team. Haonan Chen supervised the work and provided critical comments. Yabin Gou and Haonan Chen wrote the manuscript. Hong Zhu contributed to critical comments and revisions. Lulin Xue reviewed and edited the manuscript.

**Declaration of Competing Interest**

The authors declare that they have no known competing financial interests or personal relationships that could have appeared to influence the work reported in this paper.

**Acknowledgments**

This research is primarily supported by the National Natural Science Foundation of China under Grants 41375038 and the Zhejiang Provincial Natural Science Fund through award LY17D050001. The work of H. Chen is supported by Colorado State University and the National Oceanic and Atmospheric Administration (NOAA) through Cooperative Agreement NA19OAR4320073. The authors acknowledge the anonymous reviewers for their careful review and comments on this article. We also thank Dr. Lin Deng in Shanghai Typhoon Institute of China Meteorological Administration for the discussion on typhoon microphysical processes, Senior Engineer Yuanyuan Zheng and Fen Xu in Jiangsu Institute of Meteorological Sciences for the discussion on radar measurements during this particular event, and Bo Si and Xiaolong Huang for double checking the locations and measurement quality of meteorological stations. The S-band polarimetric radar, disdrometer, and rain gauge data are provided by the Chinese Meteorological Administration and are available upon request.

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

**Table 1.** Evaluation scores of six-hour rainfall accumulations based on six radar QPE relationships.

| Scores | Radar QPE relationships | | | | | |
|---|---|---|---|---|---|---|
| | $R(Z_H^M)$ | $R(Z_H^C)$ | $R(K_{DP})$ | $R(Z_H^M,Z_{DR}^M)$ | $R(Z_H^C,Z_{DR}^C)$ | $R(Z_H^C,\hat{Z}_{DR})$ |
| $E_{RMS}$(mm) | 35.2066 | 50.0166 | 48.4374 | 82.269 | 97.2031 | 45.6924 |
| $E_{NMA}$ | 29.0485 | 31.9173 | 30.8652 | 42.499 | 44.6513 | 30.3174 |
| $E_{CC}$ | 0.7634 | 0.7954 | 0.7995 | 0.7075 | 0.7201 | 0.7971 |

**Table 2.** Evaluation scores of $R(Z_H^C,\hat{Z}_{DR})$ calculated by different datasets.[1]

| Scores | The DSD dataset to estimate $\hat{Z}_{DR}$ and to derive $R(Z_H^C,\hat{Z}_{DR})$ | | | |
|---|---|---|---|---|
| | $S_{III}$ | $S_0$ | $S_I$ | $S_{II}$ |
| $E_{RMS}$(mm) | 40.0033 | 45.6924 | 65.3023 | 82.8893 |
| $E_{NMA}$ | 28.6757 | 30.3174 | 36.2891 | 41.2624 |
| $E_{CC}$ | 0.7940 | 0.7971 | 0.7905 | 0.7879 |

[1] $S_{III}$ includes more size–velocity pairs than $S_0$

**Table 3.** Radar QPE relationships at six different meteorological stations.

| Stations | Radar QPE Relationships | | |
|---|---|---|---|
| | $R(Z_H)$ | $R(K_{DP})$ | $R(Z_H, Z_{DR})$ |
| XJ | $0.0502 \times Z_H^{0.6332}$ | $50.3159 \times K_{DP}^{0.7755}$ | $0.0077 \times Z_H^{0.9308} Z_{DR}^{-4.0151}$ |
| LH | $0.0397 \times Z_H^{0.6678}$ | $53.0847 \times K_{DP}^{0.7775}$ | $0.0093 \times Z_H^{0.909} Z_{DR}^{-3.9326}$ |
| HJ | $0.0202 \times Z_H^{0.7398}$ | $58.0381 \times K_{DP}^{0.8320}$ | $0.0077 \times Z_H^{0.9390} Z_{DR}^{-4.2782}$ |
| DT | $0.0332 \times Z_H^{0.6775}$ | $41.8480 \times K_{DP}^{0.8314}$ | $0.0062 \times Z_H^{0.9526} Z_{DR}^{-4.1799}$ |
| YH | $0.0174 \times Z_H^{0.7131}$ | $45.1785 \times K_{DP}^{0.8264}$ | $0.0084 \times Z_H^{0.9086} Z_{DR}^{-3.5505}$ |
| WL | $0.0203 \times Z_H^{0.6891}$ | $54.1236 \times K_{DP}^{0.8177}$ | $0.0072 \times Z_H^{0.9426} Z_{DR}^{-4.0677}$ |

(a) High waves along WL coast  (b) Landslide in Yong Jia town  (c) Waterlogging in WL town

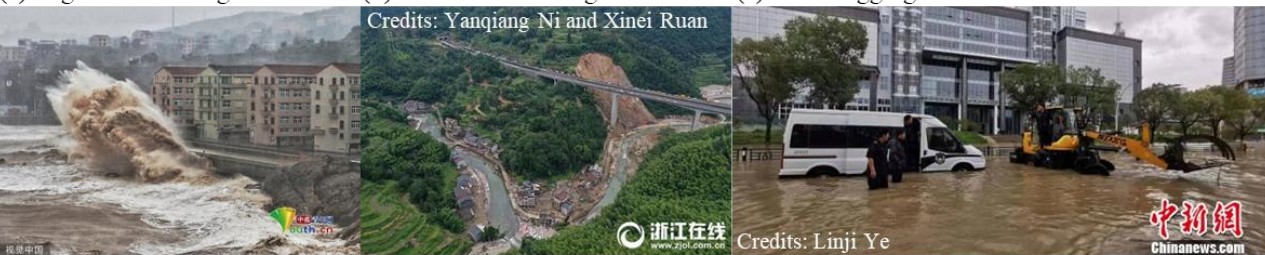

(d) Waterlogging in LH town          (e) Waterlogging in YH town  (f) Waterlogging in XJ town

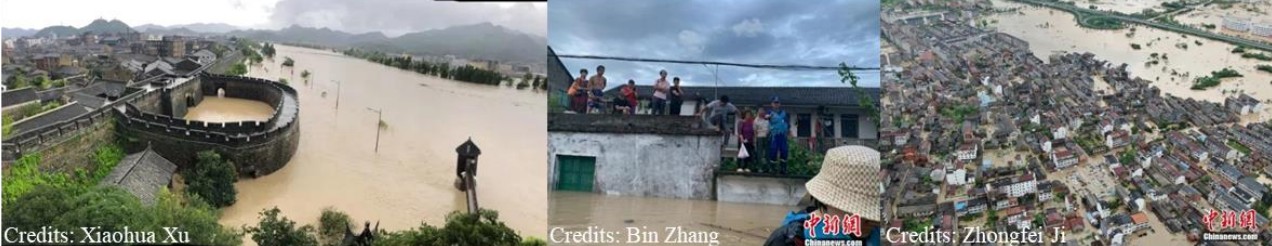

**Fig. 1. The disastrous situation in WZ and TZ due to the landfall of super typhoon Lekima: (a) high waves along Wen Ling (WL) coast of TZ city; (b) landslide in the northern mountain area of Yong Jia (YJ) in WZ city; (c)-(f) serious waterlogging in WL, Lin Hai (LH), Yu Huan (YH), and Xian Ju (XJ) town of TZ city.**
**Photo (a) is available at http://picture.youth.cn/qtdb/201908/t20190810_12036586.htm.**
**Photo (b) is available at https://baijiahao.baidu.com/s?id=1641647981934061656&wfr=spider&for=pc.**
**Photo (c) is available at  https://new.qq.com/omn/20190810/20190810A0FZUT00.html?pc.**
**Photo (d) is available at https://new.qq.com/omn/20190828/20190828A0KGLT00.html.**
**Photo (e) is available at https://www.newssz.com/sz/2019/0818/94241-1/.**
**Photo (f) is available at https://m.chinanews.com/wap/detail/undefined/zw/8925613.shtml.**

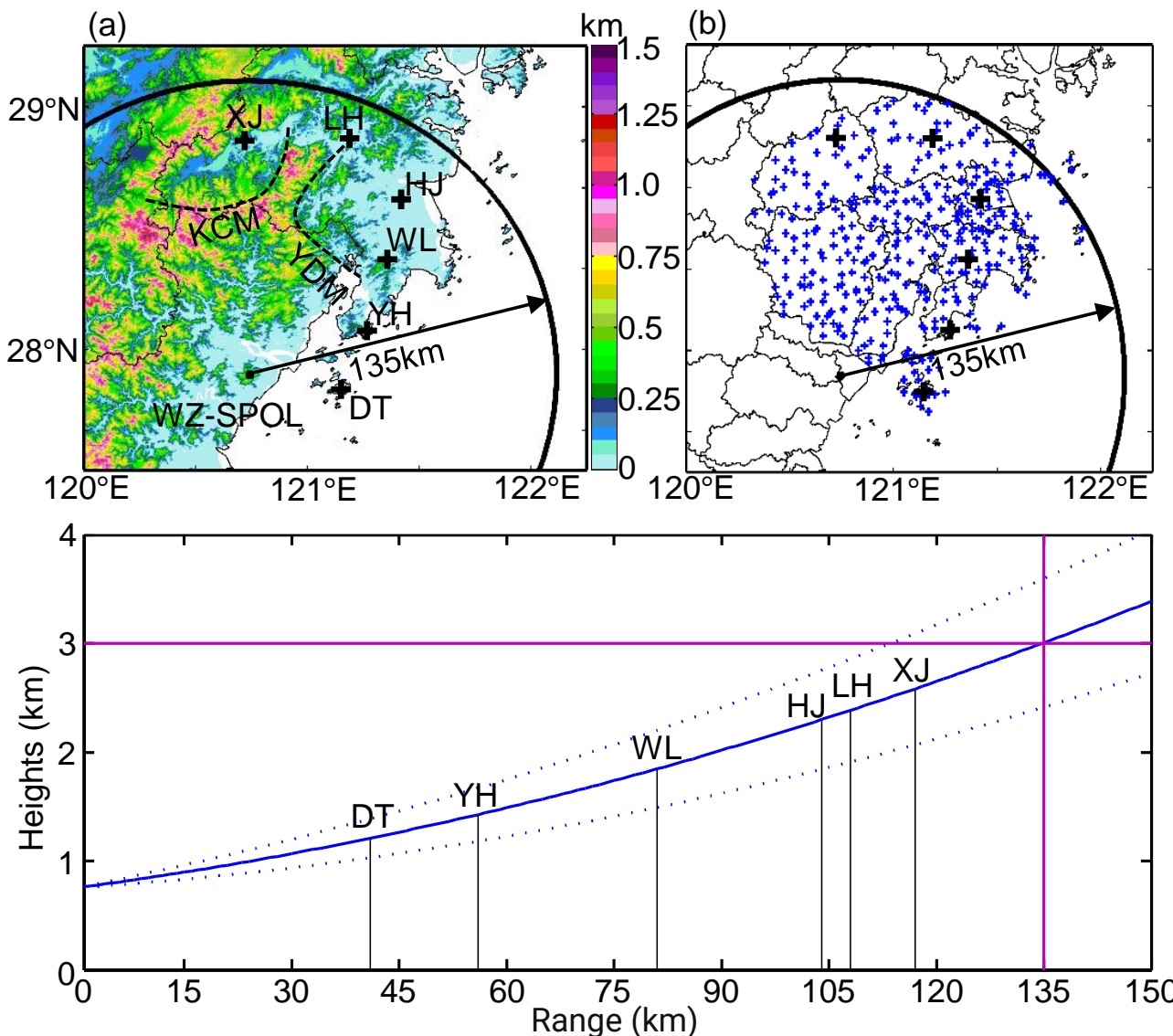


**Fig. 2. (a) Terrain elevation and disdrometer network around the WZ-SPOL radar (735 m); (b) rain gauge network around the disaster center area; (c) the height of radar beam shown as a function of measurement range in standard atmospheric conditions. Two dashed lines refer to the GWS of YDM and KCM. The black "+" in (a) and (b) refer to six national meteorological sites and the blue "+" in (b) refers to regional meteorological sites. The solid and dotted blue curves in (c) refer to the height of the radar beam center and its radius boundaries; the vertical black lines mark the range distance of national meteorological stations (heights<0.1 km) from radar; two orthogonal purple lines refer to the altitude of 3 km and range of 135 km.**



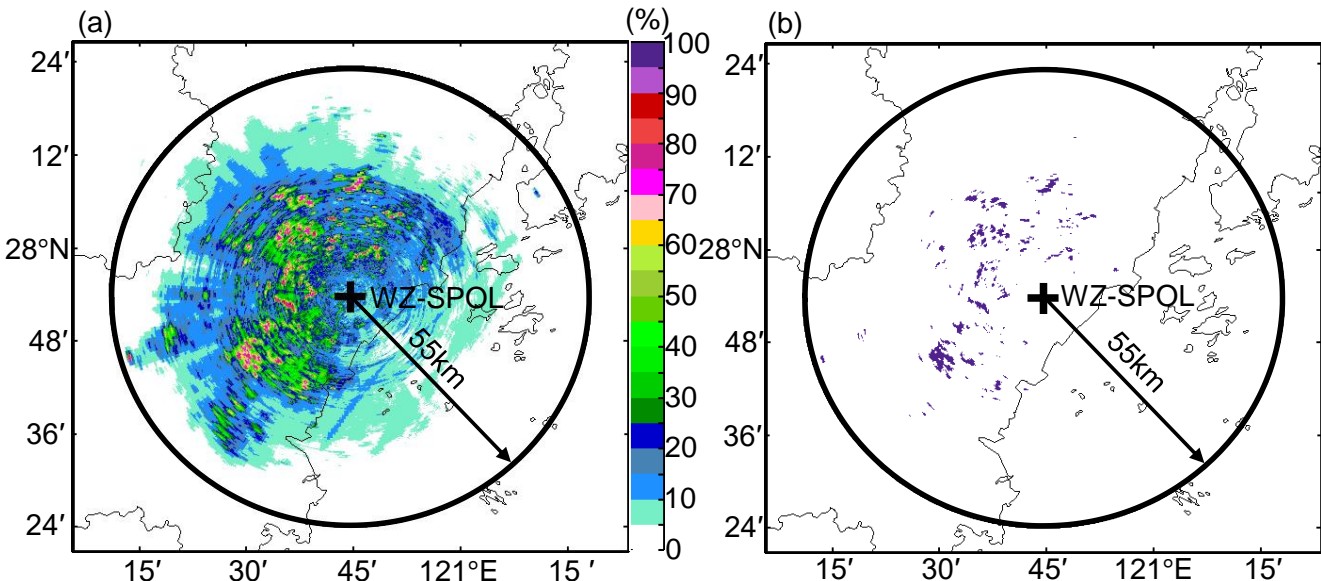

Fig. 3. (a) Statistics of pixels with $Z_H$>0 dBZ within 55 km from the WZ-SPOL radar; (b) residual static ground clutter mask of the
       WZ-SPOL radar.

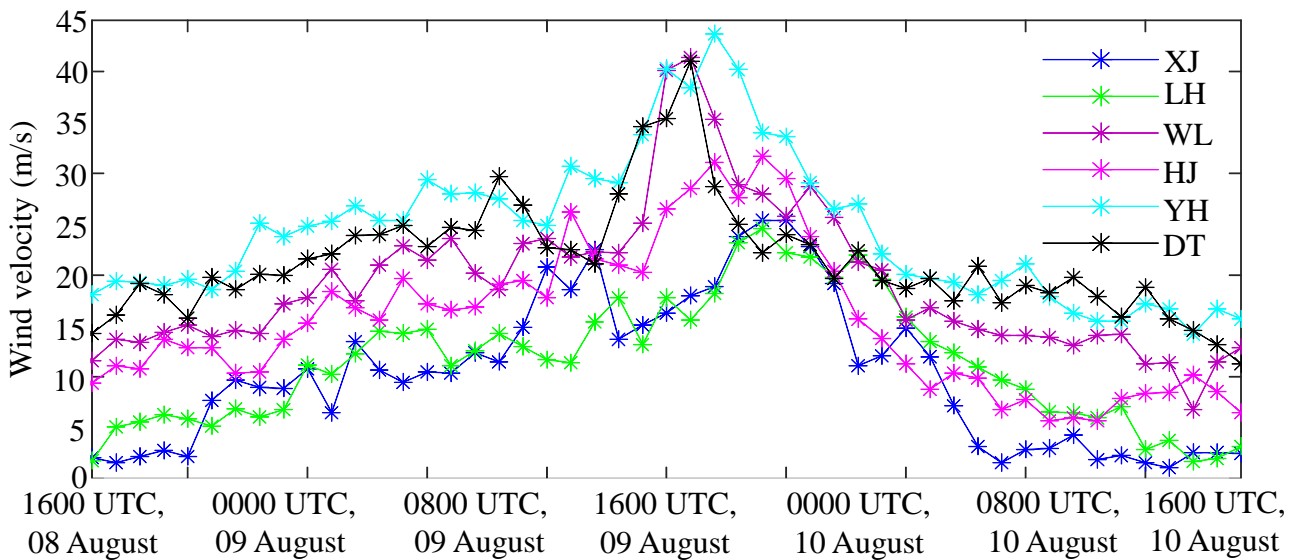

**Fig. 4. Time series of hourly maximum wind speed at the six national meteorological stations between 1600 UTC, 08 August 2019 and 1600 UTC, 10 August 2019.**


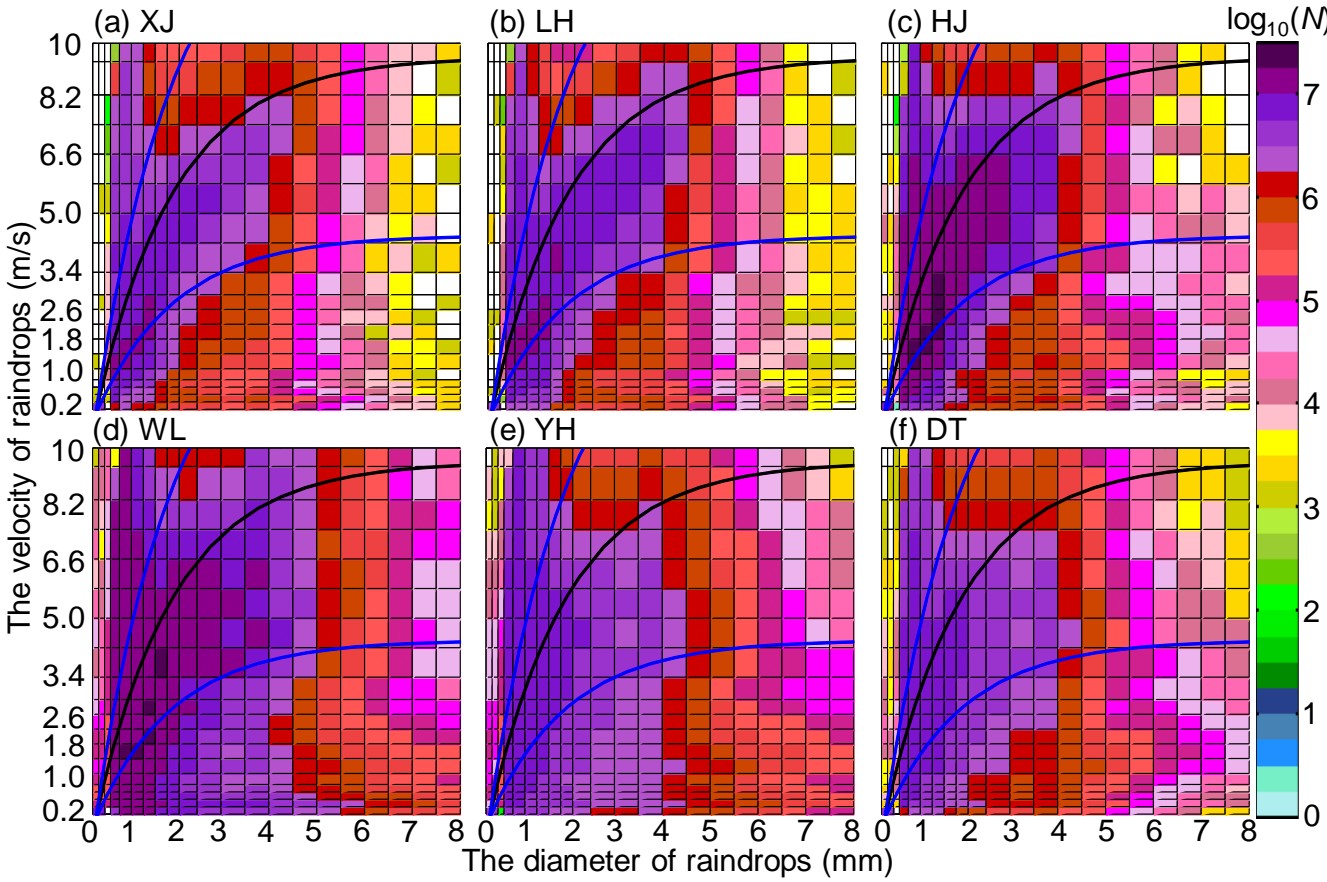

**Fig. 5.** The original size–velocity dataset collected at (a) XJ, (b) LH, (c) HJ, (d) WL, (e) YH, and (f) DT. The black and blue lines refer to the fall speed $V_B$, $0.5V_B$, and $1.5V_B$ calculated in Eq. 3, respectively.

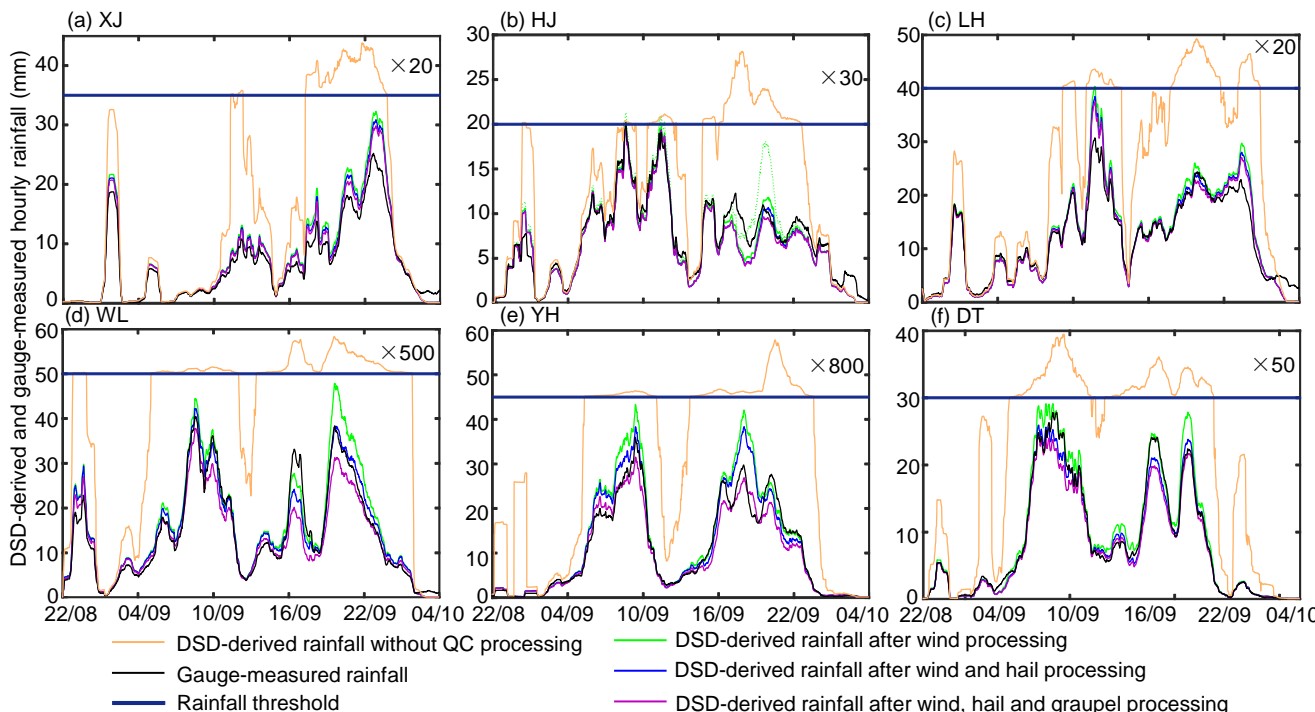

Fig. 6. Time series of DSD-derived and gauge hourly rainfall: (a)-(f) are obtained from XJ, HJ, LH, WL, YH, and DT, respectively, during 2200 UTC 08 August 2019 and 0400 UTC 10 August 2019. The number following "×" refers to $C_T$, and bold dark blue straight lines indicate the threshold of $R_T$ of each station according to Eq. (5). The green dotted line in (b) is conditioned by $V_M \in [0.4\ V_B, 1.5V_B]$, and other green solid lines are conditioned by $V_M \in [0.5V_B, 1.5V_B]$.

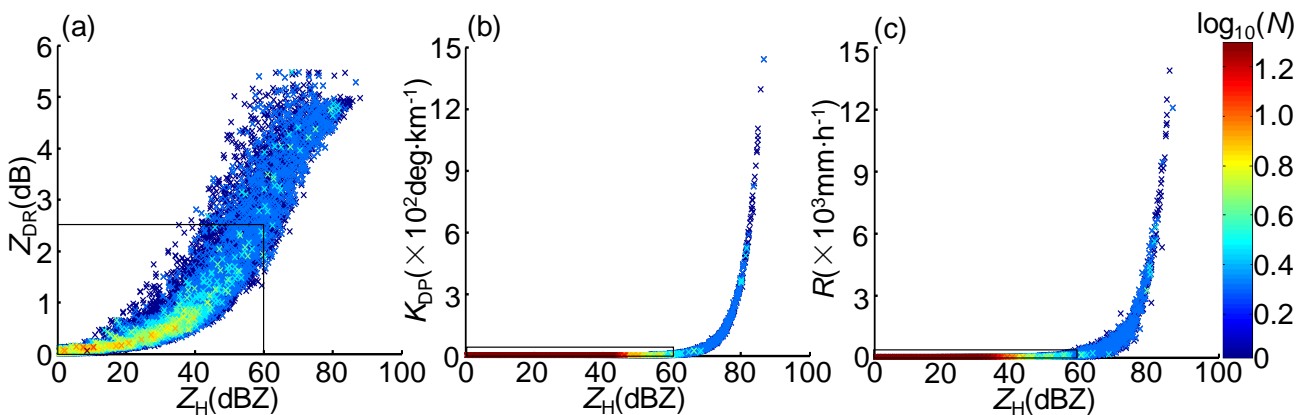

**Fig. 7.** The scattergrams of DSD-derived polarimetric radar variables without QC processing: (a) $Z_{DR}$ vs. $Z_H$; (b) $K_{DP}$ vs. $Z_H$; (c) $R$ vs. $Z_H$. The rectangles in (a)-(c) indicate the ranges of DSD-derived variables after final QC processing.

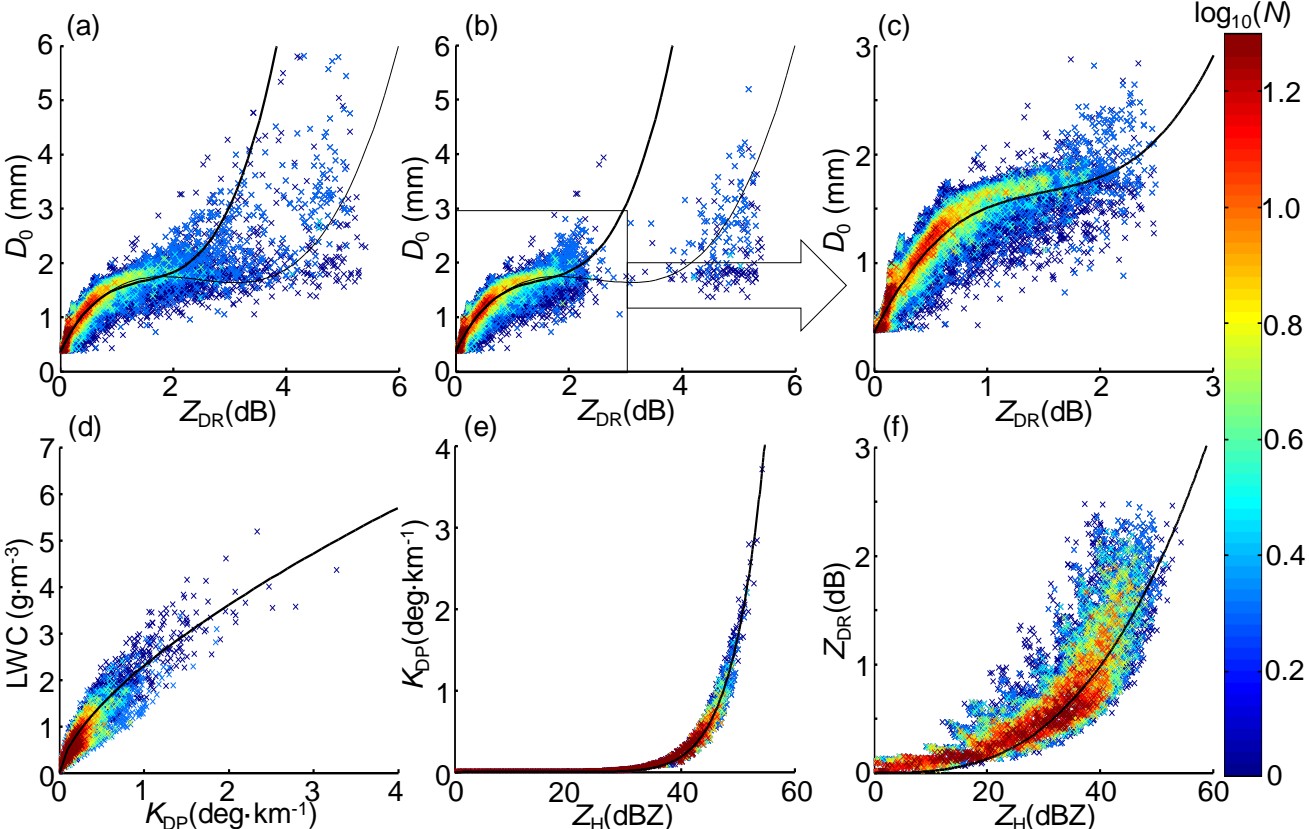

**Fig. 8.** Scattergrams between polarimetric radar variables: (a) $D_0$ vs. $Z_{DR}$ after eliminating wind contaminations. (b) is based on (a), but after removing the hail and graupel contaminations further. (c) is based on (b), but after further eliminating the residual graupel contaminations with $Z_{DR}>2.5$dB. (d), (e) and (f) are LWC vs. $K_{DP}$, $K_{DP}$ vs. $Z_H$, and $Z_{DR}$ vs. $Z_H$ based on the same dataset as (c). The thick black lines in (a)-(c) stand for Eq. 5; the thin black lines in (a) and (b) indicate the overfitted results, and the black curves in (d)-(f) stand for Eq. 6, Eq. 7 and Eq. 2a, respectively.

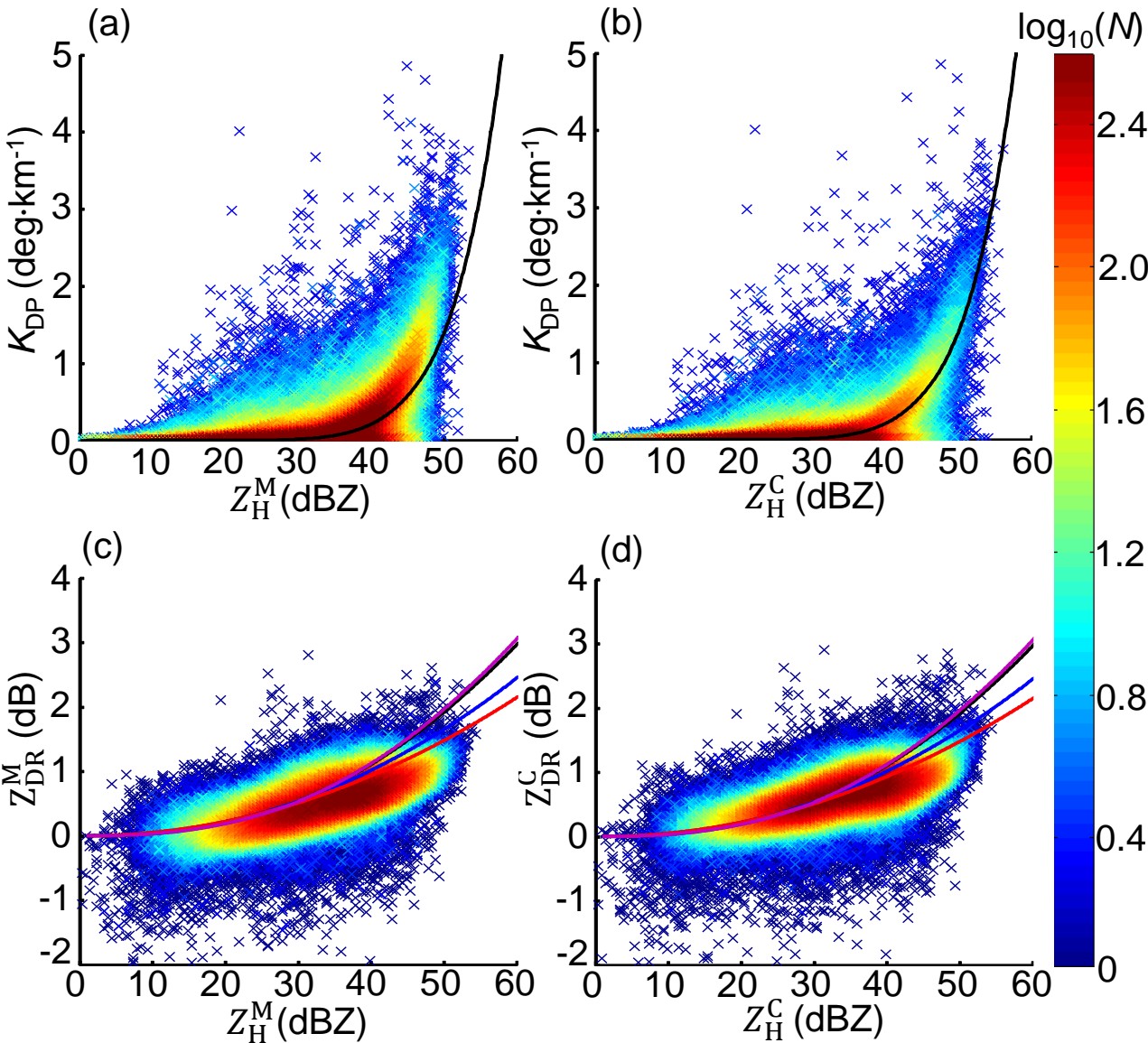


Fig. 9. The scattergram between polarimetric measurements from WZ-SPOL radar: (a) $K_{DP}$ vs $Z_H^M$; (b) $K_{DP}$ vs $Z_H^C$; (c) $Z_{DR}^M$ vs $Z_H^M$; (d) $Z_{DR}^C$ vs $Z_H^C$. Measurements of all six stations derive the black curves; the blue, red, and purple curves in (c) and (d) stand for Eqs.11a-11c derived from $S_I$-$S_{III}$.

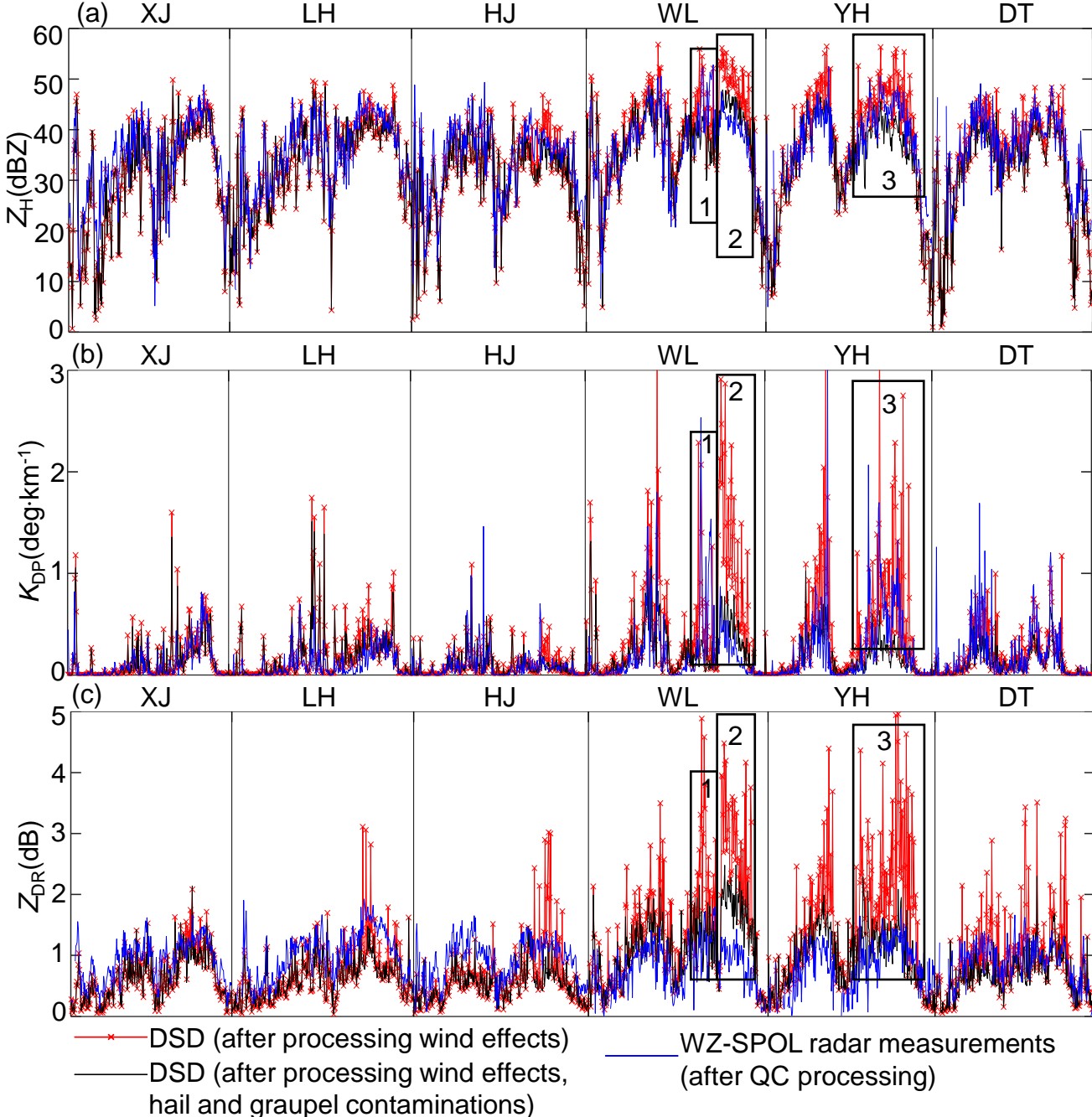

Fig. 10. (a) Time series of radar-measured $Z_H^C$ and DSD-derived $Z_H$ at the six meteorological stations shown in Fig. 6 during 2200 UTC 08 August 2019 and 0400 UTC 10 August 2019; (b) Similar to (a), but for radar-measured $K_{DP}$ and DSD-derived $K_{DP}$; (c) Similar to (a), but for radar-measured $Z_{DR}^C$ and DSD-derived $Z_{DR}$.


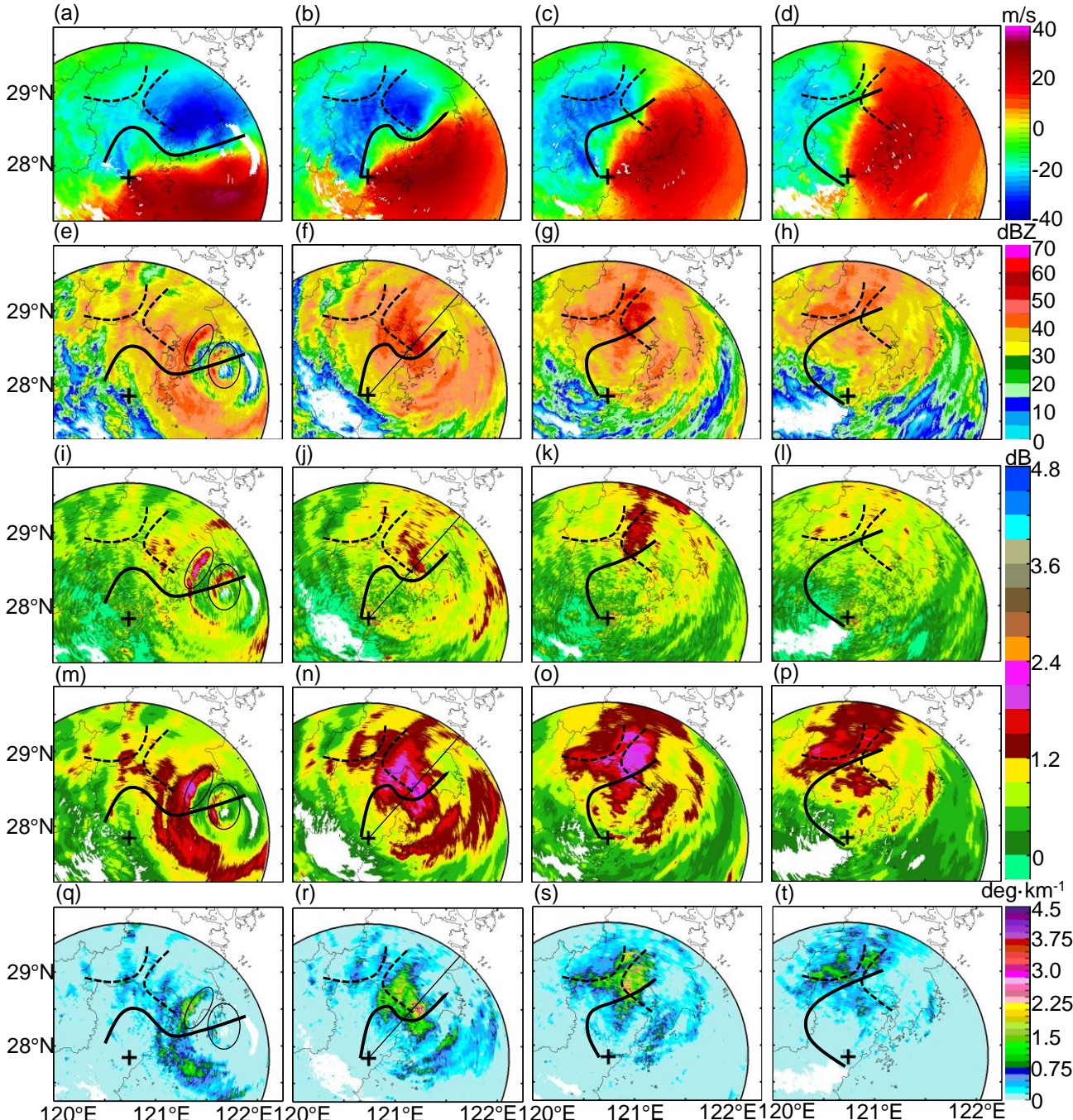

**Fig. 11. WZ-SPOL radar measurements during typhoon Lekima: (a)-(d) are Doppler velocity $V_R$ at 1601 UTC, 1759 UTC, 2002 UTC, and 2200 UTC, 09 August 2019, respectively; (e)-(h), (i)-(l), (m)-(p), (q)-(t) are $Z_H^C$, $Z_{DR}^C$, $\hat{Z}_{DR}$ and $K_{DP}$ simultaneously as (a)-(d). The solid black lines refer to wind shear deduced from $V_R$. The black dashed lines refer to the GWS of KCM and YDM, and "+" indicates the location of the WZ-SPOL radar. The ellipses in (e), (i), (m), and (q) indicate where hydrometeor size sorting occurred. The black lines along the radial profiles in (f), (j), (n), and (r) indicate the azimuthal angle shown in Fig. 18.**


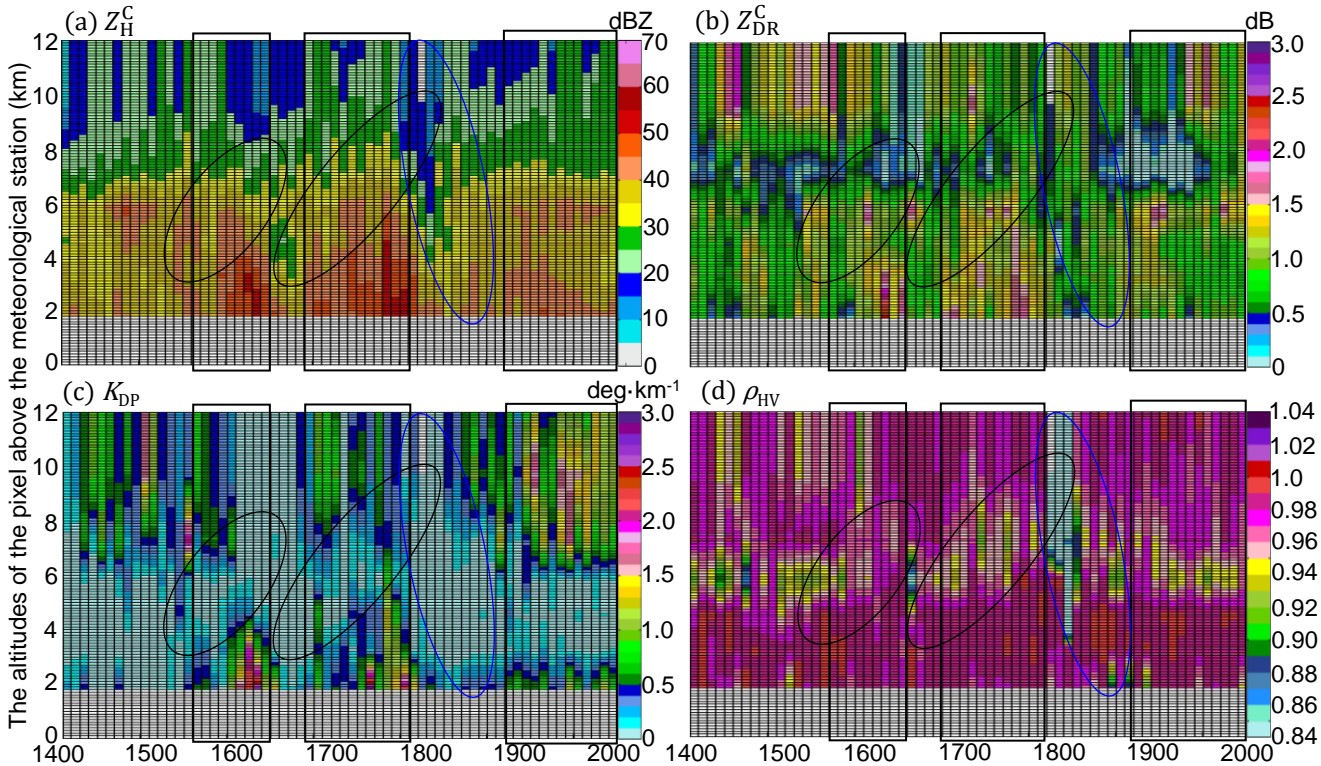

**Fig. 12. (a) Time series of vertical polarimetric radar variables upon the WL station between 1400 UTC 08 August 2019 and 2000 UTC 09 August 2019: (a) $Z_H^C$, (b) $Z_{DR}^C$, (c) $K_{DP}$, and (d) $\rho_{HV}$. The black rectangles indicate developing convective storms; the black ellipses surround the potential updrafts; the blue ellipses surround the subsiding signatures of ice or mixed-phase particles.**

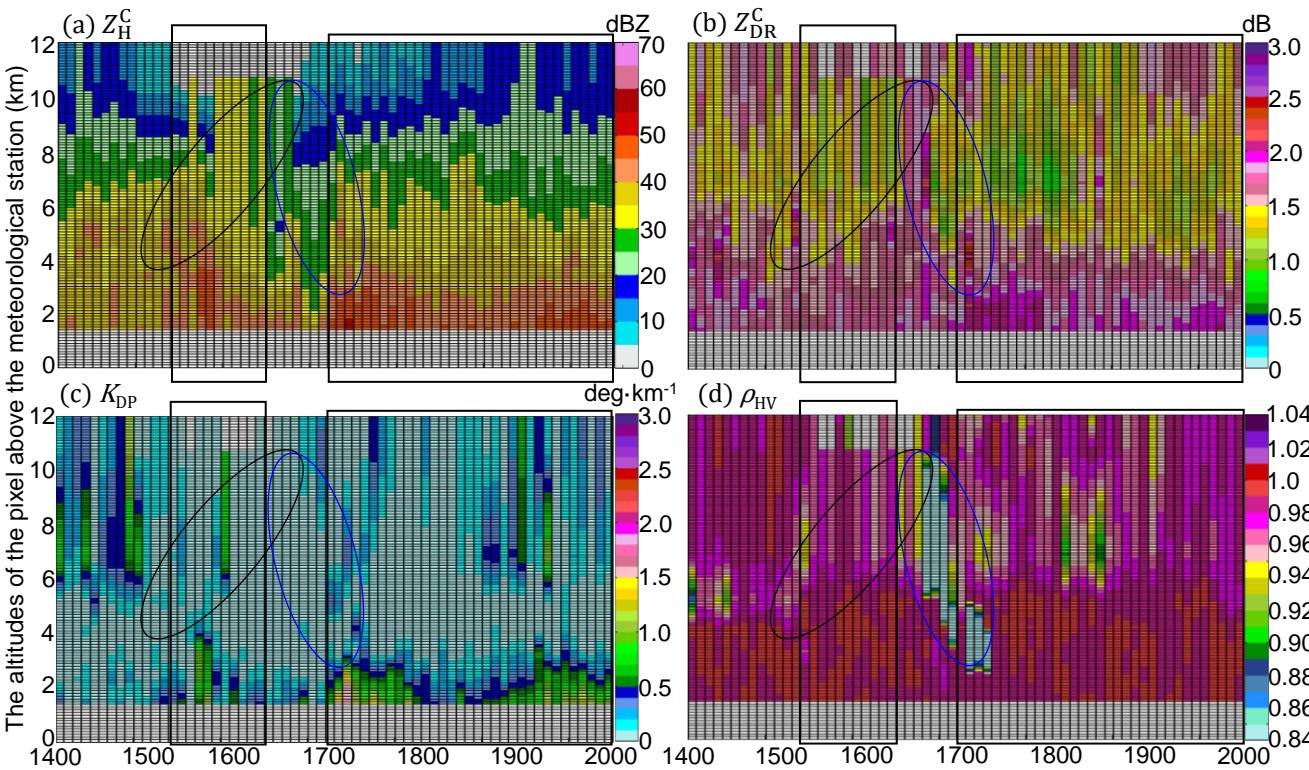


**Fig. 13. Same as Fig. 12 but for the YH station.**

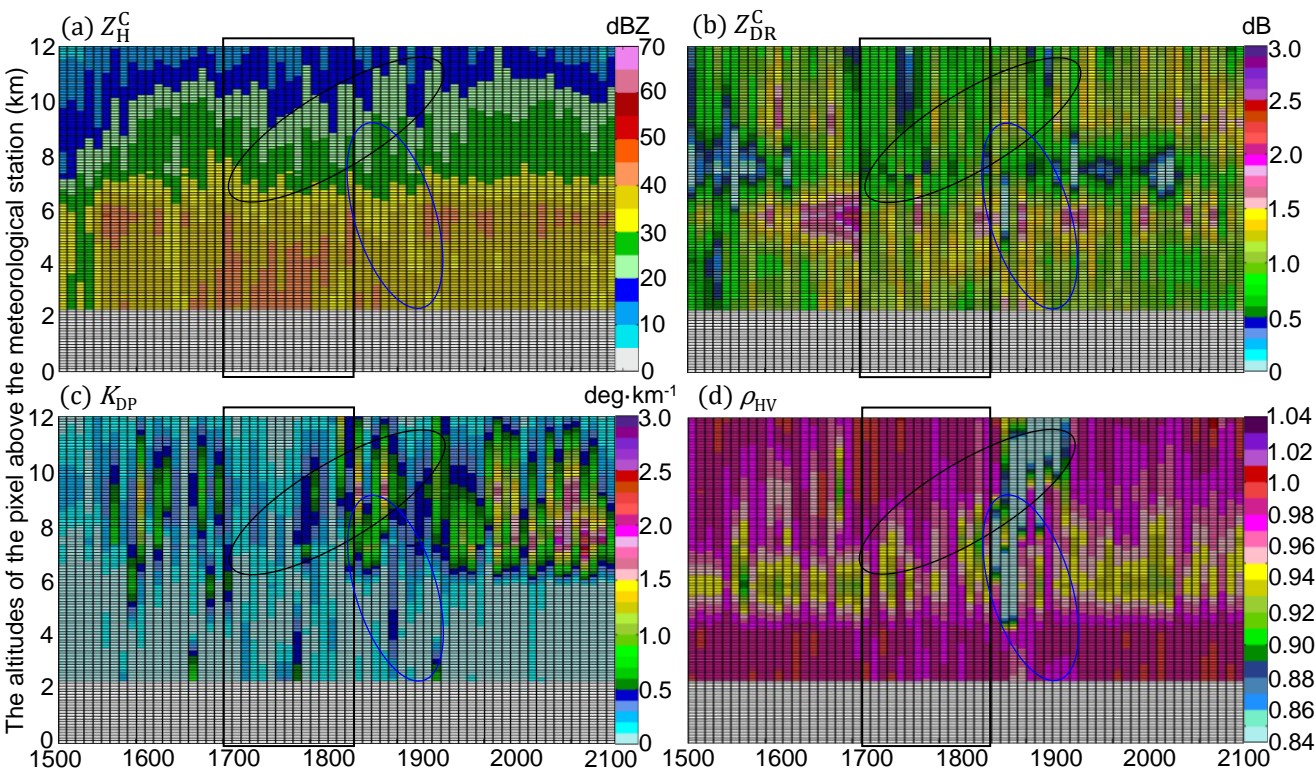

Fig. 14. Same as Fig. 12 but for the HJ station between 1500 UTC 08 August 2019 and 2100 UTC 09 August 2019.

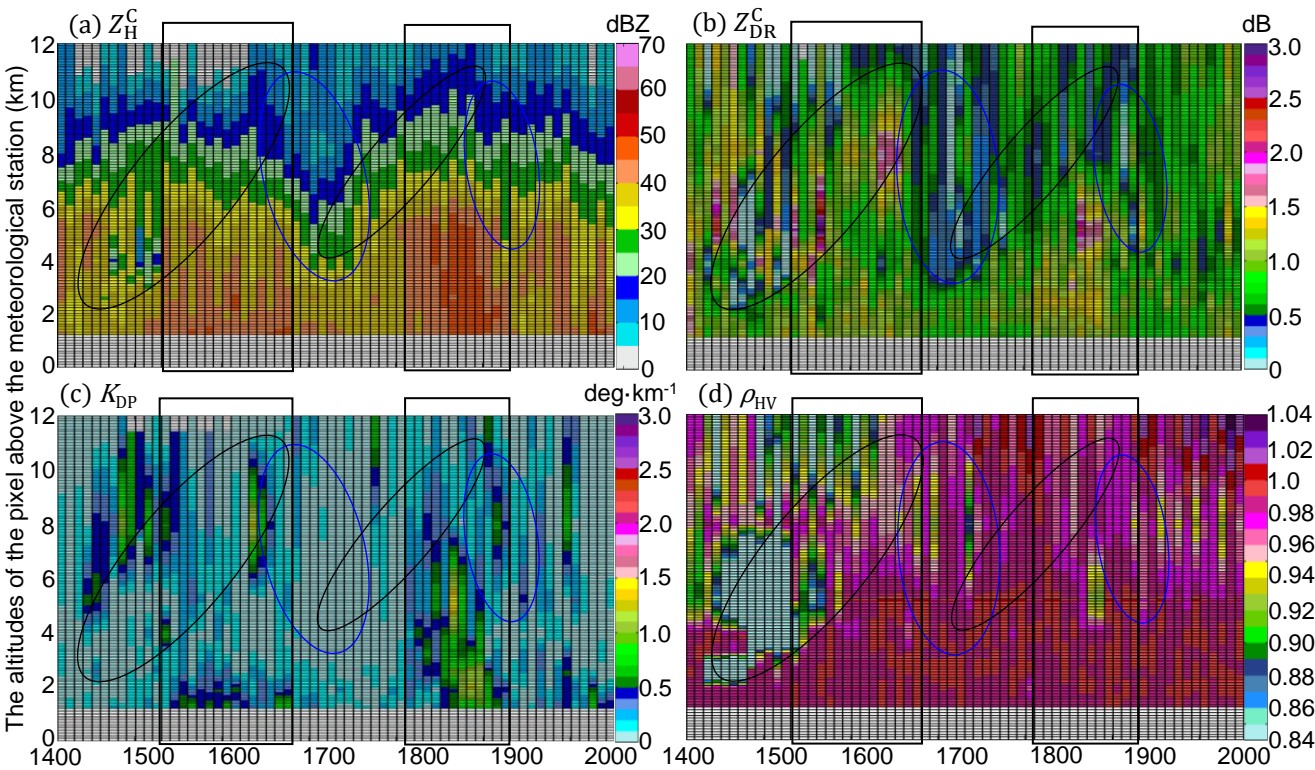

**Fig. 15. Same as Fig. 12 but for the DT station.**

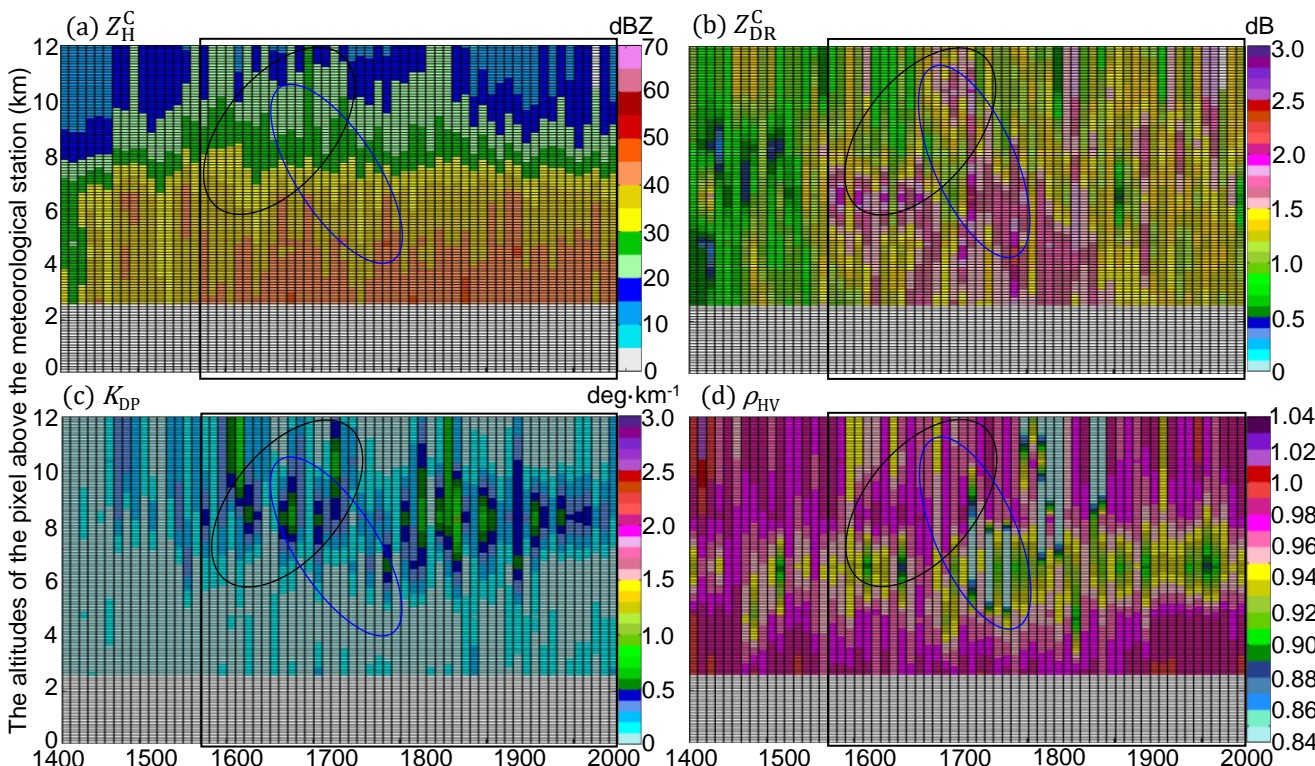

**Fig. 16. Same as Fig. 12 but for the LH station.**

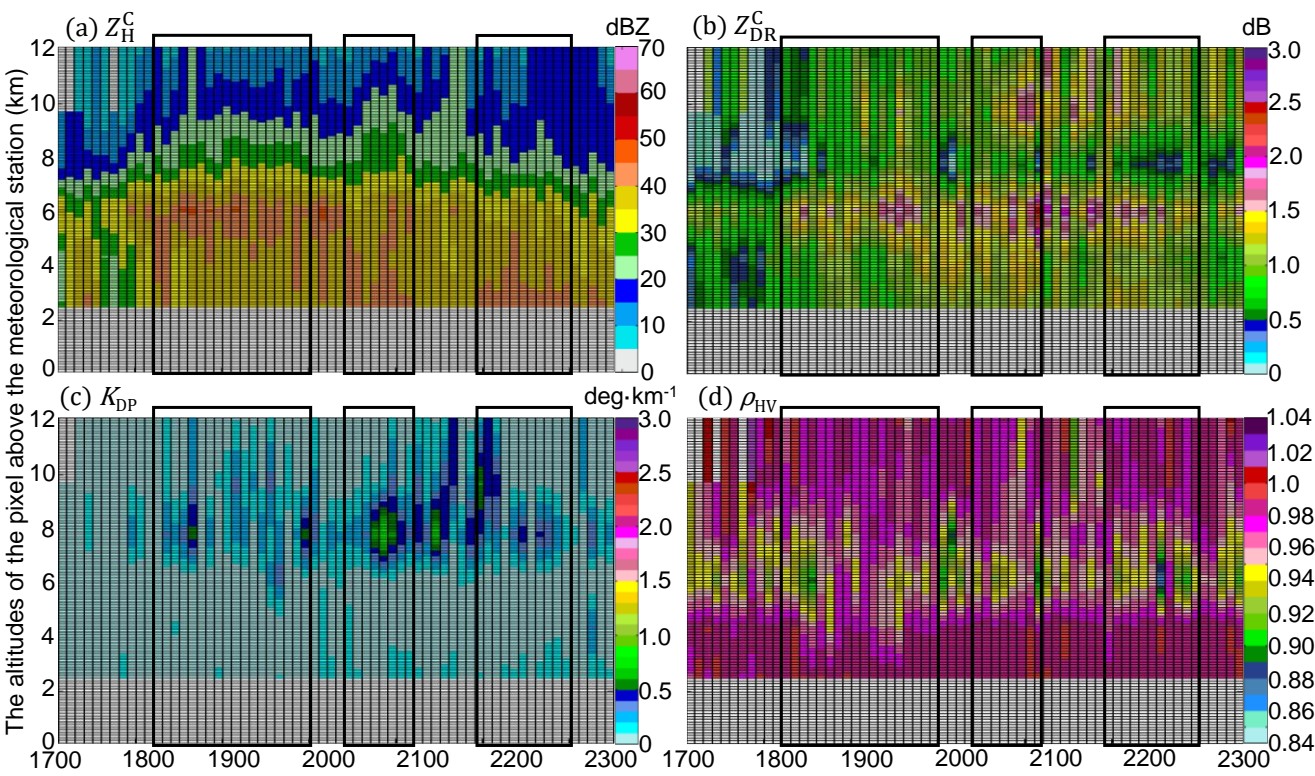


**Fig. 17. Same as Fig. 12 but for the XJ station between 1500 UTC 08 August 2019 and 2100 UTC 09 August 2019.**

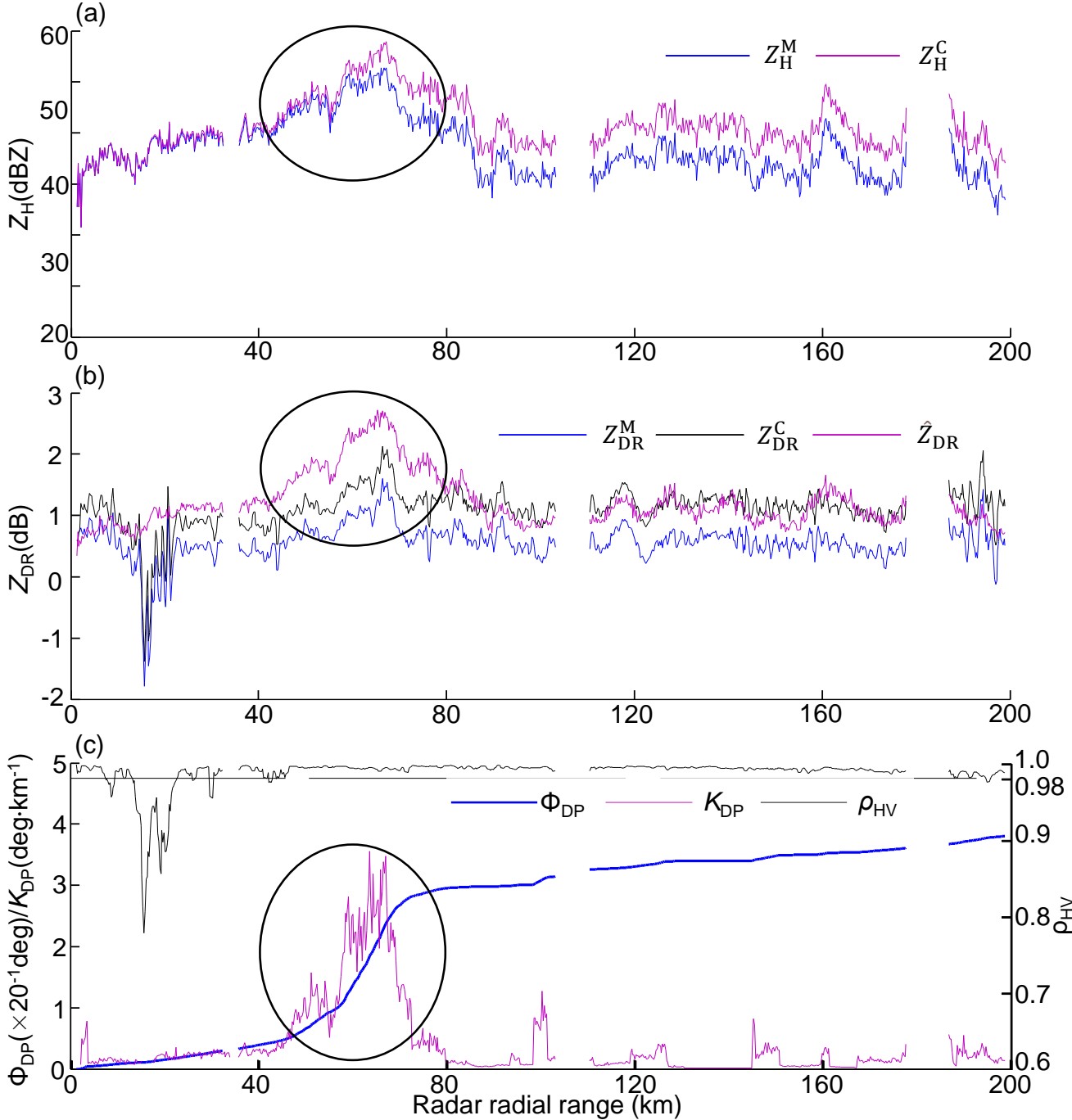

**Fig. 18. WZ-SPOL radar along a radial profile (azimuth angle = 46°) at an elevation angle of 0.5° at 1759 UTC, 09 August 2019:**
**(a) $Z_H^C$ and $Z_H^M$, (b) $Z_{DR}^M$, $Z_{DR}^C$ and $\hat{Z}_{DR}$, (c) $\Phi_{DP}$, $K_{DP}$, and $\rho_{HV}$. This azimuth angle is marked in Fig. 11.**

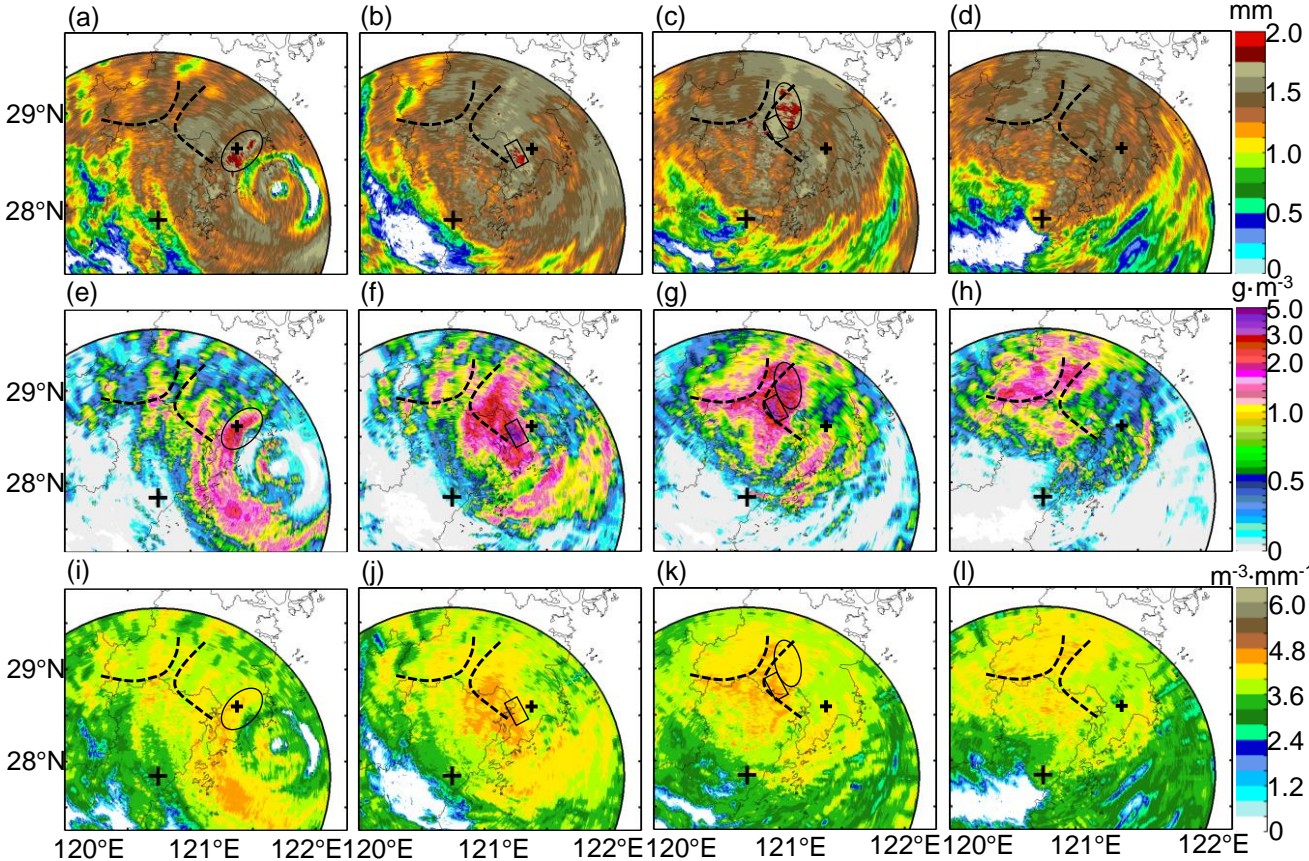

**Fig. 19. Radar-retrieved DSD parameters during typhoon Lekima: (a)-(d) are $D_0$ at 1601 UTC, 1759 UTC, 2002 UTC, and 2200 UTC, 09 August 2019, respectively; (e)-(h) and (i)-(l) (t) are LWC and $\log_{10}(N_w)$ at the same time as (a)-(d). Two dashed lines refer to the GWS of YDM and KCM. The large "+" indicates the location of the WZ-SPOL radar site, and the little "+" indicates the location of the WL station.**

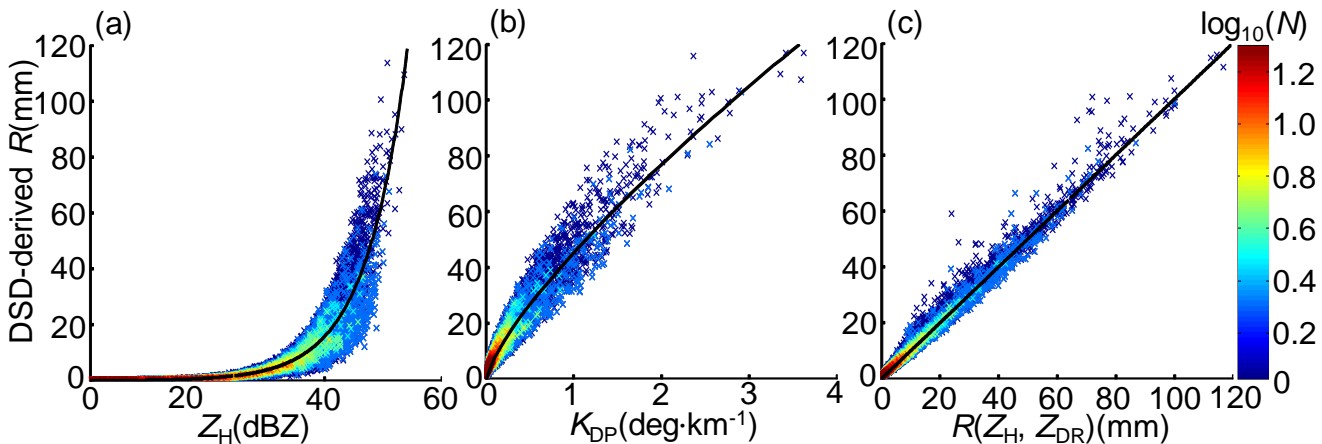

**Fig. 20. The scattergram of (a) DSD-derived $R$ vs. $Z_H$, (b) DSD-derived $R$ vs. $K_{DP}$, and (c) DSD-derived $R$ vs. $R$ estimated by fitted $R(Z_H, Z_{DR})$.**

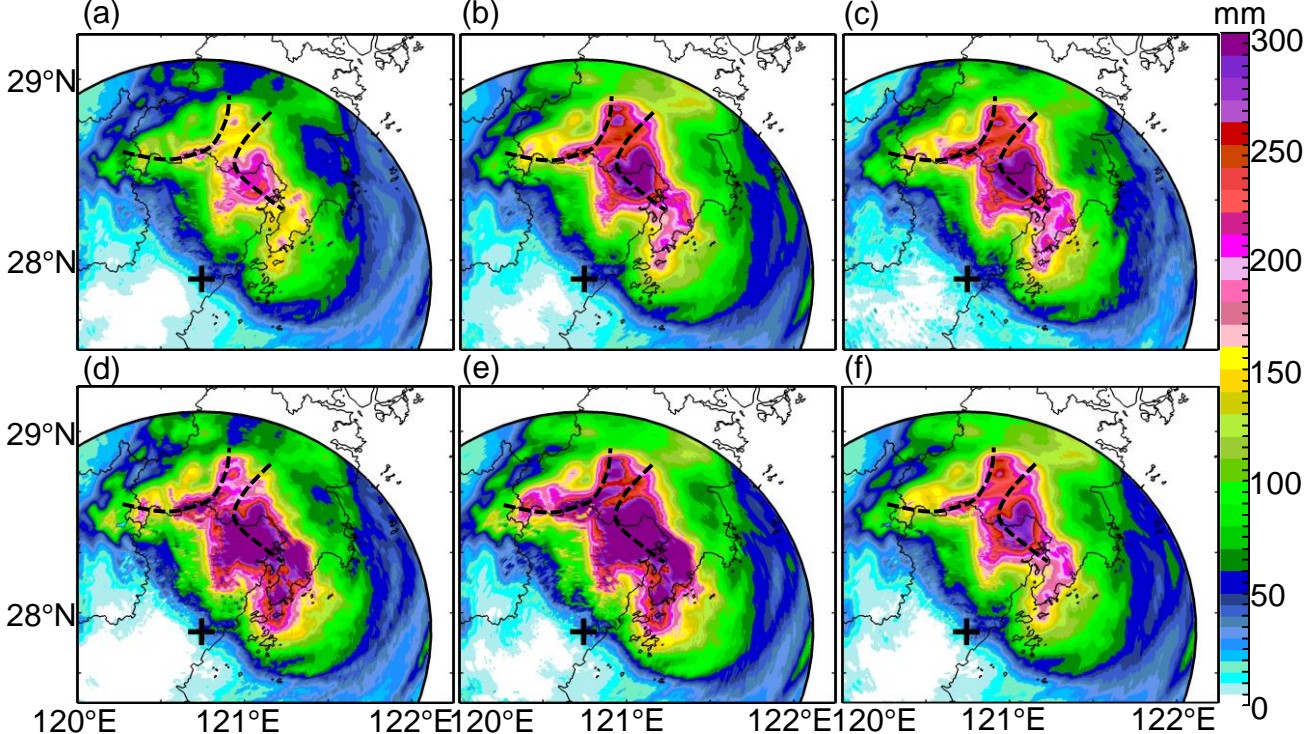

**Fig. 21. Six-hour rainfall estimates derived from (a) $R(Z_H^M)$, (b) $R(Z_H^C)$, (c) $R(K_{DP})$, (d) $R(Z_H^M, Z_{DR}^M)$, (e) $R(Z_H^C, Z_{DR}^C)$, and (f) $R(Z_H^C, \widehat{Z}_{DR})$ at 2200 UTC, 09 August 2019. Two dashed lines refer to the GWS of YDM and KCM, and "+" refers to the WZ-SPOL radar site.**

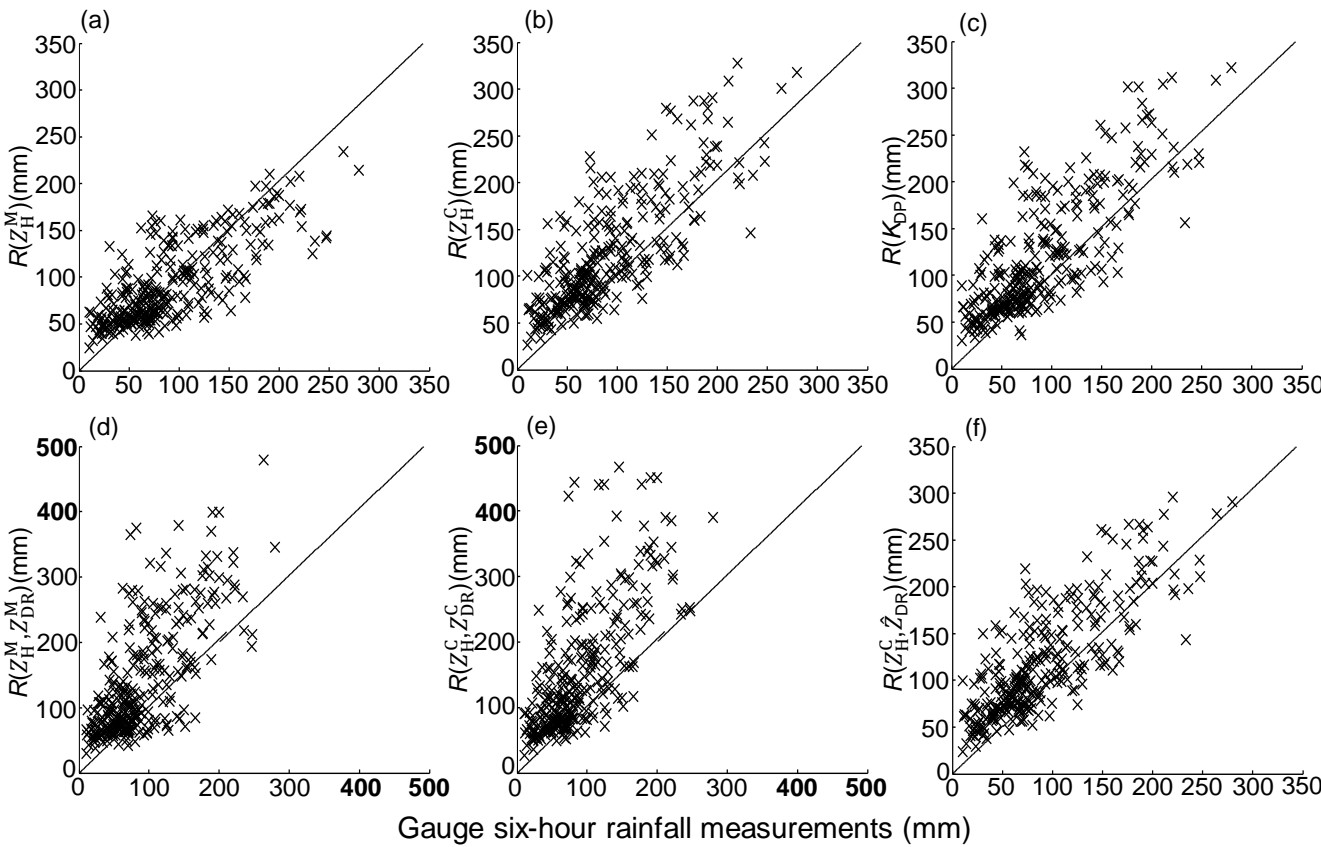

**Fig. 22. The scattergram of six-hour rainfall estimates from radar versus corresponding gauge rainfall measurements. The radar rainfall estimates are derived at 2200 UTC, 09 August 2019, using (a) $R(Z_H^M)$, (b) $R(Z_H^C)$, (c) $R(K_{DP})$, (d) $R(Z_H^M, Z_{DR}^M)$, (e) $R(Z_H^C, Z_{DR}^C)$, and (f) $R(Z_H^C, \widehat{Z}_{DR})$.**

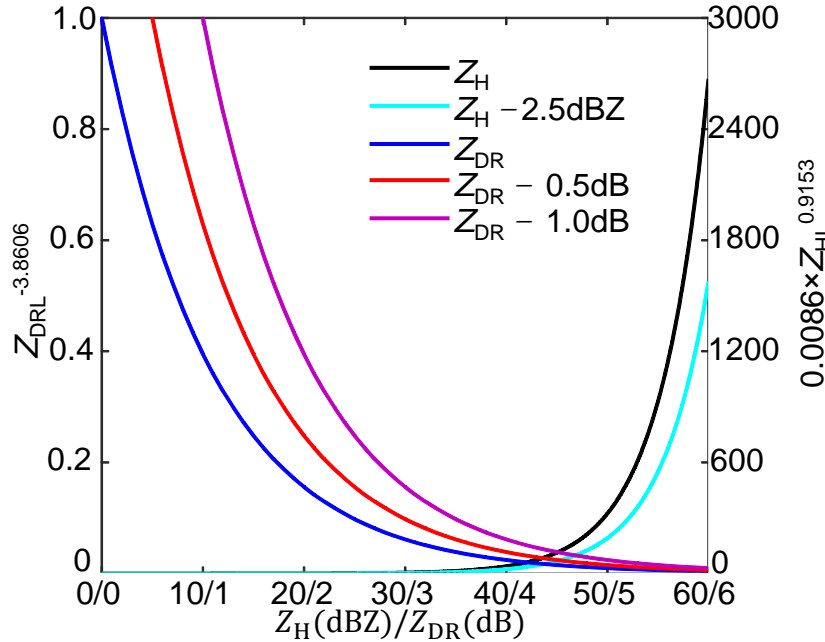

**Fig. 23.** The contribution of $Z_H$-related and $Z_{DR}$-related terms in the $R(Z_H, Z_{DR})$ relationship with different $Z_H$ and $Z_{DR}$ biases. The
$R(Z_H, Z_{DR})$ relationship is detailed in Eq. 12c. $Z_{HL}$ and $Z_{DRL}$ refer to $Z_H$ and $Z_{DR}$ at a linear scale.

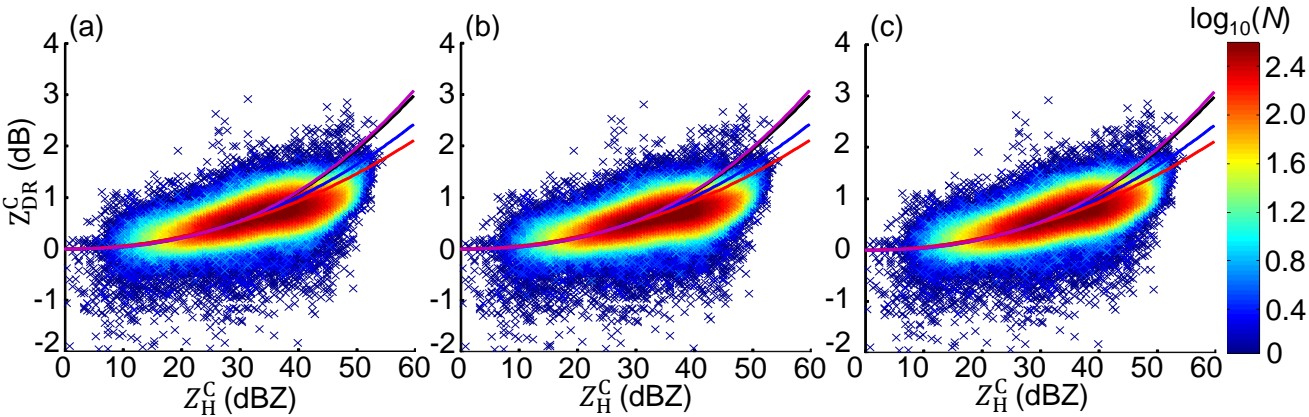

**Fig. 24.** Scattergrams between $Z_{DR}^{C}$ and $Z_{H}^{C}$, respectively, utilizing $\widehat{Z}_{DR}$ estimated by (a) Eq. 11a (the blue curve), (b) Eq. 11b (the red curve), and (c) Eq. 11c (the purple curve). The black curve stands for Eq. 2a.


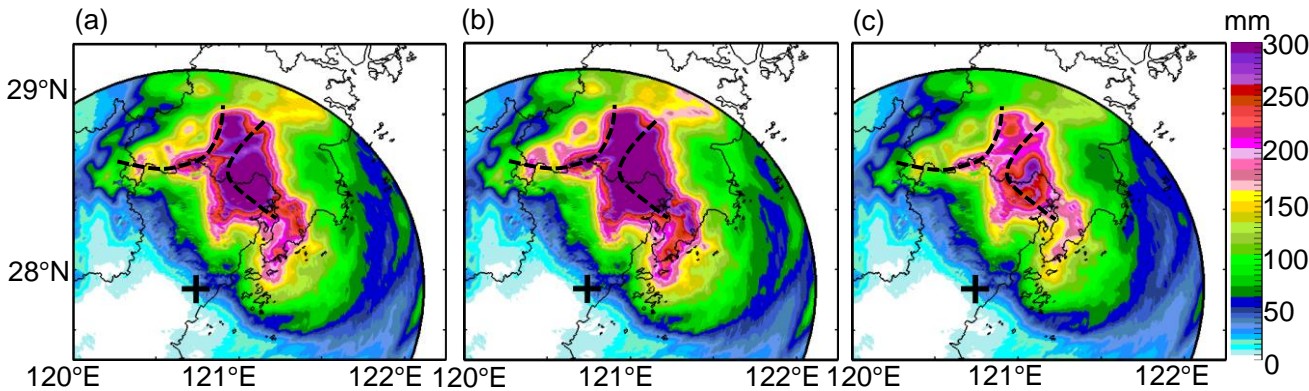

**Fig. 25. The same as Fig. 21, but (a)-(c) were calculated with Eqs. 14a-14c, respectively.**

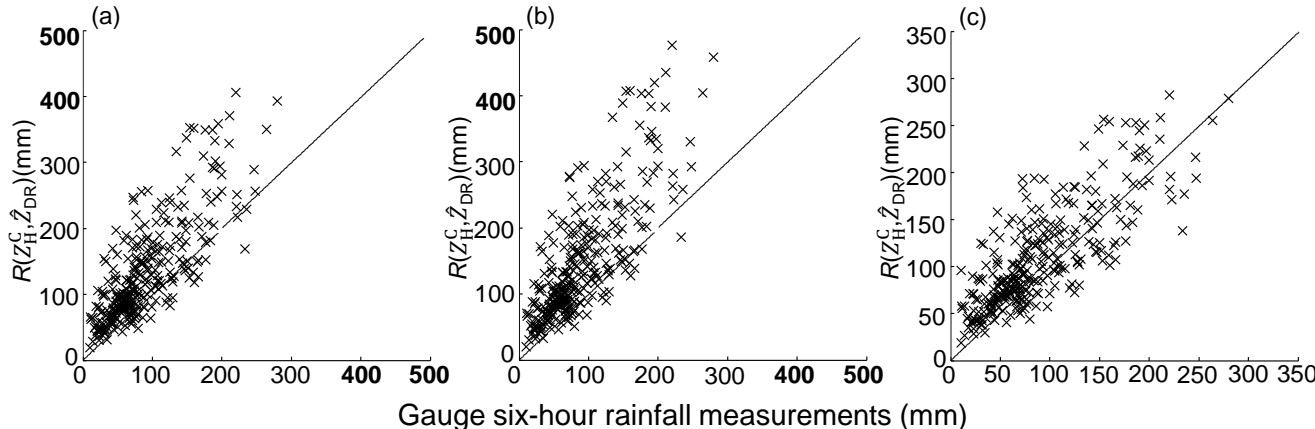

**Fig. 26. The scattergram of six-hour rainfall estimates from $R(Z_H^C, \widehat{Z}_{DR})$ versus corresponding gauge rainfall measurements at 2200 UTC, 09 August 2019: (a)-(c) are, respectively, for results calculated using Eqs. 14a-14c.**