# Peer review of "Microphysical Processes of Super Typhoon Lekima (2019) and Their Impacts on Polarimetric Radar Remote Sensing of Precipitation"

_Atmospheric Chemistry and Physics, 2022_

## Referee Comment (RC1)

The manuscript "Microphysical Characteristics of Super Typhoon Lekima (2019) and Its Impacts on Polarimetric Radar Remote Sensing of Precipitation" uses a case study of Typhoon Lekima to demonstrate a new technique for attenuation correction on Z and $Z_{DR}$. The method is interesting, but much more evidence needs to be presented. It is recommended the authors to include more dual polarization observational signatures to validate their hypothesis instead of theoretical calculation derived speculations. In addition, please revise the manuscript for grammar check.

1. Line 18, consider change "expected values" to "intrinsic values".
2. Line 19, what is $Z_H^C$ means here? What is different between $Z_H$ and $Z_H^C$? The sentence "As a result ... overestimate precipitation" is confusing, please clarify.
3. Line 20-23, do you mean the rain rate R from calibrated $Z_{DR}$ (ideally intrinsic $Z_{DR}$) and Z relationship outperforms the rest? Then is attenuation also part of the radar measurement that is needed here?
4. Line 27, for flood prediction do you mean Quantitative Precipitation Forecasts (QPF)? QPE has no forecast capabilities. In addition, one of the most important QPE operational platforms is missing here, consider adding this reference:

   Zhang, J., Howard, K., Langston, C., Kaney, B., Qi, Y., Tang, L., Grams, H., Wang, Y., Cocks, S., Martinaitis, S., Arthur, A., Cooper, K., Brogden, J., & Kitzmiller, D. (2016). Multi-Radar Multi-Sensor (MRMS) Quantitative Precipitation Estimation: Initial Operating Capabilities, Bulletin of the American Meteorological Society, 97(4), 621-638. Retrieved Jul 23, 2022, from https://journals.ametsoc.org/view/journals/bams/97/4/bams-d-14-00174.1.xml
5. Line 60, debris blockage (tree leafs, insects, and etc) can also contribute to the surface rain gauge errors.
6. Line 108, please spell out the full names for the two city locations. To emphasis the importance of this area, consider adding the population density as the background for Fig 1b.
7. Line 111, "For this aim" change to either "With this aim" or "On this aim".
8. Line 112, "WZ-SPOL radar is deployed on a mountain", what is the elevation of this mountain here, I only see almost flat surface in Fig 1a. Also, what is the VCP used here? Any negative elevation scans? I'm asking this because radar on highland could loss significant coverage at low levels.
9. Line 126-128, is Fig 1b showing all the surface rain gauges or the ones without any interruptions during this event? It is better to show only the rain gauges that will be used in this paper or at least mark the working ones with more highlighted color.
10. Line 129, where is this ratio of 10 comes from? Please add reference if there is any or explain your reasoning.
11. Line 132-133, "which made it the strongest typhoon landing China in 2019 and the third strongest landing typhoon in history of Zhejiang since 1949", this is repetitive, consider deleting.
12. Line 135, please clarify 100mm precipitation was accumulated during how long of a period.
13. Line 140, change "or" to "and".
14. Line 187-188. Both theorical analysis and observation prove that the intrinsic $Z_{DR}$ of light rain is between 0.25-0.35dB. Usually, even using light rain as a natural calibrator with intrinsic average

$Z_{DR}$ of 0.25 dB for Z = 20 – 22 dBZ may not be sufficiently accurate, see the figure below, even between 20-22 dBZ, the variability of $Z_{DR}$ is quite large. Since this is a typhoon case, consider using dry aggregated snow technique above the melting layer for calibration purpose, either QVP or RDQVP method will do the trick.

$$< Z_{DR} > = 0.25 \, dB, \qquad SD(Z_{DR}) = 0.16 \, dB$$

15. Line 203-205, between 0000 UTC, 09 August 2019 and 0000 UTC 11 August 2019 (looks like you are missing a few hours, Fig 4 stops at 1600 UTC 10 August), I don't see maximum hourly wind mostly over 20 m/s. In fact, most stations wind speed is less than 20 m/s, except around 1600 UTC 09 August.

16. Sec 3.1.1. More explanation is needed here. Especially for equation (5). Is R(t) means the maximum rain fall rate per min for each rain gauge or a period of time? How is Ct determined? It seems Ct is a constant throughout this event but different at each locations. And in all, where is this equation comes from?

17. Line 260-270, please provide some supplementary figures like RHIs (Z, $Z_{DR}$, $K_{DP}$, and Rhohv preferably) to demonstrate the existence and differences of hail/graupel between WL/YH/DT and XJ/HJ/LH stations.

18. Line 312-314, please provide some supplementary CAPPI figures for the serious underestimation of $Z_{DR}$ with larger Z near the surface. If this is simply because of size sorting near the eyewall, then it is not an issue, but how it should be. See fig 9 in Hu et al., 2020 and fig 3 in Homeyer et al., 2021.

In addition, based on Fig 11, the distance between typhoon's eyewall and half of its strongest precipitation core looks quite far away from the S-band radar here. Please also include the lowest scan 0.5 degree altitude from sea level with increasing distance line plot in the supplementary figures.

Hu, J., Rosenfeld, D., Ryzhkov, A., & Zhang, P. (2020). Synergetic Use of the WSR-88D Radars, GOES-R Satellites, and Lightning Networks to Study Microphysical Characteristics of Hurricanes,

Journal of Applied Meteorology and Climatology, 59(6), 1051-1068. Retrieved Jul 23, 2022, from https://journals.ametsoc.org/view/journals/apme/59/6/JAMC-D-19-0122.1.xml

Homeyer, C. R., Fierro, A. O., Schenkel, B. A., Didlake, A. C., Jr., McFarquhar, G. M., Hu, J., Ryzhkov, A. V., Basara, J. B., Murphy, A. M., & Zawislak, J. (2021). Polarimetric Signatures in Landfalling Tropical Cyclones, Monthly Weather Review, 149(1), 131-154. Retrieved Jul 23, 2022, from https://journals.ametsoc.org/view/journals/mwre/149/1/mwr-d-20-0111.1.xml

19. Line 324-336, please provide some supplementary figures like RHIs and CAPPI here to demonstrate your expected hydrometeors do behave as suggest in the manuscript. One of the advantages of dual polarization is to distinguish between different hydrometeors if needed and very much encouraged.

20. Fig 11. Are the panels showing 0.5 degree or CAPPI at certain altitude? Is Z composite? Please specify this information in the caption.

21. Line 365, $Z_{DR}$ is not sensitive to concentration but sensitive to size.

22. Line 375, I would agree on the big raindrops here since it is usually expected to be warm-rain dominating near the eyewall region. The authors need to provide evidence if hail/grauple do co-exist here, not from derived internal consistence but dual polarimetric signatures/or maybe lightning distribution spatially.

23. Line 384-394, this is the main selling point in this manuscript. In order to demonstrate the "large-sized raindrops tend to break apart during the falling processes but broken droplets resulted in increased concentration of raindrops for higher collision efficiency", please provide RD-QVP time series for all the dual polarization variables. If indeed breakup dominates in the warm rain levels, one should expect a downward negative gradient of Z (6th moment to the size of raindrop) and $Z_{DR}$.

24. Sec 3.3, I highly recommend the authors to include R(A) using specific attenuation result here for comparison as R(A) can be quite accurate in warm rain dominated precipitation processes. As suggested in the manuscript, R(A)'s advantages include :(i) insensitivity of AH to raindrop size distribution (DSD) variability (Ryzhkov et al., 2014); (ii) KDP is a better indicator of rain rate and liquid water content (LWC, g·m-3) than ZH since KDP is more tightly connected to the precipitation particle size distribution; (iii) R(KDP) and R(AH) inherit the immunity of ΦDP to miscalibration, attenuation, partial beam blockage, and wet radome effects.

---

## Referee Comment (RC2)

**General remarks:**

This paper examined radar quantitative precipitation estimation (QPE) in typhoon Lekima (2019) by comparing surface-based multiple-observational datasets. The discussion on selfconsistency between radar and theoretically derived variables is interesting to me. Also, overall quality-control processes in radar and disdrometer are quite beneficial in the radar community. These should be positive points in this manuscript. However, it would be more interpretation for microphysical processes using polarimetric variables. Also, some sentences (or words) should be revised. To enhance the paper, I have the following more specific suggestions.

**Specific comments:**

 The microphysical characteristics would be discussed more in the manuscript, because the part related to microphysical processes was in only one section 3.2. Otherwise, the title and abstract should be revised. The author spends most of the manuscript describing data processing (including radar and disdrometer) and comparing observed radar variables and theoretically derived variables. Also, the microphysical processes are included in the summary.
Line 83: the reviewer recommends adding more background on why the author chooses the typhoon case. This is because the typhoon case is inappropriate for radar-based QPE related to strong winds and mixed-phase hydrometeor particles. As the author mentioned, there are enormous possibilities for measurement errors in radar, rain gauge, and disdrometer. It could be helpful why the author selected the typhoon case even though there can be large measurement errors.

3. The words are quite not understandable. What is the meaning of "dynamic precipitation microphysical processes"? It seems very complicated to understand the word in the sentences. Please rewrite (or) the words. The others can find as minor suggestions.

4. The reviewer suggests that the authors can use three-dimensional structures to understand microphysical processes. Also, it would be helpful if you plot contoured frequency by altitude diagrams (CFADs) with the dual-polarimetric variable in analyzing microphysical processes. There are many works of literature to understand microphysical processes in deep convective clouds. Below is one piece of literature the authors can refer to,

- Friedrich, K., Kalina, E. A., Aikins, J., Gochis, D., & Rasmussen, R. (2016). Precipitation and Cloud Structures of Intense Rain during the 2013 Great Colorado Flood, *Journal of Hydrometeorology*, *17*(1), 27-52.

5. Lines 375-394: it would be helpful if the author could show some figures with vertical structures with polarimetric variables in their microphysical processes (i.e., accretion, coalescence, and breakup).

6. Figure 10: the differences of KDP and ZDR were quite significant in interpreting some microphysical processes. It seems that the author needs additional quality control in the radar variable.

7. Line 376-377: the melting graupels (or hail) are important in their change size for ZDR measurement. It would be helpful if the author could analyze with thermodynamic profiles.

Minor comments:

1. Line 39: this sentence would not be correct. As far as I know, dual-polarimetric variables are used for the operational purpose in radar QPE. For instance, MRMS has used available dual-polarimetric variables in radar QPE. Please see the below reference,

- Ryzhkov A, Zhang P, Bukovčić P, Zhang J, Cocks S. Polarimetric Radar Quantitative Precipitation Estimation. *Remote Sensing*. 2022; 14(7):1695. https://doi.org/10.3390/rs14071695

2. Line 99: what is the "special microphysical processes"?

3. Line 109: what is the "regional central cities"?

4. Line 127–128: please rewrite this sentence. "only gauge observations without any interruptions are utilized in this study"

5. Line 131–141: please consider that these sentences could move to the introduction section.

6. Line 156: why did the author select the threshold (Freq>50%)? I think the ground clutter could be well identified in clear air conditions.

7. Line 357: please add more interpretation about this sentence. What is the meaning of microphysical composition?

8. Figures 9-13: which radar elevation did you use for these analyses?

9. Figure 10: Lines are not clear. Please replot the figure.

---

## Author Comment (AC1)

General remarks:

This paper examined radar quantitative precipitation estimation (QPE) in typhoon Lekima (2019) by comparing surface-based multiple-observational datasets. The discussion on self-consistency between radar and theoretically derived variables is interesting to me. Also, overall quality-control processes in radar and disdrometer are quite beneficial in the radar community. These should be positive points in this manuscript. However, it would be more interpretation for microphysical processes using polarimetric variables. Also, some sentences (or words) should be revised. To enhance the paper, I have the following more specific suggestions.

Response: We thank the reviewer for the careful review and suggestions that improved this article. We have extensively revised the manuscript based on the reviewer's comments.

Specific comments:

1. The microphysical characteristics would be discussed more in the manuscript, because the part related to microphysical processes was in only one section 3.2. Otherwise, the title and abstract should be revised. The author spends most of the manuscript describing data processing (including radar and disdrometer) and comparing observed radar variables and theoretically derived variables. Also, the microphysical processes are included in the summary.

    Response: Thanks for pointing this out. In the revision, we have included more detailed discussions about the microphysical processes occurred during the landfall of Lekima, including the falling melting solid particles, the overwhelming breakup over coalescence in radar sampling volumes (above the ground), and orographic enhancement of precipitation around the GWS of YDM.

    In addition to the self-consistency/consistency described in the previous version, we have partitioned Section 3.2 into Sections 3.2.1 and 3.2.2 to highlight the microphysical features of the falling melting solid particles and the overwhelming breakup over coalescence. We have also added Section 3.3.3 to discuss the precipitation particle falling processes given the vertical gap between radar sampling volumes and the surface (i.e., Fig. 1c).

    The self-consistency/consistency not only supports the credibility of various measurements but also serves for the verification of the microphysical processes: breakup-dominated $Z_{DR}$ is relatively smaller than that of coalescence-dominated $Z_{DR}$; therefore, different $Z_{DR}$-$Z_H$ relationships are anticipated, which is in line with a previous simulation study in Kumjian and Prat (2014).

    Reference:

    Kumjian, M. R., and O. P. Prat, 2014: The Impact of Raindrop Collisional Processes on the Polarimetric Radar Variables, *Journal of the Atmospheric Sciences*, **71(8)**, 3052-3067.

2. Line 83: the reviewer recommends adding more background on why the author chooses the typhoon case. This is because the typhoon case is inappropriate for radar-based QPE related to strong winds and mixed-phase hydrometeor particles. As the author mentioned, there are enormous possibilities for measurement errors in radar, rain gauge, and disdrometer. It could be helpful why the author selected the typhoon case even though there can be large measurement errors.

    Response: Thanks for this very good comment. We agree with the reviewer that radar-based QPE can have large uncertainty due to various reasons. In fact, because of the complex microphysical processes during typhoons, the falling mixed-phase hydrometeor particles were rarely studied before, and how the dominant breakup/coalescence affects the practical performances of radar QPE is unknown. As such, we meant to disentangle this challenging problem by focusing on the microphysical variations during landfall of Lekima, which is the strongest landing typhoon in Zhejiang since 1949. More importantly, it is the first super typhoon landed on the coast of Zhejiang after the polarimetric upgrade of the WZ-SPOL radar. We wanted to use this opportunity to exploit the radar-inferred microphysical processes, including radar QPE.

3. The words are quite not understandable. What is the meaning of "dynamic precipitation microphysical processes"? It seems very complicated to understand the word in the sentences. Please rewrite (or) the words. The others can find as minor suggestions.

Response: Basically, we want to highlight the complicated (and changing) precipitation microphysics. We have rephrased this sentence in the revision. Now it reads "The impacts of dominant collision-breakup or collision-coalescence on radar QPE performance are also quantified in Section 3."

4.  The reviewer suggests that the authors can use three-dimensional structures to understand microphysical processes. Also, it would be helpful if you plot contoured frequency by altitude diagrams (CFADs) with the dual-polarimetric variable in analyzing microphysical processes. There are many works of literature to understand microphysical processes in deep convective clouds. Below is one piece of literature the authors can refer to, Friedrich, K., Kalina, E. A., Aikins, J., Gochis, D., & Rasmussen, R. (2016). Precipitation and Cloud Structures of Intense Rain during the 2013 Great Colorado Flood, *Journal of Hydrometeorology*, *17*(1), 27-52.
    Response: Thanks for this great suggestion. In fact, we started from the CFADs, which are hard to compare when we have too many of them. Another reviewer suggested using RHIs, CAPPIs, or QVPs. In the revision, we decided to use the time series of vertical polarimetric variables (Figs. 12-17) to analyze the microphysical processes during the landfall of Lekima. The main reasons are (i) RHIs are only available along one radial direction; QVPs account for the microphysical process in an average way (azimuthal average through radar measurements at high elevation angles), and their representativeness for one pixel is uncertain. (ii) Radar observes hydrometers above the ground, and the near-surface level measurements are not available (that is also why we need to use other instrument such as rain gauge and disdrometers to verify surface measurements). (iii) The combined analysis of vertical polarimetric radar time series and the surface DSD-simulated counterparts is very useful in checking the microphysical evolutions of precipitation.

5.  Lines 375-394: it would be helpful if the author could show some figures with vertical structures with polarimetric variables in their microphysical processes (i.e., accretion, coalescence, and breakup).
    Response: Vertical structures of polarimetric radar measurements (at the selected meteorological stations) are included as suggested! Thanks for this great suggestion!

6.  Figure 10: the differences of $K_{DP}$ and $Z_{DR}$ were quite significant in interpreting some microphysical processes. It seems that the author needs additional quality control in the radar variable.
    Response: Originally, we want to use this figure to emphasize the self-consistency between radar variables, and consistency between rain gauges and disdrometers, since these would demonstrate the credibility of radar and surface measurements. After seeing this comment and checking the residual differences in Fig. 10, we decided to interpret the differences in a more thorough way. After extensive analysis of the quality control in the radar variables, we concluded that both radar and surface measurements are reliable robust, the residual $Z_{DR}$ differences can be attributed to the microphysical processes. Three polarimetric radar signatures account for the dominant breakup: (i) radar-measured $Z_{DR}^{C}$-$Z_{H}^{C}$ scattergram infers breakup-dominated small size drops since small $Z_{DR}^{C}$ is expected for a given $Z_{H}^{C}$, which agrees well with the simulation in Kumjian and Prat (2014). (ii) without strong updrafts, the vertical column of $Z_{DR}$ also presented more decreasing trend in the lower atmospheric layers (see Figs. 12-17). (iii) In the time series comparison (Fig. 10), radar-measured $Z_{H}^{C}$ and $K_{DP}$ agree well with DSD-derived counterparts, but radar-measured $Z_{DR}^{C}$ is larger than DSD-derived $Z_{DR}$ at HJ, XJ, and LH due to the dominant breakup. If coalescence dominates in the vertical gap between radar measurements and the surface, the latter would be larger than the former.

    Reference:
    Kumjian, M. R., and Prat, O. P. 2014. The Impact of Raindrop Collisional Processes on the Polarimetric Radar Variables, *Journal of the Atmospheric Sciences*, **71(8)**, 3052-3067.

7.  Line 376-377: the melting graupels (or hail) are important in their change size for ZDR measurement. It would be helpful if the author could analyze with thermodynamic profiles.
    Response: We totally agree with the reviewer. Unfortunately, we (operational weather forecast office) did not collect any radiosonde observations in this interesting study domain during this event. We did have temperature and other meteorological datasets at surface meteorology stations. According to the following figure, WL, YH, and DT suffered from a temperate dropdown (the minimum temperature was 24.9°) before the center area

of the typhoon landed on WL, indicating that some cooler hydrometeors were falling. But whether these hydrometeors are ices/graupels/hail is not clear.

[Figure]

Fig. 1. The time series of temperature at six national meteorological stations on 09 August 2019 (UTC). The vertical light blue line indicated the landfalling time of Lekima.

Minor comments:

1. Line 39: this sentence would not be correct. As far as I know, dual-polarimetric variables are used for the operational purpose in radar QPE. For instance, MRMS has used available dual-polarimetric variables in radar QPE. Please see the below reference,
   - Ryzhkov A, Zhang P, Bukovčić P, Zhang J, Cocks S. Polarimetric Radar Quantitative Precipitation Estimation. *Remote Sensing*. 2022; 14(7):1695. https://doi.org/10.3390/rs14071695
   Response: Sorry for this mistake. We have fixed this in the revision.

2. Line 99: what is the "special microphysical processes"?
   Response: Based on our latest analysis, $Z_H$ and $Z_{DR}$ in radar sampling volumes above the GWS of YDM are characterized by the breakup-dominated small size drops. Raindrops transitioned to coalescence-dominated large-sized raindrops near the surface around the GWS of YDM due to topographical enhancement. We have clarified this in the revision.

3. Line 109: what is the "regional central cities"?
   Response: The regional central city is an official way of dividing cities in mainland China according to the urban system planning. It refers to provincial capital cities and sub-provincial cities with important regional significance. We have removed this term in the revision to avoid possible confusion.

4. Line 127–128: please rewrite this sentence. "only gauge observations without any interruptions are utilized in this study"
   Response: We have rephrase this sentence, now it reads "Only gauges with continuous measurements (no interruptions due to malfunction and network issues) are used in this study."

5. Line 131–141: please consider that these sentences could move to the introduction section.
   Response: Changed as suggested!

6. Line 156: why did the author select the threshold (Freq>50%)? I think the ground clutter could be well identified in clear air conditions.
   Response: Thanks for this great point. Yes. We have actually included some clear air echoes in the map in Fig. 3a to mitigate residual clutters left after applying the CMD algorithm in Hubbert et al. (2009). A threshold on

$Z_H(Z_H>0$ dBZ) and Freq>50% was used mainly to incorporate the potential fluctuations of $Z_H$ around the ground clutters.

Reference:
Hubbert, J., M. Dixon, and S. Ellis, 2009: Weather Radar Ground Clutter.. Part II: Real-Time Identification and Filtering. *J. Atmos. Ocean. Technol.*, **26**, 1181–1197.

7.  Line 357: please add more interpretation about this sentence. What is the meaning of microphysical composition?
    Response: The microphysical composition refers to either large size or small size raindrops dominant in radar sampling volumes, which will determine the distribution of radar-measured $Z_{DR}^C$ versus $Z_H^C$. We have further explained this in the revision (Line XXX in the revised manuscript).

8.  Figures 9-13: which radar elevation did you use for these analyses?
    Response: The 0.5° scan elevation angle is primarily used in these analyses. We have clarified this in the revision (Line XXX in the revised manuscript).

9.  Figure 10: Lines are not clear. Please replot the figure.
    Response: Done!

---

## Author Comment (AC2)

The manuscript "Microphysical Characteristics of Super Typhoon Lekima (2019) and Its Impacts on Polarimetric Radar Remote Sensing of Precipitation" uses a case study of Typhoon Lekima to demonstrate a new technique for attenuation correction on $Z$ and $Z_{DR}$. The method is interesting, but much more evidence needs to be presented. It is recommended the authors to include more dual polarization observational signatures to validate their hypothesis instead of theoretical calculation derived speculations. In addition, please revise the manuscript for grammar check.

Response: We sincerely thank the reviewer for the careful review of his manuscript. We have made extensive changes based on the reviewers' comments: (i) We have included more polarimetric signatures (Figs. 12-17) mainly to describe the vertical evolutions of the falling solid particles. (ii) We also rechecked the grammar to ensure the quality of this manuscript is good.

1. Line 18, consider change "expected values" to "intrinsic values".

Response: (i) We used "expected values" rather than "intrinsic values" to emphasize the differences between $\hat{Z}_{DR}$ and $Z_{DR}^{C}$, the former is calculated through radar-measured $Z_{H}^{C}$ and the latter is attenuation-corrected. Their deviation relates to the microphysical process that needed to be explained.

2. Line 19, what is $Z_{H}^{C}$ means here? What is different between $Z_{H}$ and $Z_{H}^{C}$? The sentence "As a result … overestimate precipitation" is confusing, please clarify.

Response: (i) $Z_{H}$ means horizontal reflectivity, $Z_{H}^{C}$ means attenuation-corrected horizontal reflectivity for discrimination. We introduce $Z_{H}^{M}$ and $Z_{DR}^{M}$, which stands for measured $Z_{H}$ and $Z_{DR}$ (no correction), $Z_{H}^{C}$ and $Z_{DR}^{C}$ which stands for the corrected $Z_{H}$ and $Z_{DR}$.

(ii) We have revised this part as "The twin-parameter radar rainfall estimates based on measured $Z_{H}$ ($Z_{H}^{M}$) and $Z_{DR}$ ($Z_{DR}^{M}$), and their corrected counterparts $Z_{H}^{C}$ and $Z_{DR}^{C}$, i.e., $R(Z_{H}^{M}, Z_{DR}^{M})$ and $R(Z_{H}^{C}, Z_{DR}^{C})$, both tend to overestimate rainfall around the GWS of YDM, mainly ascribed to the unique microphysical process in which the breakup-dominated small-sized drops above transition to the coalescence-dominated large-sized drops falling near the surface." (Lines 20-24). The abnormal overestimation $R(Z_{H}^{C}, Z_{DR}^{C})$ is strange, but it indeed occurred and it related to an microphysical process.

3. Line 20-23, do you mean the rain rate R from calibrated $Z_{DR}$ (ideally intrinsic $Z_{DR}$) and Z relationship outperforms the rest? Then is attenuation also part of the radar measurement that is needed here?

Response: (i) Yes. $R(Z_{H}^{C}, \hat{Z}_{DR})$ outperforms the rest according to $E_{NMA}$ and $E_{RMS}$ at least in this case. (ii) The attenuation on Z is necessary, therefore, we used $Z_{H}^{C}$ which means attenuation-corrected $Z_{H}$.

4. Line 27, for flood prediction do you mean Quantitative Precipitation Forecasts (QPF)? QPE has no forecast capabilities. In addition, one of the most important QPE operational platforms is missing here, consider adding this reference:
Zhang, J., Howard, K., Langston, C., Kaney, B., Qi, Y., Tang, L., Grams, H., Wang, Y., Cocks, S., Martinaitis, S., Arthur, A., Cooper, K., Brogden, J., & Kitzmiller, D. (2016). Multi-Radar MultiSensor (MRMS) Quantitative Precipitation Estimation: Initial Operating Capabilities, Bulletin of the American Meteorological Society, 97(4), 621-638. Retrieved Jul 23, 2022, from https://journals.ametsoc.org/view/journals/bams/97/4/bams-d-14-00174.1.xml

Response: (i) "for flood prediction" does not mean Quantitative Precipitation Forecasts (QPF). The radar rainfall estimates field needs to be inputted into some hydrological models or treated as an initial radar rainfall field of some extrapolate algorithms for the QPF. So radar QPE indirectly serves for flood prediction. (ii) Thanks for the reminder, Zhang et al., 2016 was indeed missed during the revision process, and we have added it.

5. Line 60, debris blockage (tree leafs, insects, and etc) can also contribute to the surface rain gauge errors.
Response: Thanks for the suggestions, and we have added these factors.

6. Line 108, please spell out the full names for the two city locations. To emphasize the importance of this area, consider adding the population density as the background for Fig 1b.

Response: Thanks. (i) Wenzhou and Taizhou were mentioned for the first time in Lines 79 and 90 (the original manuscript), so we wrote their abbreviations here. (ii) we have added the description of population density into this part.

7.  Line 111, "For this aim" change to either "With this aim" or "On this aim".
    Response: Changed as suggested!

8.  Line 112, "WZ-SPOL radar is deployed on a mountain", what is the elevation of this mountain here, I only see almost flat surface in Fig 1a. Also, what is the VCP used here? Any negative elevation scans? I'm asking this because radar on high land could loss significant coverage at low levels.
    Response: Thanks for your constructive question, and it enlightens our understanding of the microphysical process around the GWS of YDM.
    (i) The height of this radar is 735 m, on a little hill, and the star masks the DEM of the mountain, and we had removed the star.
    (ii) VCP 21 is used here with the lowest elevation angle of 0.5° (described in Section 2.2). No negative elevation angles were used; if the negative elevation angles were used, the clutter (i.e., ground or sea clutters) and blockage (i.e., by YDM and KCM) will significantly degrade radar measurements (i.e., $Z_H$, $Z_{DR}$, $K_{DP}$, $\rho_{HV}$). It is not high enough to avoid such contaminations from these two issues).

9.  Line 126-128, is Fig 1b showing all the surface rain gauges or the ones without any interruptions during this event? It is better to show only the rain gauges that will be used in this paper or at least mark the working ones with more highlighted color.
    Response: Thanks. The stations marked are all working ones. "The ones without any interruptions" means interruptions in an hour. If gauge rainfall measurements of one station temporally interrupt in one hour, the other measurements without interruption (in another hour) are still used. We have revised this sentence for clarity.

10. Line 129, where is this ratio of 10 comes from? Please add reference if there is any or explain your reasoning.
    Response: (i) We first saw this ratio in Marzen 2004: two ratios, 5 and 10, are suggested, but 10 or (or less 0.1 than for intercomparison) is used to remove the strongly suspected gauge rainfall recordings when gauge rainfall recording exceeds 1 mm. It means that if the rain gauge has measured 1 mm (in an hour), radar rainfall needs to exceed 10 mm (significant rainfall) to remove this gauge measurement. In this tropical rain situation (evaporation can be negligible), most gauge measurements will be kept because this ratio is hard to exceed.

    Marzen, J. L., 2004: Development of a Florida high-resolution multisensor precipitation dataset for 1996-2001–Quality control and verification. MS thesis, Department of Meteorology, The Florida State University, 86 pp.

11. Line 132-133, "which made it the strongest typhoon landing China in 2019 and the third strongest landing typhoon in history of Zhejiang since 1949", this is repetitive, consider deleting.
    Response: Changed as suggested!

12. Line 135, please clarify 100 mm precipitation was accumulated during how long of a period.
    Response: Thanks. Lekima affected mainland China for 44 hours; this 100 mm precipitation refers to the area with rainfall exceeding 100 mm during this period. We add this information to this sentence (Lines 96-97).

13. Line 140, change "or" to "and".
    Response: Changed as suggested!

14. Line 187-188. Both theorical analysis and observation prove that the intrinsic $Z_{DR}$ of light rain is between 0.25-0.35dB. Usually, even using light rain as a natural calibrator with intrinsic average $Z_{DR}$ of 0.25 dB for $Z = 20 - 22$ dBZ may not be sufficiently accurate, see the figure below, even between 20-22 dBZ, the variability of $Z_{DR}$ is quite large. Since this is a typhoon case, consider using dry aggregated snow technique above the melting layer for calibration purpose, either QVP or RDQVP method will do the trick.
    Response: Thanks for your suggestion.

(i) The calibration of $Z_{DR}$ of light rain is usually carried in zenith mode and the falling raindrops tend to exhibit $Z_{DR} \approx 0dB$ in this situation. Still, $Z_{DR}$ bias recovers after long-term consecutive scanning. So additional $Z_{DR}$ bias correction is necessary.

(ii) $Z_{DR}$ of the dry aggregated snow is efficient if one volume gate is fully filled with aggregated snow; $Z_{DR} \approx 0dB$ for aggregates is an important basis for this method. But it is too over-confident, and it has its own flaws; several fundamental issues of QVP need to be demonstrated here:

    (a): the shapes of solid particles above the melting layer are more complex than that of raindrops; dry aggregated snow is not the only form of solid particles. If you use a hydrometeor identification in advance, you will find that any identification algorithm requires that $Z_{DR}$ is well calibrated. Therefore, a potential deadlock: hydrometeor identification requires that $Z_{DR}$ is well calibrated, but $Z_{DR}$ calibration needs aggregated snow to be identified first.

    (b) NUBF (non-uniform beam filling) effects can be significant on high elevations for QVP, which present little $Z_H^M$ but large $Z_{DR}^M$ at the far side of radial profile ($Z_{DR}^M$ may be larger than $\hat{Z}_{DR}$). Positive $Z_{DR}$ bias is anticipated in this situation, which will misguide the $Z_{DR}$ bias.

    (c) Wet radome effects may only affect the radial directions with strong horizontal winds (rain pours on one side of the radar radome and flows downward). $Z_{DR}$ bias on QVP (due to average) might be not significant and cannot represent the $Z_{DR}$ bias on the low elevation angles.

(iii) Since the $Z_{DR}$ calibration technique is questioned, some background considerations have to be further described here. It is enlightened by Bringi et al.2001 that all $Z_{DR}$ at the far side of one radial profile is expected to approximate 0 dB if its "intrinsic" $Z_H < 20dBZ$ (it usually refers to $Z_H^C$). Accordingly, $Z_{DR}^C$ on one radial profile should approximate to their $\hat{Z}_{DR}$ not only in the rainstorm area ($Z_H^C > 20dBZ$), but also at the far side of one radial profile where $\hat{Z}_{DR}$ is close to 0dB; $\hat{Z}_{DR}$ will not become very large conditioned by $Z_H^C < 20dBZ$. Appropriate $Z_{DR}$ bias adjustment effectively helps in this $Z_{DR}$ approximation along the whole range profile.

$$< Z_{DR} > = 0.25\, dB, \qquad SD(Z_{DR}) = 0.16\, dB$$

15. Line 203-205, between 0000 UTC, 09 August 2019 and 0000 UTC 11 August 2019 (looks like you are missing a few hours, Fig 4 stops at 1600 UTC 10 August), I don't see maximum hourly wind mostly over 20 m/s. In fact, most stations wind speed is less than 20 m/s, except around 1600 UTC 09 August

Response: Thanks, this part has been distorted during the revision process before submitting this paper. We have revised this sentence and "Nearly all" is removed. Now it reads "These measurements were variously affected by the strong winds, with the hourly maximum wind speed exceeding 20 m·s⁻¹, as depicted in Fig. 4". Lines 225-226.

16. Sec 3.1.1. More explanation is needed here. Especially for equation (5). Is R(t) means the maximum rain fall rate per min for each rain gauge or a period of time? How is Ct determined? It seems Ct is a constant throughout this event but different at each locations. And in all, where is this equation comes from?
Response: Thanks for your question. $R_T$ and $C_T$ have no additional special physical meanings. They just serve for the convenient comparison of different rainfall timeseries.
(i) No. $R_T$ does not mean the maximum rain fall rate per min for each rain gauge or a period of time. It is a hourly rainfall threshold, which is (manually) set a little larger (to avoid the overlap with the other timeseries) than the maximum DSD-derived hourly rainfall (after mitigating the wind effects) during the precipitation period of each station.
(ii) $C_T$ is not constant, and it is (manually) set for each station, and for the convenient comparison with other rainfall timeseries in one figure. For example, $C_T$ of YH and WL are both extremely large (800 and 500), but they are relatively smaller at the other stations.
(iii) $R_T$ and $C_T$ are decided both manually, we designed this equation and $R_T$ and $C_T$ serve for organizing DSD-derived rainfall without any QC and DSD-derived rainfall after QC in the same Figure. Because the former (in the following Fig. 1) is far larger ($\gg$) than the latter.

[Figure]

Fig. 1. DSD-derive rainfall timeseries before QC processing.

17. Line 260-270, please provide some supplementary figures like RHIs ($Z$, $Z_{DR}$, $K_{DP}$, and Rhohv preferably) to demonstrate the existence and differences of hail/graupel between WL/YH/DT and XJ/HJ/LH stations.
Response: Thanks sincerely for your constructive suggestions, and they indeed enlighten us to improve the quality of the paper. Considering the advantages of VPR, RHI and CAPPI, the combined utilization of vertical polarimetric radar timeseries upon the stations and the surface DSD-simulated counterparts may better interpret what had occurred during the landfall of Lekema. Therefore, we have added six vertical radar measurements ($Z_H$, $Z_{DR}$, $K_{DP}$, and $\rho_{HV}$ in Figs.12-17) to detail the polarimetric signatures of falling solid particles, and a new part is organized as Section 3.2.1.

18. Line 312-314, please provide some supplementary CAPPI figures for the serious underestimation of $Z_{DR}$ with larger Z near the surface. If this is simply because of size sorting near the eyewall, then it is not an issue, but how it should be. See fig 9 in Hu et al., 2020 and fig 3 in Homeyer et al., 2021.

In addition, based on Fig 11, the distance between typhoon's eyewall and half of its strongest precipitation core looks quite far away from the S-band radar here. Please also include the lowest scan 0.5 degree altitude from sea level with increasing distance line plot in the supplementary figures.

Hu, J., Rosenfeld, D., Ryzhkov, A., & Zhang, P. (2020). Synergetic Use of the WSR-88D Radars, GOES-R Satellites, and Lightning Networks to Study Microphysical Characteristics of Hurricanes, Journal of Applied Meteorology and Climatology, 59(6), 1051-1068. Retrieved Jul 23, 2022, from https://journals.ametsoc.org/view/journals/apme/59/6/JAMC-D-19-0122.1.xml

Homeyer, C. R., Fierro, A. O., Schenkel, B. A., Didlake, A. C., Jr., McFarquhar, G. M., Hu, J., Ryzhkov, A. V., Basara, J. B., Murphy, A. M., & Zawislak, J. (2021). Polarimetric Signatures in Landfalling Tropical Cyclones, Monthly Weather Review, 149(1), 131-154. Retrieved Jul 23, 2022, from https://journals.ametsoc.org/view/journals/mwre/149/1/mwr-d-20-0111.1.xml

Response: Thanks, and we have read the figures in your recommended papers.

(i) CAPPI is constructed by interpolating radar measurements from different elevation angles (not only the lowest elevation angle will be utilized, more uncertain). Its coverage is limited at lower altitudes and its capability to represent the $Z_H$ and $Z_{DR}$ near the surface is limited.  Most pixels in Fig.11 come from the lowest elevation angles of 0.5° may be better, radar measurements from the second elevation angle of 1.5° only occupies the limited GC-masked pixels (as depicted in Fig.3b).

(ii) HSS only partly accounts for the location/position inconsistency of large values of $Z_H$, $Z_{DR}$ and $K_{DP}$, but it cannot account for the large deviation between $Z_{DR}^C$ and $\hat{Z}_{DR}$ when the large values of $Z_H$, $Z_{DR}$ and $K_{DP}$ coincide. The overwhelming breakup over coalescence account for this phenomenon. Resultantly, more small-sized raindrops for a given $Z_H^C$ but their intrinsic $Z_{DR}$ is relatively smaller due to dominant breakup than dominant coalescence.
(iii) We added the lowest scan 0.5° altitude from sea level with an increasing distance line plot as Fig. 2c, adding the distance and altitude differences of each meteorological station (Fig. 2c).

19. Line 324-336, please provide some supplementary figures like RHIs and CAPPI here to demonstrate your expected hydrometeors do behave as suggest in the manuscript. One of the advantages of dual polarization is to distinguish between different hydrometeors if needed and very much encouraged.
Response: Thanks for your suggestion.  We have added consecutive time-series of the vertical polarimetric radar measurements (three-dimensional radar measurement which consists of more CAPPI at different altitude layers) upon each station to detail the possible falling hydrometeors, as depicted in Figs.12-17. In this way, the microphysical evolutions of precipitation upon each station and the connection between falling solid particles and their impacts on the surface measurements can be clearly demonstrated.

20. Fig 11. Are the panels showing 0.5 degree or CAPPI at certain altitude? Is Z composite? Please specify this information in the caption.
Response: Thanks. They belong to hybrid radar measurements; most pixels are from radar measurements on the elevation angle of 0.5°, only the masked pixels in Fig. 3b are from the elevation angle of 1.5°.

21. Line 365, $Z_{DR}$ is not sensitive to concentration but sensitive to size.
Response: Changed as suggested!

22. Line 375, I would agree on the big raindrops here since it is usually expected to be warm-rain dominating near the eyewall region. The authors need to provide evidence if hail/grauple do coexist here, not from derived internal consistence but dual polarimetric signatures/or maybe lightning distribution spatially.
Response: Thanks. The time series of the vertical polarimetric variables upon WL provide evidence of falling solid particles. In Fig.12, there emerged an important signature that $\rho_{HV}<0.84$ gradually subsided, accompanying a subsiding near-zero $Z_{DR}$.

23. Line 384-394, this is the main selling point in this manuscript. In order to demonstrate the "large-sized raindrops tend to break apart during the falling processes but broken droplets resulted in increased concentration of raindrops for higher collision efficiency", please provide RD-QVP time series for all the dual polarization variables. If indeed breakup dominates in the warm rain levels, one should expect a downward negative gradient of Z (6th moment to the size of raindrop) and $Z_{DR}$.
Response: Thanks for your comment.
(i) RD-QVP may not be a good choice due to the overall average operations using all scanning measurements on one elevation angle. (ii) We summarized three evidences for the dominant breakup:

(a) Radar-measured $Z_{DR}^C$-$Z_H^C$ scattergram tends to breakup-dominated little-sized drops since little $Z_{DR}^C$ is anticipated for a given $Z_H^C$, which agrees well with the simulation result of Kumjian and Prat. 2014, in which Fig. 8 compares the breakup-only and coalescence-only $Z_{DR}$-$Z_H$ relations and breakup-dominated $Z_{DR}$ is smaller than coalscence-dominated $Z_{DR.}$

(b) Without significant updrafts, the vertical $Z_{DR}$ columns also presented more decreasing $Z_{DR}$ signatures downward to the lower atmospheric layers in Figs.12-17.

(c) In time series comparison (Fig. 10), radar-measured $Z_H^C$ and $K_{DP}$ agree well with DSD-derived counterparts, but radar-measured $Z_{DR}^C$ is larger than DSD-derived $Z_{DR}$ at HJ, XJ, and LH. If coalescence dominated in the vertical gap between radar measurements and the surface, the latter should be larger than the former (but the converse is true).

Kumjian, M. R., and Prat, O. P. 2014. The Impact of Raindrop Collisional Processes on the Polarimetric Radar Variables, Journal of the Atmospheric Sciences, 71(8), 3052-3067

24. Sec 3.3, I highly recommend the authors to include R(A) using specific attenuation result here for comparison as R(A) can be quite accurate in warm rain dominated precipitation processes. As suggested in the manuscript, R(A)'s advantages include :(i) insensitivity of AH to raindrop size distribution (DSD) variability (Ryzhkov et al., 2014); (ii) KDP is a better indicator of rain rate and liquid water content (LWC, g·m-3) than ZH since KDP is more tightly connected to the precipitation particle size distribution; (iii) R(KDP) and R(AH) inherit the immunity of ΦDP to miscalibration, attenuation, partial beam blockage, and wet radome effects.

Response: Thanks for your recommendation.

(i) We indeed described the advantage in the introduction; however, you are over-confident in $R(A_H)$, everything has two sides. We just did not emphasize its disadvantages which is beyond the scope of this paper: $A_H$ and α are usually simultaneously derived, and the sensitivity of α to temperature cannot be denied, which occurs with the ascending altitude of the propagation of one radar beam. We have to add this issue to avoid confusing the other readers.

(ii) In Gou et al. (2018) and Gou et al. (2019), we tested the practical performances of $R(A_H)$ separately in two cases. Both exhibited an overall overestimation trend and performed inferior to $R(K_{DP})$. It's why we didn't incorporate $R(A_H)$ into this paper (similar results were anticipated). Our results do not deny the advantage of $R(A_H)$, $R(A_H)$ needs to be optimized. However, we notice that the suggested optimization $R(A_H)$ in Ryzhkov et al. 2022 relies on $α(Z_{DR})$, but breakup-dominated $Z_{DR}$ is different from coalescence-dominated $Z_{DR}$, as indicated in this paper and the simulation results in Kumjian and Prat.2014. Therefore, we need to do more work for such an optimization, which is far beyond the scope of the topic of this paper.

(iii) It is not our primary purpose to determine which radar QPE algorithm performs best among all possible forms. The utilization of $R(Z_H, Z_{DR})$ plays an important role in verifying the dominant breakup over the coalescence and the transition from breakup to coalescence around the GWS of YDM.

Gou Y, Chen H, Zheng J. 2019. Polarimetric Radar Signatures and Performance of Various Radar Rainfall Estimators during an Extreme Precipitation Event over the Thousand-Island Lake Area in Eastern China. Remote Sensing.,11(20):2335. https://doi.org/10.3390/rs11202335.

Ryzhkov A, Zhang P, Bukovčić P, Zhang J, Cocks S. Polarimetric Radar Quantitative Precipitation Estimation. *Remote Sensing*. 2022; 14(7):1695. https://doi.org/10.3390/rs14071695.

---

## Referee Report (RR1)

General remarks:

Thank you for the significant change to the manuscript. It has been greatly improved and allows the reader to follow the results and scientific story more easily. The reviewer suggests the paper can be worthy of prompt publication.

---

## Author Response (AR2)

**Responses to Review Comments on ACP-2022-495**

Again, we would like to express our sincere thanks to the Handling Editor and anonymous reviewers for their great comments and suggestions. We have further revised the manuscripts based on the reviewers' comments. Detailed responses are provided below.

1. Regarding $Z_{DR}$ calibration, the current U.S. NEXRAD is following the order of: (1) Bragg Scattering, (2) Dry Snow, and (3) Light Rain method with weighting number (3) to be the lowest consider its lowest accuracy and largest standard deviation. Since this is a typhoon case, Bragg scattering is not an option, dry snow method should be the best choice left. I'd agree you want to make sure dry snow is dominated in order to use this method. The idea of using RDQVP (for better coverage at low levels) can significantly improve melting layer top detection and avoid low level non-meteorological noise. A typhoon/hurricane case can be benefited to have quite large areas of stratiform precipitation for a RDQVP profile to be built. Anyway, this is a suggestion but not required for $Z_{DR}$ calibration here.
Response: We truly appreciate this great suggestion. We totally agree with the reviewer that RDQVP built from large areas of stratiform precipitation during a hurricane event could potentially improve melting layer top detection and avoid low level non-meteorological noise. We will consider RDQVP in our future work. Thanks again for the suggestion.

2. Referring Kumjian and Prat (2014) is a nice touch. But some RHIs of Z, $Z_{DR}$, $K_{DP}$, and Rhohv would be more direct evidence to show the stronger break-up processes here.
Response: We thank the reviewer for the kind words. Indeed, Kumjian and Prat (2014) is a best reference we should cite and *compare* in this particular study. We also agree with the reviewer that RHIs would be useful to show the stronger break-up processes here. In Fig. 1 below, an example with simultaneously decreasing $Z_H$, $Z_{DR}$, and $K_{DP}$ toward the surface (highlighted by the rectangular boxes). Nevertheless, as the reviewer may be aware, the stronger break-up processes (i.e., breakup-only processes) might not be always perceived through real RHI measurements since the collision process is usually controlled by the coalescence-breakup balance; some large-sized drops may form due to coalescence, but such an increase will also be balanced by break-up. Another fundamental process is that larger-sized drops fall at a faster speed than small-sized drops, which can be validated through Eq. 3 along with disdrometer measurements. Combining these two processes, $Z_{DR}$ may also increase toward the surface (but might not exceed $\hat{Z}_{DR}$) in situations with coalescence-breakup balance.

Therefore, it might be easier to differentiate the dominant break-up or coalescence through the difference between $Z_{DR}$ and $\hat{Z}_{DR}$, and drop size increase is not enough to make $Z_{DR}$ approximate or exceed $\hat{Z}_{DR}$ (more small-sized drops in DSD dataset) if break-up overwhelms coalescence; conversely, $Z_{DR}$ may be approximate or exceed $\hat{Z}_{DR}$ (due to more large-sized drops) if coalescence overwhelms break-up. In this way, more pixels are found with $Z_{DR}^C < \hat{Z}_{DR}$ in the following Fig. 2 (below the ML), particular within the range of 60 km.

3. One last concern and this may be picky, but I would also agree with the other reviewer's opinion that a strong typhoon case is not appropriate for radar QPE comparison considering all the biases from observation side. Admitted it is high impact case, but this also make analysis of individual microphysical process much more complicated.
Response: We thank the reviewer for this very good point. As mentioned in the main manuscript, we meant to select a challenging high-impact event for this study. Motivated by the reviewer's comments, we have added more discussions about the limitations of current research on microphysical process, as well as radar QPE comparison. Just for the reviewer's information, we have been putting significant efforts into the microphysical process and radar QPE in other similar (and challenging) events. Indeed, it is rather difficult to justify all the findings based on polarimetric radar observations, especially when there is a lack of in-situ measurements (which, unfortunately, is another challenge). We hope the reviewer is Okay with such an investigation.

[Figure]

Fig. 1. RHIs of WZ-SPOL radar observations pointing to the DT station at 1759 UTC, 09 August 2019: (a) $Z_H^C$ (dBZ), (b) $Z_{DR}^C$ (dB),(c) $K_{DP}$ (deg/km), (d) $\rho_{HV}$.

Fig. 2. RHIs of WZ-SPOL radar observations pointing to the DT station at 1759 UTC, 2019 (a) $Z_{DR}^C$ (dB), (b) $\hat{Z}_{DR}$ (dB).